# Decoding Generalization from Memorization in Deep Neural Networks

## Abstract

Overparameterized Deep Neural Networks that generalize well have been key to the dramatic success of Deep Learning in recent years. The reasons for their remarkable ability to generalize are not well understood yet. It has also been known that deep networks possess the ability to memorize training data, as evidenced by perfect or high training accuracies on models trained with corrupted data that have class labels shuffled to varying degrees. Concomitantly, such models are known to generalize poorly, i.e. they suffer from poor test accuracies, due to which it is thought that the act of memorizing substantially degrades the ability to generalize. It has, however, been unclear why the poor generalization that accompanies such memorization, comes about. One possibility is that in the process of training with corrupted data, the layers of the network irretrievably re-organize their representations in a manner that makes generalization difficult. The other possibility is that the network retains significant ability to generalize, but the trained network somehow "chooses" to readout in a manner that is detrimental to generalization. Here, we provide evidence for the latter possibility by demonstrating, empirically, that such models possess information in their representations for substantially improved generalization, even in the face of memorization. Furthermore, such generalization abilities can be easily decoded from the internals of the trained model, and we build a technique to do so from the outputs of specific layers of the network. We demonstrate results on multiple models trained with a number of standard datasets.

## 1 Introduction

Prior to the advent of Deep Learning, the conventional wisdom for long[1], was that in building a predictive model, the model should have as few parameters as possible and this number should certainly be less than the number of training samples that one was fitting. The dogma was that, otherwise, the model would exactly fit the training points, but invariably generalize poorly to unseen data, i.e. overfit. This intuition was also largely borne out by the models of the day. Modern Deep Learning, however, has gone on to show the opposite, namely that overparameterized models not only don't necessarily overfit, but that they can generalize remarkably well to unseen data. However, over a decade later, we still do not satisfactorily understand why this is so. Interestingly, it has been shown (Zhang et al., 2017; 2021) that when one shuffles class labels of data points from standard training datasets to varying degrees, deep networks can still have high/perfect training accuracy on such corrupted training data; however, this appears to typically be accompanied by poor performance on unseen test data (that have true labels). This phenomenon has been called *memorization*, since it is thought that the model rote-learned the training data without acquiring the ability to generalize to unseen examples. It has been known (Arpit et al., 2017) that early on in training, such models start off by having better generalization ability; however generalization worsens as training accuracy increases across epochs of training. In such trained models, there have been efforts to spatially localize the origin of memorization. While certain studies (Cohen et al., 2018; Stephenson et al., 2021) have suggested that memorization occurs in the latter layers of the network, more recent work (Maini et al., 2023) suggests that memorization occurs in all layers of the network, and is

---

[1]von Neumann famously said, "With four parameters I can fit an elephant, and with five I can make him wiggle his trunk." (Dyson et al., 2004)

largely attributable to a few units. Indeed, there have been suggestions that for networks trained on real-world data (that isn't deliberately corrupted), memorization could play a critical role in their extraordinary performance (Feldman, 2020). While there is early work studying memorization in this setting (Feldman & Zhang, 2020), there are methodological issues that have slowed progress. Progress on understanding memorization in corrupted models could enable a better understanding of memorization and generalization in deep networks trained on real-world data.

An open question arising in this context is about the detailed mechanisms that lead to poor generalization in models trained with shuffled labels. A natural hypothesis governing such mechanisms, stated informally, is that, during training, the network organizes class-conditional representations, in a manner suited to doing well on the (corrupted) training data. Since this data is significantly noisy, on being given unseen data with true labels, it fundamentally lacks the ability to have good prediction performance, leading to poor generalization. An alternative hypothesis is that layerwise representations in the network retain the ability to generalize easily, but that the network somehow chooses to readout in favor of high training accuracy in a manner that incidentally causes poor generalization performance. A corollary to this alternative hypothesis is that one ought to be able to construct a decoder for the outputs of the network's layers that does well on the corrupted training data, while also having good generalization, i.e. high test set accuracy.

Here, surprisingly, we show evidence for this alternative hypothesis. In particular, we study the organization of subspaces of class-conditioned training data on layerwise outputs of a number of deep networks. We estimate these subspaces using Principal Components Analysis (PCA). In order to remain agnostic to the information decoded by subsequent layers, we build a simple classifier that leverages the geometry of the layer output of an incoming datapoint, relative to these class-conditioned subspaces. Specifically, we measure the angle between this output vector and its projection on each of these class-conditioned subspaces and the classifier predicts this datapoint's class to be the class whose subspace has the minimum such angle. A useful consequence of this formulation is the following. The existence of class-conditioned subspaces estimated from training data on which the aforementioned minimum angle subspace classifier has good test accuracy implies that the deep network can, in principle, generalize well.

**Main Contributions**

1. For models trained using standard methods & datasets with training data corrupted by label noise, while the model has poor test accuracy, we can build a simple classifier with dramatically better test accuracy that uses only the model's hidden layer outputs obtained for the (corrupted) training set.

2. For the aforementioned models, if the true training class labels are known post hoc, i.e. after the model is trained, we can build a simple classifier, with significantly better generalization performance than in (1). This is true, in many cases, even for models where training class labels are shuffled with equal probability. This demonstrates that the layers of the network maintain representations in a manner that is amenable to straightforward generalization to a degree not previously recognized.

3. On the other hand, we asked if a model trained on the true training labels similarly retained the capability to memorize easily. Adapting our technique to this setting, we find that in a few cases, we can extract a high degree of memorization. The same classifier sometimes exhibits high test accuracy (on the true test labels), which further supports the idea that generalization can co-exist with memorization.

## 2 RELATED WORK

The idea of probing intermediate layers of Deep Networks isn't new. For example, (Montavon et al., 2011; Alain & Bengio, 2018) do so by using kernel PCA & linear classifiers respectively. However, this approach has not been used to investigate memorization. Indeed, (Alain & Bengio, 2018) explicitly avoid examining memorized networks from (Zhang et al., 2017) because they thought such probes would inevitably overfit. Our results are therefore especially surprising in this context.

There is evidence that DNN's learn simple patterns first, before memorizing (Arpit et al., 2017), & DNNs learn lower frequencies first (Belrose et al., 2024). (Stephenson et al., 2021) study memorized models, concluding that memorization happens in later layers, since rewinding early layer weights to

their early stopping values recovers some generalization, but rewinding later layer weights doesn't. On the contrary, our results suggest that later layers in most models investigated retain significant ability to generalize, & we demonstrate this without modifying the weights of the trained network.

There is an important line of theoretical work in deep linear models (Saxe et al., 2013) where the question of generalization has been studied. In this context, (Lampinen & Ganguli, 2018) offer a theoretical explanation for the phenomenon of memorization in networks trained with noisy labels.

Experiments towards understanding training dynamics across layers using different Canonical Correlation Analysis have be explored (Raghu et al., 2017) and in various generalized and memorized networks is analyzed (Morcos et al., 2018). Centered Kernel Alignment in different random initializations by (Kornblith et al., 2019) and network similarity between model trained with same data and different initialization is examined by (Wang et al., 2018). Also experiments related to using measures of the representational geometry towards understanding dynamics of layerwise outputs (Chung et al., 2016; Cohen et al., 2020) and different measures such as curvature (Hénaff et al., 2019) Dimensionality which express the structures within the representations (Sussillo & Abbott, 2009; Farrell et al., 2019; Gao & Ganguli, 2015; Litwin-Kumar et al., 2017; Bakry et al., 2015; Cayco-Gajic & Silver, 2019; Yosinski et al., 2014; Stringer et al., 2019; Yosinski et al., 2014) have also been explored.

To deal with label noise, many heuristics have been explored (Khetan et al., 2017; Scott et al., 2013; Reed et al., 2014; Zhang & Sabuncu, 2018; Malach & Shalev-Shwartz, 2017) & for classification task see (Frénay et al., 2014; Ren et al., 2018; Menon et al., 2018; Shen & Sanghavi, 2019). For over parameterized models, (Li et al., 2020) shows that the memorized network weights are far away from the initial random state in order for them to overfit the noisy labels. (Stephenson & Lee, 2021) propose a theoretical model for epochwise double descent that suggests that for small-sized models, moderate amounts of noise can cause generalization error to dip later on in training.

## 3 METHODOLOGY

Using the organization of subspaces of class-conditioned training data on layerwise outputs of deep networks, we build a Minimum Angle Subspace Classifier (MASC) with the following steps:

**Creation of subspaces**: For a specific layer, we estimate subspaces for each class. The class-conditioned training data subspaces on layerwise outputs of deep networks are computed using PCA. If the empirical mean of the class-conditioned data isn't zero, PCA in effect, will provide us an affine space, i.e. a linear space that doesn't pass via the origin. However, we have determined subspaces – which are linear spaces passing through the origin – here rather than affine spaces. In order to do so, we add the negative of each sample to the dataset so it is guaranteed to have empirical mean be zero, before running PCA. This created dataset is sent to the PCA algorithm to calculate PCA components for a certain percentage of variance explained in the dataset. The span of these PCA components is the subspace $S$. We illustrate the process for a Multi-layer Perceptron (MLP) model in Figure 4 in the Appendix.

**Projection of the data point**: Layer output of an incoming data point is projected onto these class-specific subspaces.

**Label assignment using minimum angle**: For every data point, the angle between the original data point and projected data point for each class is calculated. The Minimum Angle Subspace Classifier (MASC) assigns to the datapoint, the label of the subspace having the minimum angle with the original data point.

While the subspaces are estimated using the training data alone, accuracy of the Minimum Angle Subspace Classifier is determined for the training data and the testing data separately. This process is followed for all the layers in the network independently. MASC is using labels of the dataset while creating the class-specific subspaces. For experiments in Section 4, MASC uses corrupted training labels whereas in Section 5, MASC uses true training labels to create class-specific subspaces. See Appendix A.7 for MASC algorithm. We have used 99% as the percentage of variance explained, unless otherwise mentioned.

## 3.1 Experimental Setup

We have used multiple models and datasets, namely Multi-layer Perceptron (MLP) trained on MNIST (Deng, 2012) and CIFAR-10 (Krizhevsky, 2009) datasets, Convolutional Neural Networks (CNN) [2] trained on MNIST, Fashion-MNIST (Xiao et al., 2017), and CIFAR-10 and AlexNet (Krizhevsky et al., 2012) trained on CIFAR-100 (Krizhevsky, 2009) and Tiny ImageNet (Moustafa, 2017). We have trained these models with training data having true labels ("generalized models") as well as separately using training data with labels shuffled to varing degrees ("memorized models") (Zhang et al., 2021).

For memorized models, when we say we train it with corruption degree $p$, we mean that with probability $p$, we attempt changing the label for a training datapoint. Changing the labels happens uniformly at random. Note that this may result in the label remaining the same; therefore the expected fraction of datapoints whose labels changed are $p - p/c$ where $c$ is the number of classes. So, this would mean that for corruption degrees of 20% , 40%, 60%, 80%, 100% the expected percentage of training datapoints with changed labels is 18%, 36%, 54%, 72%, 90% respectively, when $c = 10$. We have run experiments for values of $p$ being 0% (generalized model), 20% , 40%, 60%, 80%, 100% (memorized models).

A summary of the models and datasets with training set size and number of parameters is in Table 1 in the Appendix. The average training and testing accuracies of all the models over three runs are shown in Table 2 and 3 in section A.3. More details of these models, hyperparameters & training are available in Section A.2. Following standard practice in probing memorized models (e.g. (Stephenson et al., 2021)), we do not use explicit regularizers such as Dropout or batchnorm, or early stopping, unless otherwise mentioned, as a result of which our baseline test accuracy numbers are often much lower than what is usually found with standard training of these models. All the models are trained to either reach very high training accuracies (i.e. $99\% - 100\%$) or trained until 500 epochs. Some models did not result in such high accuracies, in which case, results have been shown on the model obtained at epoch 500. We trained 3 instances of each model and results displayed are averaged over these instances with the shaded region indicating the range of results also indicated in the plots.

Once the model is trained, we apply MASC on each layer of the network with respect to different subspaces. For MLP models, all the MASC experiments were performed for all the layers in the network including on the input (after it is pre-processed). For CNN models and AlexNet models, the experiments were performed on flatten layer (Flat) and fully connected layers (FC). While we ran the experiments on the input layer for CNNs, we did not do so for AlexNet.

## 3.2 Terminology

The general terminology used in this work is as follows:

**Model Training Accuracy**: The model accuracy on the training set with corrupted labels.

**Model Testing Accuracy**: The model accuracy on the testing data set with true labels.

**Minimum Angle Subspace Classifier (MASC) Accuracy on Corrupted Training**: Training accuracy of MASC on training data set with corrupted labels was used in determining the subspaces.

**Minimum Angle Subspace Classifier (MASC) Accuracy on Original Training**: Training accuracy of MASC on training data with respect to true training labels.

**Minimum Angle Subspace Classifier (MASC) Accuracy on Testing**: Testing accuracy of MASC on testing data set with true labels was used.

## 4 Enhanced generalization ability in memorized models

Models trained with corrupted labels have high training accuracy (on corrupted labels) while also having low testing accuracy (Zhang et al., 2021). We ask if we can decode the representations of the hidden layers of these memorized models to obtain better generalization.

---

[2]The CNN models were built along the lines of (Tran et al., 2022).

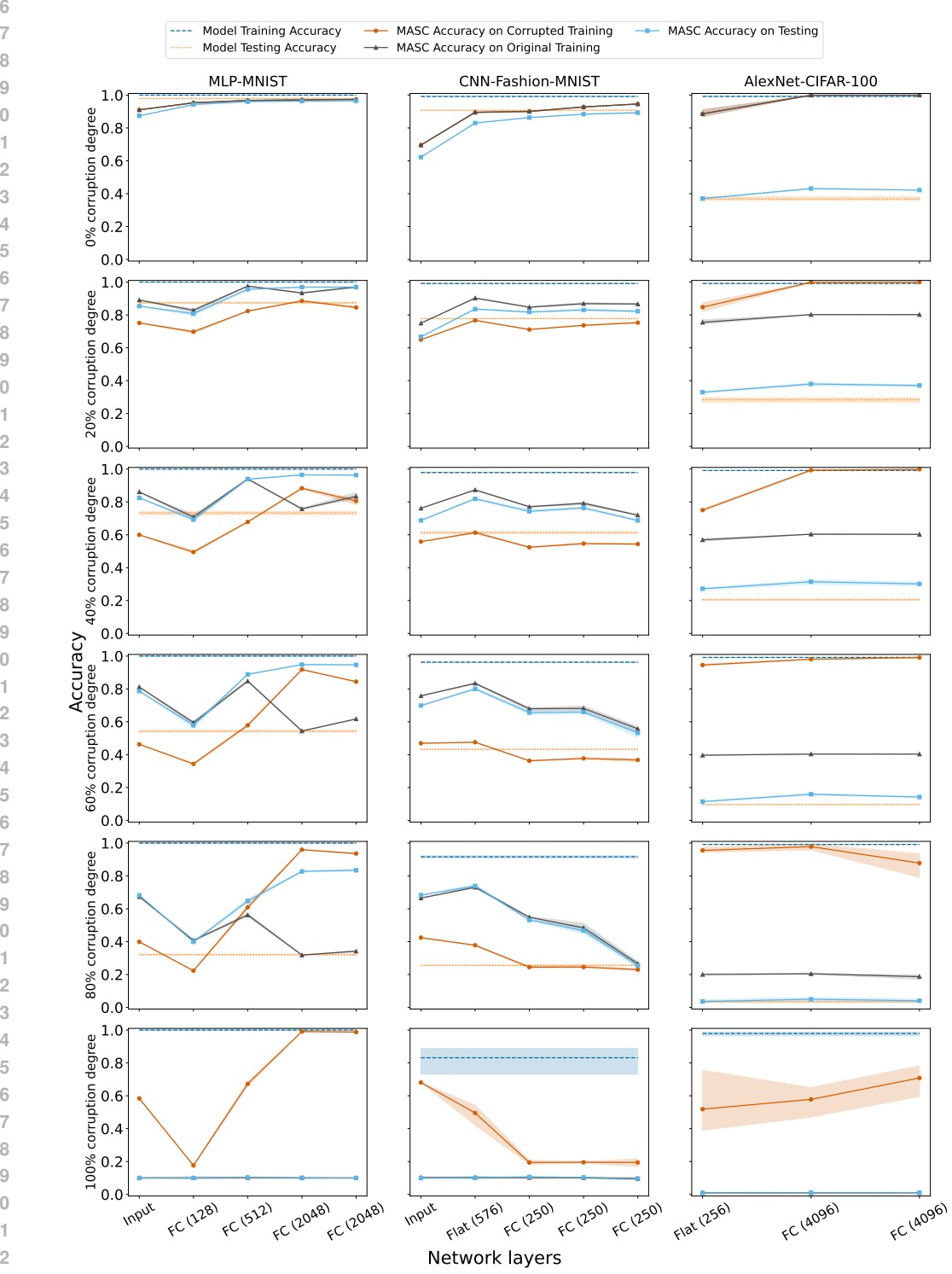

Figure 1: Minimum Angle Subspace Classifier (MASC) accuracy over the layers of the network when the data is projected onto corrupted training subspaces with the indicated corruption degree, for multiple models/datasets. Rows corresponds to plots with the same corruption degree & the columns correspond to the models, as noted. Training accuracy (dashed line) & testing accuracy (dotted line) of the model is shown. FC corresponds to fully connected layer with $ReLU$ activation whereas Flat corresponds to flatten layer without $ReLU$ activation. The number of class-wise PCA components of these models are shown in Figure 16 in section A.9 of the Appendix.

To do so, we build a Minimum Angle Subspace Classifier (MASC) using class-conditioned corrupted training subspaces obtained from the memorized models' hidden layer outputs. MASC is performed layer-wise for all the layers of the network independently as described in Section 3. MASC accuracy on corrupted training data, MASC accuracy on original training data, and MASC accuracy on testing data over the layer of MLP trained on MNIST, CNN trained on Fashion-MNIST, AlexNet trained on CIFAR-100 are shown in Figure 1. $SGD$ optimizer (Qian, 1999) was used for training MLP models, whereas $Adam$ optimizer (Kingma, 2014) was used for other models.

Importantly, for every corrupted model we have (with non-zero corruption degree), except those with 100% corruption degree, we find that our Minimum Angle Subspace Classifier (MASC) in at least one layer has better testing accuracy than the corresponding model itself. In many cases, the MASC testing accuracy is dramatically better than that of the model. This is remarkable, because, in addition to the layerwise outputs, the MASC used precisely the same information (including the same corrupted training dataset) that was available to the model itself, and yet is able to extract better generalization. This suggests that the model retains significant latent generalization ability, which is not captured in its own test-set performance. In most models, the same MASC, especially on the later layers, also approaches perfect accuracy on the corrupted training set, indicating that this improved generalization happens concurrently with memorization of training data points with shuffled labels. Below, we make more specific observations on the performance of the models.

With generalized models i.e. those with 0% corruption degree, at the later layers of the network, it is observed that in most of the cases MASC accuracy on training data approaches the models training accuracy. Similarly, MASC accuracy on testing data is comparable to or performed better than the models' test accuracy.

Even for high corruption degrees, we find that the MASC performs well. For example, with 80% corruption degree, which implies that approximately 72% of the training labels have been changed, we observed good MASC testing accuracy in many cases. Notably, the MASC test accuracy on the later layers is over 80% on MLP-MNIST, in comparison to 34% test accuracy by the model. Similarly, MASC test accuracy on one of the layers is about 75% for CNN-Fashion-MNIST, in contrast to 25% model test accuracy.

Not only does the MASC have better accuracy than the model on the test data but it also does well on the training data with the true labels. Although the model has memorized the training data with corrupted labels, outputs from certain layers have the ability to predict the trained true labels. For example, in MLP-MNIST, for low to moderate degrees of corruption, MASC on the middle layer (FC (512)) has good accuracy on the true training labels, while also retaining good accuracy on the test set. With 40% corruption degree, approximately 36% are changed labels and yet the model has good accuracy on the true training labels in at least one layer of the network. e.g. MLP-MNIST has over 90% true training accuracy at layer FC(512), CNN-Fashion-MNIST has approximately 85% in Flat (576) layer & AlexNet-CIFAR-100 has approximately 60% in FC (4096) layer. This means that almost 20% of those labels are predicted correctly even though the model was trained for 500 epochs or has reached high training accuracy on corrupted labels. In the process of doing this, the model does not have any direct information about the true labels and neither does the MASC.

One way to think about a deep network, is as one that successively transforms input representations in a manner that aids in good prediction performance. Therefore, performance of the MASC on the input is a good baseline measure to assess if subsequent layers have favorable accuracies. Naively, for models trained with corrupted data, one would expect layered representations that enable the model to do well on the corrupted training data, but not do well on the test data or the training data that have true labels. While this expectation seems to hold with respect to the model itself, we find that the layer-wise representations do not necessarily follow this expectation. That is, MASC applied to subsequent layers, often have better true training accuracy and test accuracy than the MASC applied to the input, suggesting that the deep network does indeed transform the data in a manner more amenable to correct prediction, even if its labels are dominated by noise.

MLP model trained on CIFAR-10 with $SGD$ optimizer is shown in Section A.10 along with MLP models trained on CIFAR-10 and MNIST with $Adam$ having qualitatively similar results. We ran some preliminary experiments with Dropout as a regularizer. To do so, we have trained CNN on MNIST, Fashion-MNIST, CIFAR-10 and AlexNet on CIFAR-100 with dropout. The details and results are provided in Section A.11.

## 5 GENERALIZATION VIA TRUE TRAINING LABELS WITH MEMORIZED MODELS

While the previous section demonstrated improved generalization performance by the MASC, we want to investigate if there exist better subspaces that can offer superior generalization performance. To this end, we consider the setting where the true label identities of the training set are known post training with corrupted labels. Can we extract significantly high training as well as testing performance in this case from the layerwise outputs of the network? To do so, we build MASC using subspaces obtained from training data with true labels. It is a priori unclear if MASCs trained in this manner will have high accuracy. Since the network trained assuming different labels for many of the datapoints, it is conceivable that class-wise subspaces corresponding to true labels lack structure and predictive power. We find, however, that these possibilities do not bear out.

MASC accuracy on original training data and on testing data projected on true training label subspace over the layers of the same networks from Section 4 is shown in Figure 2. For comparison, MASC accuracy on corrupted training data and testing data projected on corrupted training subspace is also shown. We find that, in many cases, accuracies on the true training labels, as well as the test set are dramatically better here than with the experiments where subspaces were determined for the corrupted training data. In fact the MASC test accuracies for the corrupted models (with non-zero corruption degree) are sometimes fairly close to the test accuracy of the uncorrupted model.

Strikingly, even for models trained with 100% corruption degree, in most cases, the MASC retains significant accuracy on the true training labels as well as the test set. This is in spite of the fact that the model itself has chance-level test-set accuracy. For example, MASC classifier has 95% test labels accuracy in last FC(2048) layer for MLP-MNIST, 69% test labels accuracy for Flat(576) layer in CNN-Fashion-MNIST, and 4% test labels accuracy for Flat(256) layer in AlexNet-CIFAR-100.

The results here are proof of principle that suggest the existence of subspaces which allow one to extract significantly high generalization performance on models trained with datapoints whose labels are shuffled to a remarkably high degree. This has two implications. On the one hand, it demonstrates that models trained with very high label noise, surprisingly, retain the latent ability to generalize very well. On the other hand, it suggests that development of new techniques to identify favorable subspaces could help markedly boost generalization performance of models, whose training data is known to have label noise.

Similar results were seen with respect to $Adam$ optimizer over the MLP layers trained on MNIST and CIFAR-10, and MLP trained on CIFAR-10 with $SGD$ optimizer as shown in Figure 21 with its respective class-wise PCA components available in Figure 22 in the Appendix. Results for model with regularization are shown in Figure 29 and 31 with its respective class-wise PCA components available in Figure 30 and 32.

The Appendix also describes a control experiment with MASC accuracies on a random initialization of the network (Section A.6, as well as comparison with early stopping test accuracies (Section A.4). We also have results corresponding to more models and datasets (Section A.8) and details of experiments for AlexNet-Tiny ImageNet (Section A.5).

## 6 INDUCING MEMORIZATION IN UNCORRUPTED MODELS

Conversely, we examined if we could build a MASC classifier on a model trained on true training labels, with the goal of memorizing training data whose labels are corrupted to varying degrees.

To do this, we take generalized models, i.e. models trained with uncorrupted training data. We then shuffle the labels of the training set to some corruption degree and construct the corresponding class-specific subspaces with respect to the layerwise outputs of the model. We then build a MASC classifier corresponding to these subspaces.

MASC accuracy on original training data and MASC accuracy on testing data over the layer same networks from Section 4 are shown in Figure 3. Additional results are available in the Appendix in Figures 23, 33, 35 and 39 and their respective class-wise PCA components are available in Figure 24, 34, 36.

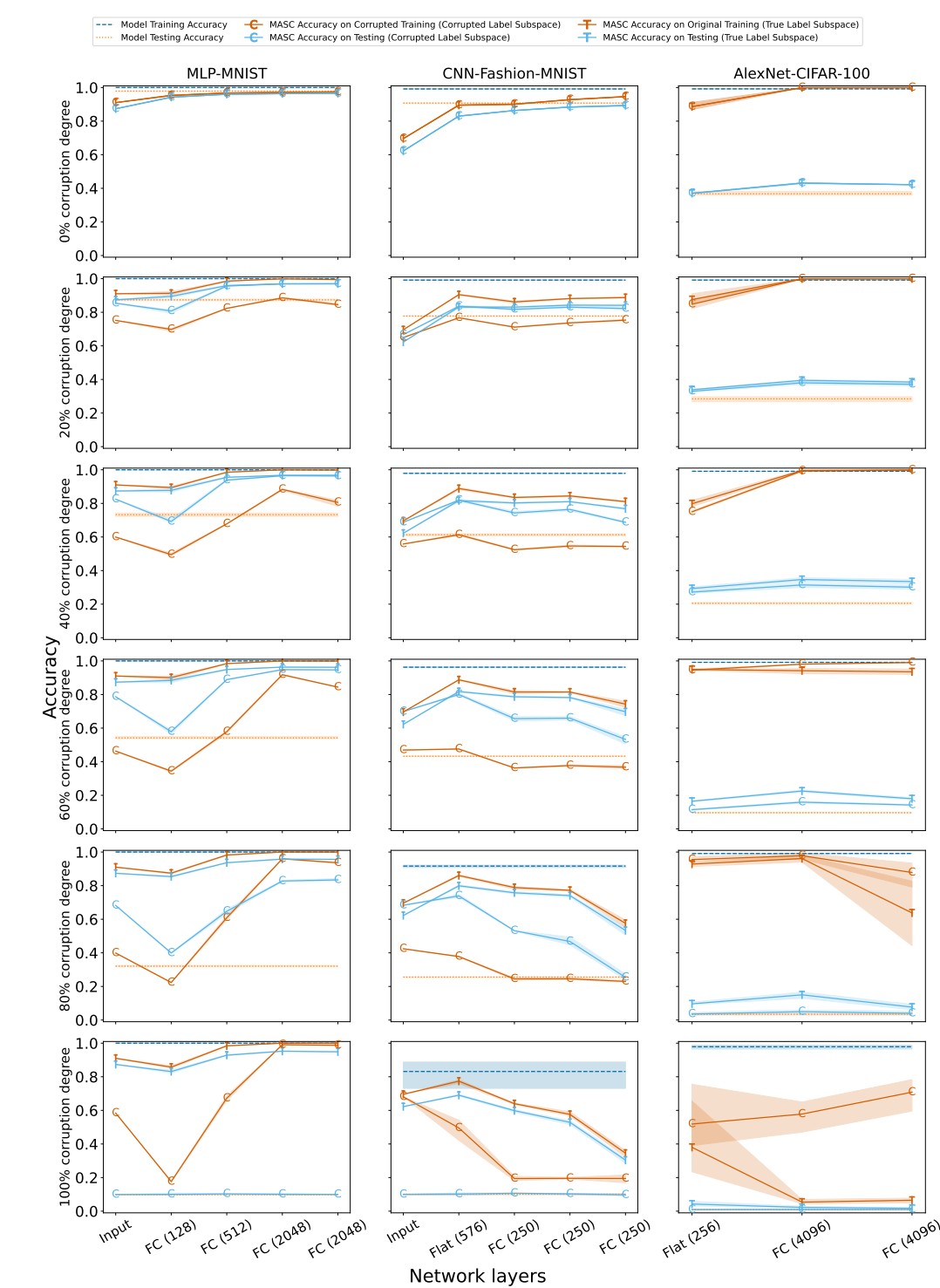

Figure 2: Minimum Angle Subspace Classifier (MASC) accuracy over the layers of the network when the data set is projected onto corrupted subspace and subspace corresponding to true training labels. Rows corresponds to plots which have the same corruption degree and the columns correspond to the models as noted. Training and testing accuracy of the model is shown. FC corresponds to fully connected layer with ReLU activation whereas Flat corresponds to flatten layer without ReLU activation. The respective number of class-wise PCA components for true training label subspaces of the models is shown in Figure 17 in section A.9 of the Appendix.

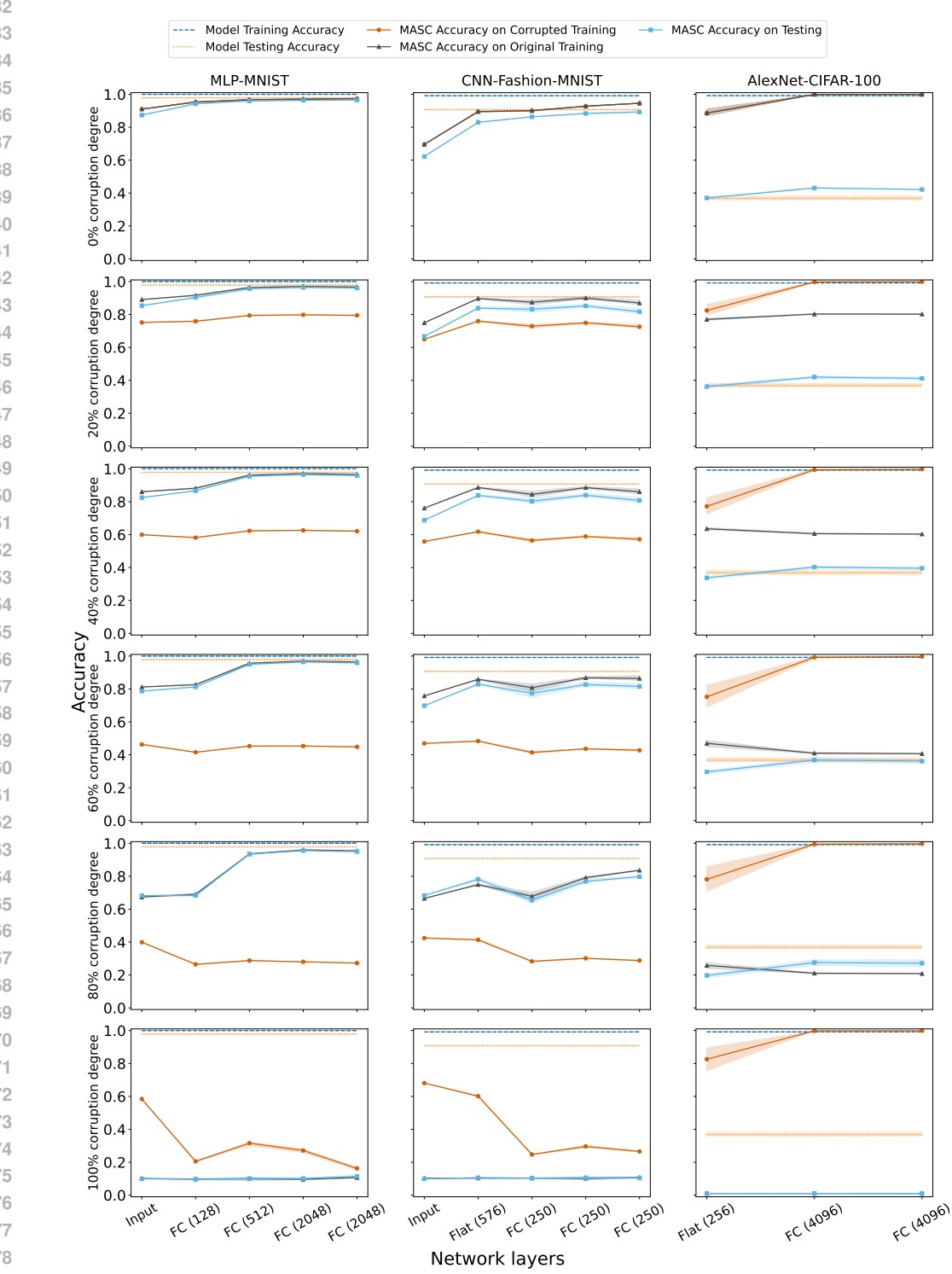

Figure 3: Minimum Angle Subspace Classifier (MASC) accuracy over the layers of the generalized network when the data set is projected onto corrupted training subspaces with the indicated corruption degree. Rows corresponds to plots which have the same corruption degree & the columns correspond to the generalized models as noted. Training & testing accuracy of the generalized model is shown. FC corresponds to fully connected layer with $ReLU$ activation whereas Flat corresponds to flatten layer without $ReLU$ activation. The respective number of class-wise PCA components of the models is shown in Figure 18 in Section A.9 of the Appendix.

Interestingly, we find that for uncorrupted model with modest model test accuracies (i.e. AlexNet-CIFAR-100), the MASC classifiers described above have high accuracies on the corrupted training set. Conversely, in most uncorrupted models with high model test accuracies (i.e. MLP-MNIST and CNN-Fashion-MNIST), we find that these MASC classifiers have more modest accuracies on the training set with corrupted labels. One exception to this, is in Figure 23 of the Appendix, where we have MLP-Adam-MNIST models with high model test accuracy. Yet, we find that a MASC classifier on the first FC(2048) layer trained with training labels corrupted to 100% corruption degree has over 90% accuracy on training data with corrupted labels. Also, MASC classifiers often have test accuracies that approach or exceed uncorrupted model test accuracies, even though they correspond to corrupted subspaces (see e.g. AlexNet on CIFAR-100).

## 7 DISCUSSION

In this work, we investigated the phenomenon of memorized networks not generalizing well, asking why the ability to generalize is apparently lost during the act of memorizing. We find, surprisingly, that the intrinsic ability to generalize remains present to a degree not previously recognized, and this ability can be decoded from the internals of the network by straightforward means.

An interesting question is about why this phenomenon even occurs; naïvely one would expect that networks, on being trained with highly noisy data, discard the ability to generalize in favor of learning noise. Are there specific inductive biases that promote such generalization? And, do such mechanisms also promote generalization in networks whose training data isn't corrupted significantly by such noise? It would also be instructive to study the dynamics of this form of generalization during training[3]. It is known (Arpit et al., 2017) that the model's test accuracy transiently peaks in the early epochs of training with corrupted data, before dropping while training accuracy of the corrupted training data rises. It is unclear whether this transient rise in model generalization is caused by the subspace organization seen here, and if so, why such subspace organization isn't degraded as much as the model's test error over further epochs of training.

The work has a number of implications. On the one-hand, it suggests that the ability to memorize and generalize may not be antithetical. Indeed, in multiple cases, we are able to construct single MASC classifiers that perform well both on the shuffled training labels as well as on the held-out test data that has true labels. Secondly, theories proposed to explain generalization in deep networks have traditionally argued for the setting where the data distribution is well-behaved, i.e. corresponding to real-world data, but not for data with shuffled labels. We suggest, in light of the present results, that such theories also ought to be able to explain why networks retain the ability to generalize even in the face of noisy training data. That is, a satisfactory understanding of generalization in deep networks should also cover the settings where the training data is noisy and its distribution is not well behaved. Thirdly and more pragmatically, techniques such as the MASC classifier might suggest a way of boosting generalization in trained Deep Networks, whose training data intrinsically contains varying degrees of label noise. While this has been beyond the scope of the present paper, possibilities of designing new techniques for learning subspaces that have good generalization ability could be explored. Indeed, it is possible that significantly better subspaces exist than the ones uncovered here, and it would be interesting to see how much the generalization accuracy can be improved by pursuing this direction. Relatedly, it is possible that other classifiers operating on layerwise outputs have better performance than MASC – a possibility that merits further exploration. Fourthly, it would be interesting to formulate a measure to study representational similarity between memorized & generalized networks to see if they use similar mechanisms. Does the answer depend on the particular class of networks (e.g. MLPs vs. CNNs)?

Finally, the results here are reminiscent of a puzzling phenomenon observed in Neuroscience. In multiple settings (Shusterman et al., 2011; Miura et al., 2012; Stringer et al., 2021), in mice, rats and humans, it has been shown that a decoder using data from a subset of neurons from specific areas in the brain of a well-trained behaving animal has accuracy significantly better than the behavioral accuracy of the animal, even though the animal is motivated to do well on the task. It may therefore be that this is a phenomenon shared between brains and machines, whose underlying mechanisms and potential trade-offs remain to be investigated.

---

[3]which has been beyond the scope of this paper.

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

## A  APPENDIX

### A.1  SUBSPACE CREATION

The process of creating subspaces, before using the MASC classifier, is shown schematically in Figure 4 for the MLP model.

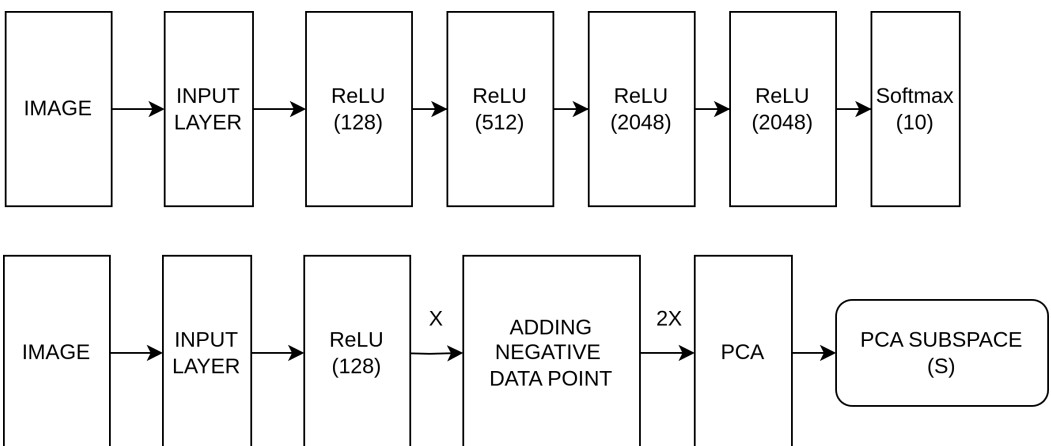

Figure 4: Class-conditioned training data subspaces on layerwise outputs of MLP using PCA. Top: Schematic of MLP model used in the work. Bottom: Creating the class-conditioned training subspace for ReLU (128) layer where 128 are the number of neurons.

### A.2  MODEL DETAILS

The MLP model has 4 hidden layers with 128, 512, 2048 and 2048 units respectively. $ReLU$ activation was used after every layer and for classification $softmax$ activation was applied. We have trained the models with two different optimizers namely, $SGD$ and $Adam$. Learning rate of 1e-3 and momentum = 0.9 was used with $SGD$ optimizer. Learning rate = 1e-4 was used for $Adam$ in experiments. Batch size of 32 was used in all the models. Data set was normalized by dividing each pixel value with 255.

CNN network has 3 blocks, each consisting of two convolutional layers, one max pooling layer. These blocks are followed by three fully connected layers. Convolutional layers have 16, 32, and 64 filters, respectively with stride=1 and filter size = 3 × 3. Max pooling layer has stride of 1 and filter size of 2 × 2. The fully connected layers at the end has 250 units each. It was trained with $Adam$ optimizer with learning rate of 0.0002. For MNIST and Fashion-MNIST batch size of 32 whereas for CIFAR-10 batch size of 128 were used. Data set was normalized by subtracting the mean and diving by the standard deviation for each channel. $ReLU$ activation was used after every layer except pooling and $softmax$ activation for classification.

AlexNet model was slightly modified to for the use of each dataset. $Adam$ optimizer with learning rate of 0.0001 was used. For CIFAR-100, batch size of 128 and for Tiny ImageNet, batch size of 500 was used. All the results with respect to testing on AlexNet trained on Tiny ImageNet are shown with the validation dataset. CIFAR-100 dataset before training was normalized by subtracting the mean and diving by the standard deviation for each channel. No data normalization was performed on Tiny ImageNet dataset.

The experiments were performed on workstations/servers with a variety of GPUs, including Nvidia GeForce RTX3080s, GeForce RTX3090s, Tesla V100s and A100s.

### A.3  TRAINING AND TESTING PERFORMANCE OF THE MODELS

Average training and testing accuracies of the models over three different runs used in this paper are shown in Tables 2 and 3.

Table 1: Training set size of the data sets and the number of parameters of the models.

| Model | Dataset | Training set size | Number of parameters |
|---|---|---|---|
| MLP | MNIST | 60,000 | 5,433,994 |
| | CIFAR-10 | 50,000 | 5,726,858 |
| CNN | MNIST | 60,000 | 344,042 |
| | Fashion MNIST | 60,000 | 344,042 |
| | CIFAR-10 | 50,000 | 456,330 |
| AlexNet | CIFAR-100 | 50,000 | 38,738,952 |
| | Tiny ImageNet | 100,000 | 39,776,464 |

Table 2: Average training accuracy in percentages of all the models over three runs over different corruption degrees (indicated in the last six columns). WD and WOD corresponds to with dropout and without dropout respectively.

| Model | Dataset | Parameter | 0% | 20% | 40% | 60% | 80% | 100% |
|---|---|---|---|---|---|---|---|---|
| **MLP** | **MNIST** | **SGD** | 99.99 | 99.99 | 99.99 | 99.99 | 100 | 100 |
| | | **Adam** | 100 | 99.87 | 99.73 | 99.77 | 99.73 | 99.66 |
| | **CIFAR-10** | **SGD** | 99.99 | 99.99 | 99.99 | 99.99 | 99.99 | 99.99 |
| | | **Adam** | 99.63 | 99.53 | 99.43 | 99.61 | 99.52 | 30.21 |
| **CNN** | **MNIST** | **WOD** | 99.90 | 99.32 | 98.62 | 97.25 | 95.11 | 94.92 |
| | | **WD** | 99.90 | 95.73 | 87.28 | 77.70 | 71.18 | 57.24 |
| | **Fashion-MNIST** | **WOD** | 99.15 | 99.14 | 97.90 | 96.25 | 91.65 | 83.14 |
| | | **WD** | 99.13 | 94.89 | 85.33 | 76.45 | 67.55 | 58.81 |
| | **CIFAR-10** | **WOD** | 99.70 | 99.29 | 99.26 | 99.03 | 99.02 | 39.69 |
| | | **WD** | 99.30 | 96.71 | 94.87 | 93.59 | 90.00 | 86.97 |
| **AlexNet** | **CIFAR-100** | **WOD** | 99.19 | 99.15 | 99.11 | 99.16 | 99.14 | 97.88 |
| | | **WD** | 99.37 | 99.07 | 98.66 | 98.20 | 97.16 | 94.39 |
| | **Tiny ImageNet** | **WOD** | 99.92 | 99.90 | 99.91 | 99.93 | 87.71 | 85.95 |

### A.4    COMPARISON OF DECODING RESULTS AND EARLY STOPPING TEST ACCURACIES

Early stopping test model accuracy was added as a reference in this section to compare the results of testing accuracy of MASC on trained model with early stopping model accuracy. MASC accuracy over the layers of the network when the data is projected onto corrupted training subspaces is shown in Figure 5 and onto true training subspaces is shown in Figure 6. Best model testing accuracy and trained model testing accuracy are shown for reference. Best model testing accuracy corresponds to the accuracy of the testing data of the model if early stopping was used.

For MASC when the data is projected onto corrupted training subspace, in AlexNet-CIFAR-100, the MASC in at least one layer shows better performance than the best model testing accuracy for less than 60% corruption degree. For MLP-MNIST, the best model (early stopping) maintains over 90% accuracy even when the data is corrupted up to 80%. Despite the increase in corruptions (except 100%corruption), the accuracy of the last layer remains close to that of the best model accuracy. For CNN-FMNIST (except 100%corruption),in at least one layer MASC performance is near to that of best model testing accuracy.

### A.5    EXPERIMENTS WITH ALEXNET MODEL TRAINED ON TINY IMAGENET

MASC test accuracy over the layers of AlexNet trained on Tiny ImageNet when the data is projected onto corrupted training subspaces with the indicated corruption degree is shown in Figure 7. Testing accuracy of the model and best model testing accuracy is shown for comparison. Best Model Testing Accuracy corresponds accuracy of the testing data of the model if early stopping was used.

Even for AlexNet-Tiny ImageNet corrupted model (with non-zero corruption degree), except those with 100% corruption degree, we find that our Minimum Angle Subspace Classifier (MASC) in

Table 3: Average testing accuracy in percentages of all the models over three runs over different corruption degrees (indicated in the last six columns). WD and WOD corresponds to with dropout and without dropout respectively.

| Model | Dataset | Parameter | 0% | 20% | 40% | 60% | 80% | 100% |
|-------|---------|-----------|------|------|------|------|------|------|
| MLP | MNIST | SGD | 97.87 | 87.38 | 73.28 | 54.16 | 32.09 | 9.81 |
| | | Adam | 98.31 | 90.49 | 76.78 | 55.59 | 31.85 | 9.77 |
| | CIFAR-10 | SGD | 56.37 | 48.62 | 40.35 | 30.55 | 19.68 | 9.80 |
| | | Adam | 52.24 | 44.11 | 35.55 | 25.24 | 15.18 | 10.07 |
| CNN | MNIST | WOD | 99.15 | 87.51 | 69.44 | 47.10 | 28.30 | 9.85 |
| | | WD | 99.13 | 86.19 | 55.99 | 37.37 | 25.85 | 9.95 |
| | Fashion-MNIST | WOD | 90.74 | 77.74 | 61.35 | 43.26 | 25.57 | 10.08 |
| | | WD | 91.36 | 74.56 | 53.20 | 34.92 | 24.32 | 9.68 |
| | CIFAR-10 | WOD | 74.95 | 60.48 | 46.15 | 30.96 | 18.32 | 9.89 |
| | | WD | 73.77 | 58.20 | 43.62 | 29.83 | 17.50 | 10.13 |
| AlexNet | CIFAR-100 | WOD | 36.75 | 28.44 | 20.53 | 9.64 | 3.43 | 0.96 |
| | | WD | 37.59 | 26.21 | 15.70 | 7.49 | 2.51 | 1.05 |
| | Tiny ImageNet | WOD | 15.88 | 9.74 | 5.44 | 2.02 | 0.73 | 0.43 |

Table 4: Subspace constructed using corrupted labels - Percentage the MASC classifier outperformed the model (AlexNet-Tiny ImageNet).

| Corruption degree | 20% | 40% | 60% | 80% |
|-------------------|------|------|------|------|
| Model Test Accuracy (%) | 9.74 | 5.44 | 2.02 | 0.73 |
| MASC Accuacy on Testing (Best layer) | 12.42 | 8.36 | 2.93 | 0.83 |
| MASC outperformed the model (%) | 27.51 | 53.67 | 45.04 | 13.69 |

at least one layer has better testing accuracy than the corresponding model itself. In Table 4, the MASC accuracy on testing for the best layer as well as by what percentage the MASC classifier outperformed the model for the best layer for each corruption degrees 20%, 40%, 60% and 80% is documented.

We have also performed MASC with 90% variance explained for PCA on AlexNet trained on Tiny ImageNet. The comparison between 99% and 90% variance captured for PCA is shown in Section A.12.

MASC test accuracy over the layers of AlexNet trained on Tiny ImageNet when the data set is projected onto corrupted training and true training subspace is shown in Figure 8. In Table 5, the MASC accuracy on testing for the best layer as well as by what percentage the MASC classifier outperformed the model for the best layer for each corruption degrees 20%, 40%, 60% and 80% is documented.

We have also performed an addition experiments with 99.9% PCA variance explained on AlexNet trained on Tiny ImageNet. We find that with high percentage of PCA variance explained, the MASC performs better on training dataset with true labels. The results for MASC of 99.9% , 99% and 90% variance explained are shown in Figure 38.

## A.6 RANDOMLY INITIALIZED CONTROL MODELS VS TRAINED MODELS

This section covers a set of control experiments to show MASC performance on random initialized model and contrast this with the trained models presented in the main text. We have verified that, for every such control model, the model training and testing accuracies for the randomly initialized models is at chance level, for the corresponding dataset in question.

MASC accuracy on testing for randomly initialized model and trained model when data is projected on corrupted training subspaces is shown in Figure 9 and 11. Trained model training and testing accuracies are shown for reference.

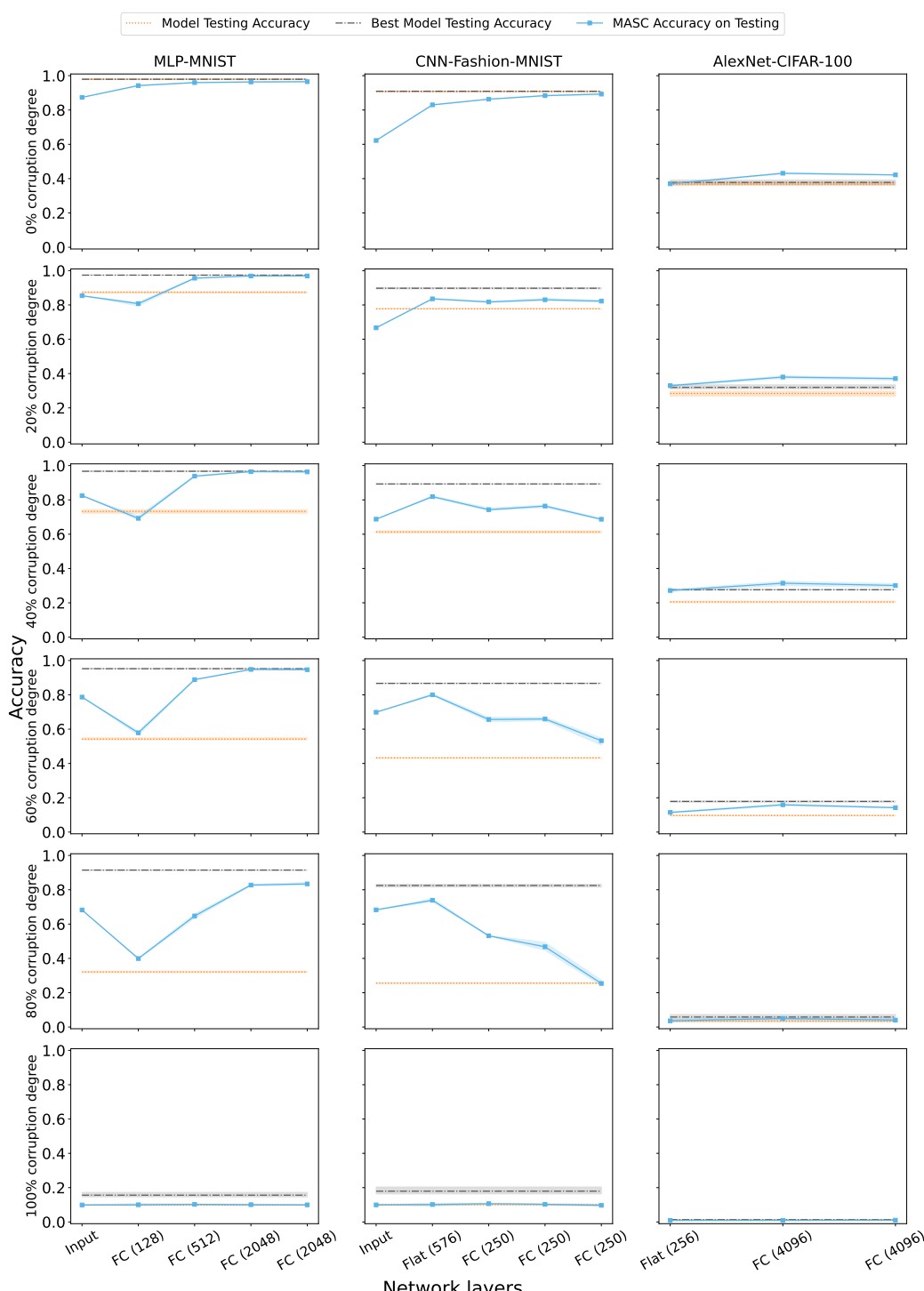

Figure 5: MASC test accuracy over the layers of the network when the data is projected onto corrupted training subspaces with the indicated corruption degree. Best Model Testing Accuracy corresponds accuracy of the testing data of the model if early stopping was used.

We find that indeed accuracies of the MASC classifier on the random initialization outperforms the network, except for low corruption degrees (i.e. $<= 20\%$ corruption degree). However, in the experiments where subspaces are trained on corrupted training data from corrupted models, by-and-large, the MASC classifier usually, and on at least one layer outperforms the MASC classifier

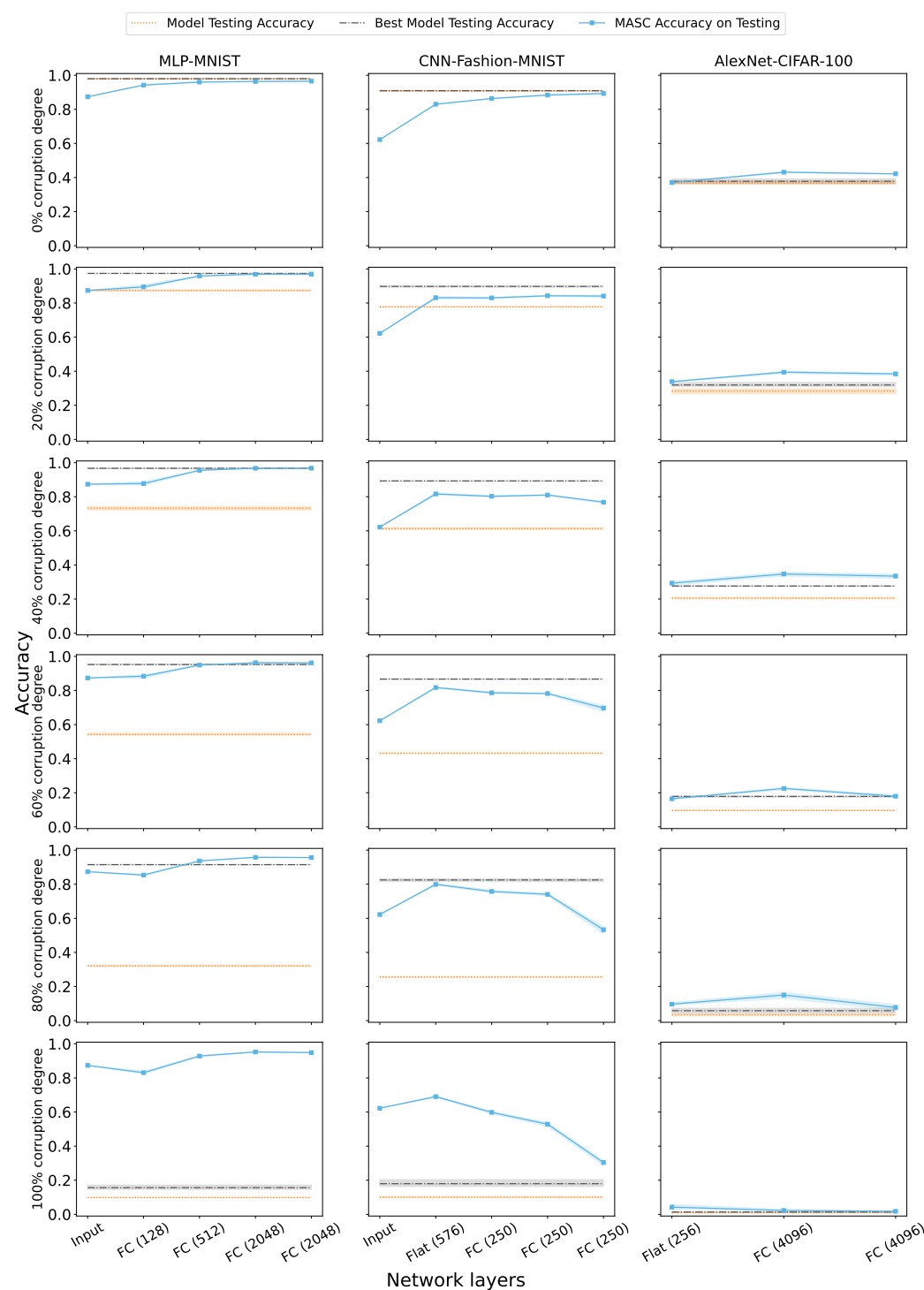

Figure 6: MASC test accuracy over the layers of the network when the data set is projected onto subspace corresponding to true training labels. Best Model Testing Accuracy corresponds accuracy of the testing data of the model if early stopping was used.

trained on the random initialization with exceptions being the 80% corruption degree models on MLP-MNIST, AlexNet-Tiny ImageNet and 100% corruption degree on CNN-FashionMNIST.

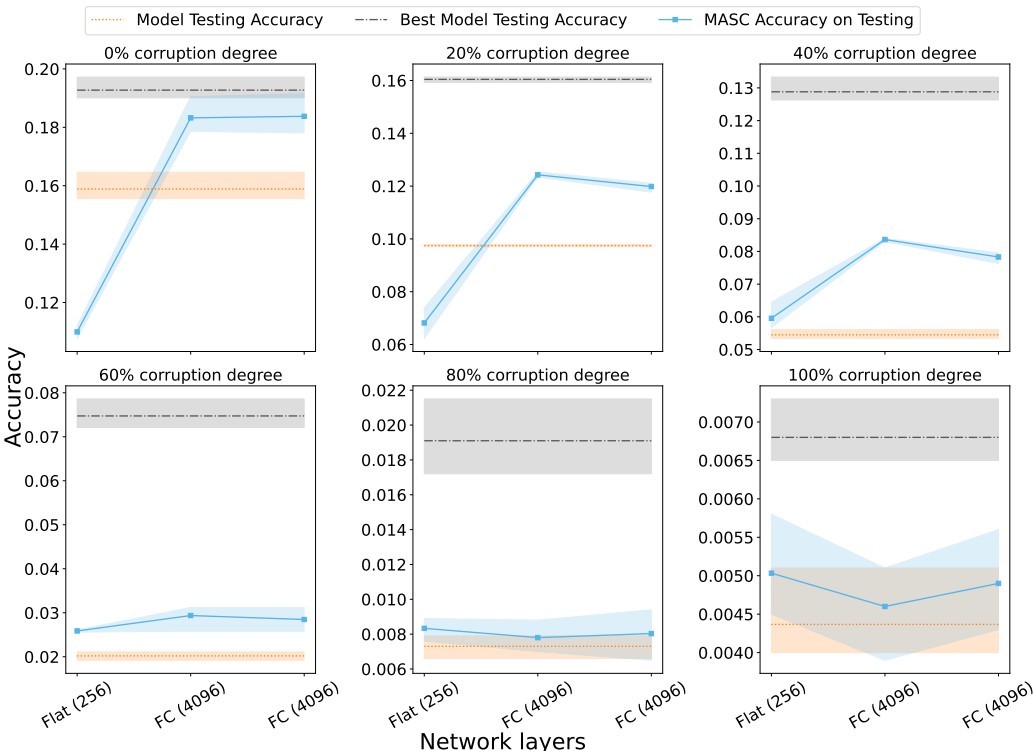

Figure 7: MASC test accuracy over the layers of AlexNet trained on Tiny ImageNet when the data is projected onto corrupted training subspaces with the indicated corruption degree. Testing accuracy of the model and best model testing accuracy is shown for comparison. Best Model Testing Accuracy corresponds accuracy of the testing data of the model if early stopping was used.

Table 5: Subspace constructed using true labels - Percentage the MASC classifier outperformed the model (AlexNet-Tiny ImageNet).

| Corruption degree | 20% | 40% | 60% | 80% |
|---|---|---|---|---|
| Model Test Accuracy (%) | 9.74 | 5.44 | 2.02 | 0.73 |
| MASC Accuacy on Testing (Best layer) | 12.99 | 10.02 | 5.19 | 2.27 |
| MASC outperformed the model (%) | 33.36 | 84.19 | 156.93 | 210.95 |

MASC accuracy on testing for random initialized model and trained model when data is projected on subspaces corresponding to true training labels is shown in Figure 10 and 12. Notably, for the experiments where subspaces are constructed with true labels on corrupted models, the MASC classifier on these models outperforms the MASC classifier on random initializations usually and certainly in at least one layer on every model tested. These results are consistent with the main message of the paper, namely that even with memorized models, the layerwise representations of the models are organized in a manner that they develop significant ability to generalize over and above that bestowed by a random initialization, and in particular, they do not lose this ability, as one might have naively expected, due to label noise. If they were losing this ability, then the MASC classifier on the subspaces would end up performing significantly worse than the MASC classifier run on randomly initialized models.

Although it is interesting that random projection have good generalization capabilities, it is not surprising as this has been shown by (Alain & Bengio, 2018) and studied by others (Jarrett et al., 2009).

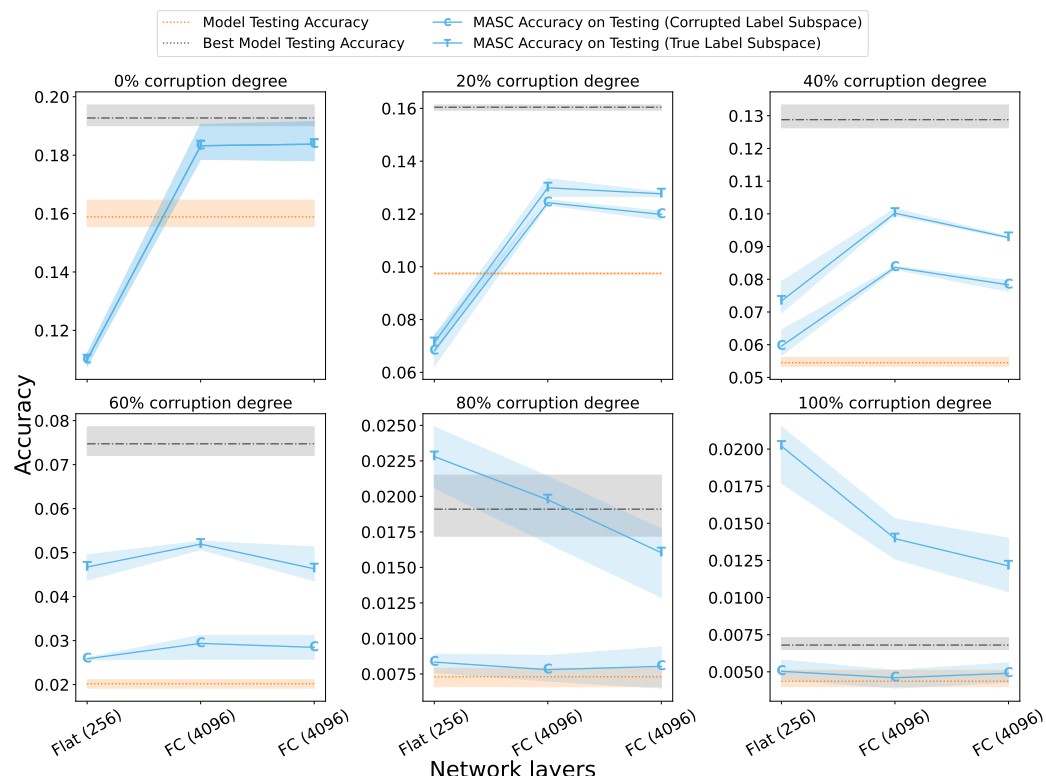

Figure 8: MASC test accuracy over the layers of AlexNet trained on Tiny ImageNet when the data set is projected onto corrupted training and true training subspace. Testing accuracy of the model and best model testing accuracy is shown for comparison. Best Model Testing Accuracy corresponds accuracy of the testing data of the model if early stopping was used.

## A.7 MINIMUM ANGLE SUBSPACE CLASSIFIER ALGORITHM

For a given data point $x$ from training or testing set, given a layer output data point $x_l$ from layer $l$ when input $x$ is passed through the network and its corresponding training subspaces $\{S_k\}_{k=1}^K$, we use Minimum Angle Subspace Classifier (MASC) Algorithm 1 for predicting class labels $y(x_l)$

For training dataset $\mathcal{D}\{(x_i, y_i)\}_{i=1}^m \in \mathbb{R}^d \times \mathbb{R}$, where each $x_i \in \mathbb{R}^d$ and $y_i \in \{C_k\}_{k=1}^K$ are input-label pairs, we estimate training subspaces $\{S_k\}_{k=1}^K$ for all classes $K$ and given layer $l$ of the neural network using Algorithm 2 and 3. For experiments in Section 4, Algorithm 2 uses corrupted training labels whereas in Section 5, Algorithm 2 uses true training labels to create class-specific subspaces.

**Minimum Angle Subspace Classifier (MASC) Accuracy on Corrupted Training**: After using Algorithm 1, the accuracy with respect to MASC predicted class labels of training data set and corrupted labels of training data are calculated.

**Minimum Angle Subspace Classifier (MASC) Accuracy on Original Training**: After using Algorithm 1, the accuracy with respect to MASC predicted class labels of training data set and true labels of training data is calculated.

**Minimum Angle Subspace Classifier (MASC) Accuracy on Testing**: After using Algorithm 1, the accuracy with respect to MASC predicted class labels of testing data set and original labels of testing data is calculated.

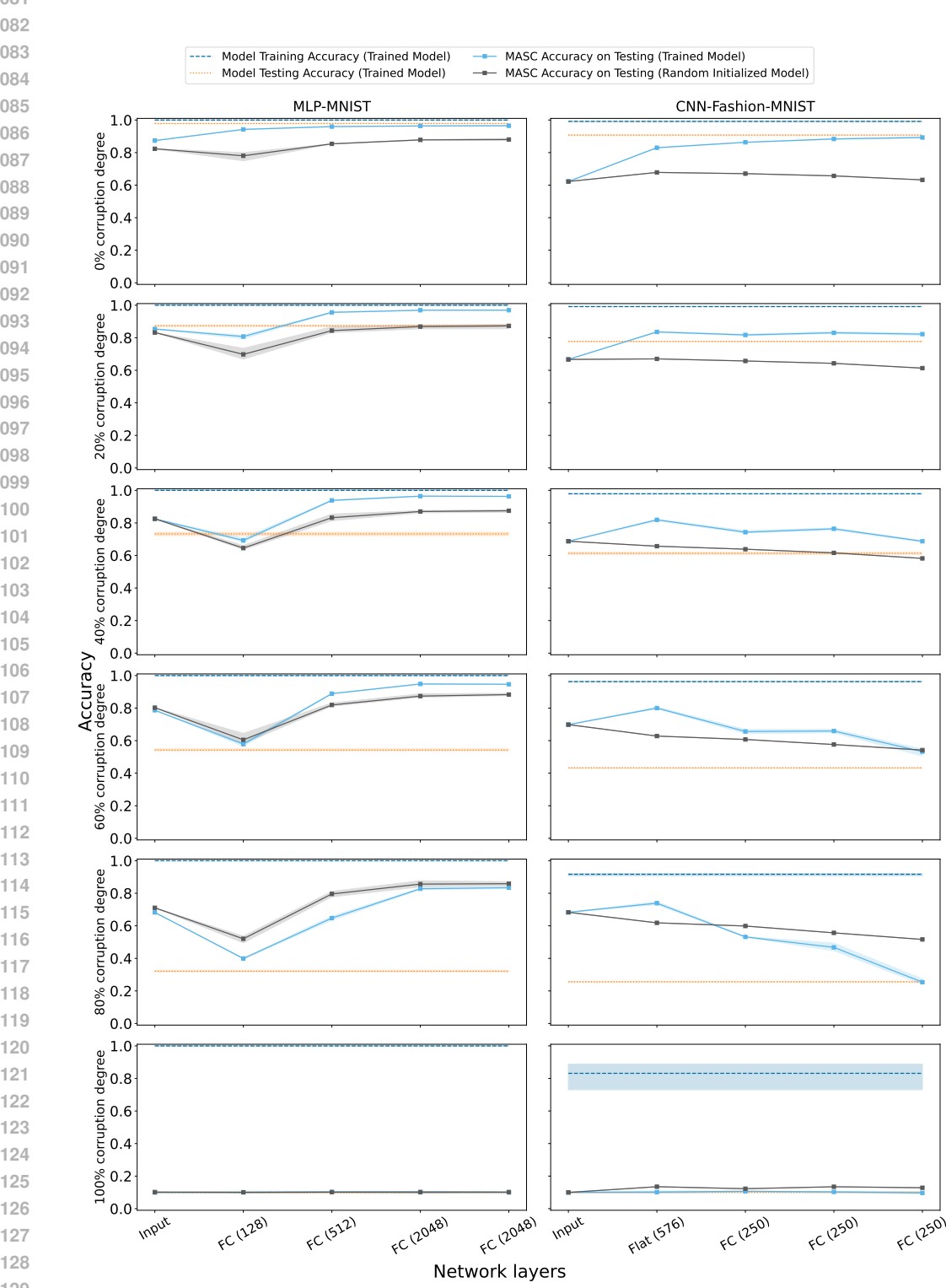

Figure 9: MASC accuracy over the layers of trained and random initialized network when the data is projected onto corrupted training subspaces with the indicated corruption degree.

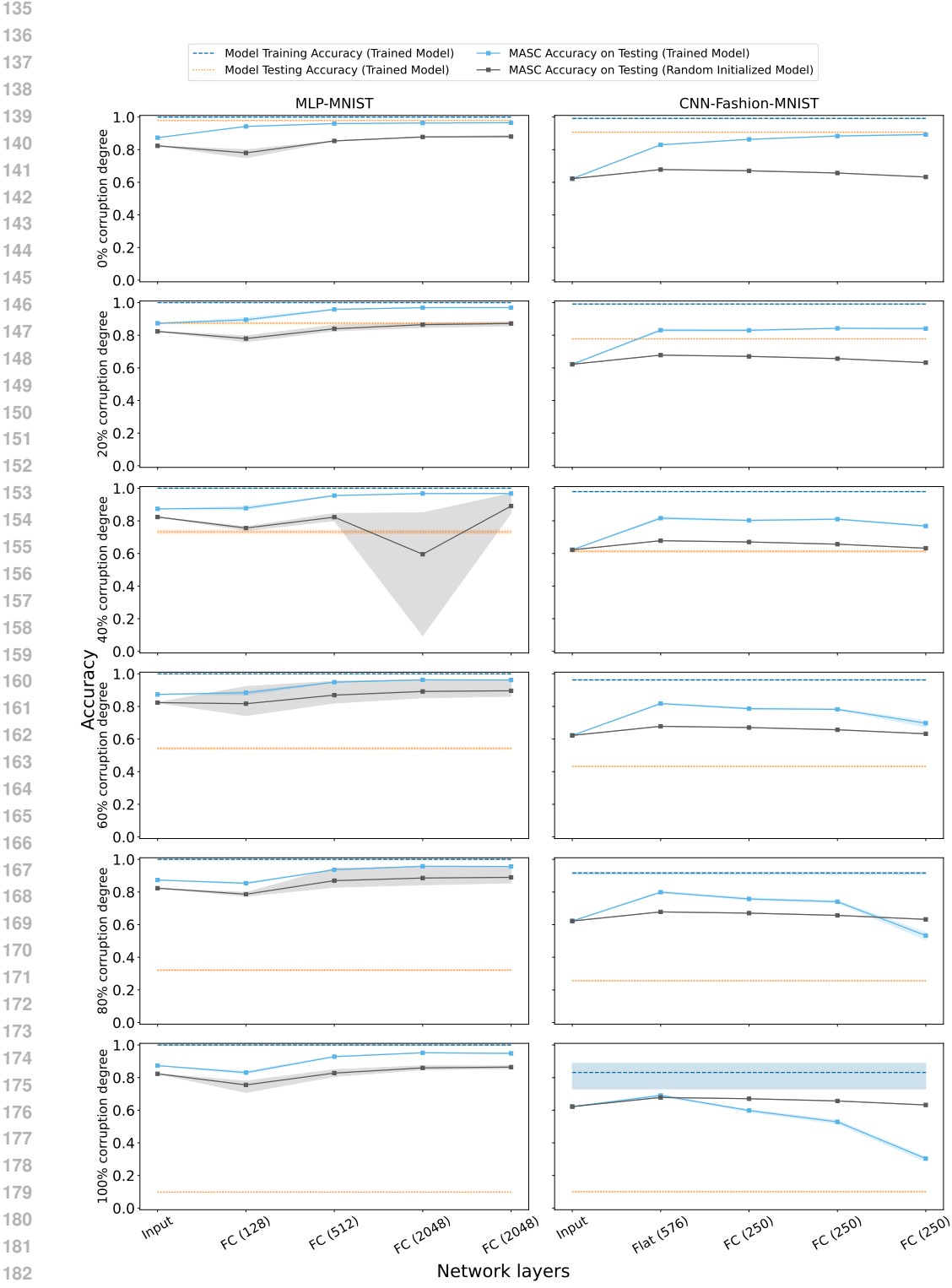

Figure 10: MASC accuracy over the layers of trained and random initialized network when the data set is projected onto subspace corresponding to true training labels. Testing accuracy of the trained model is shown for comparison.

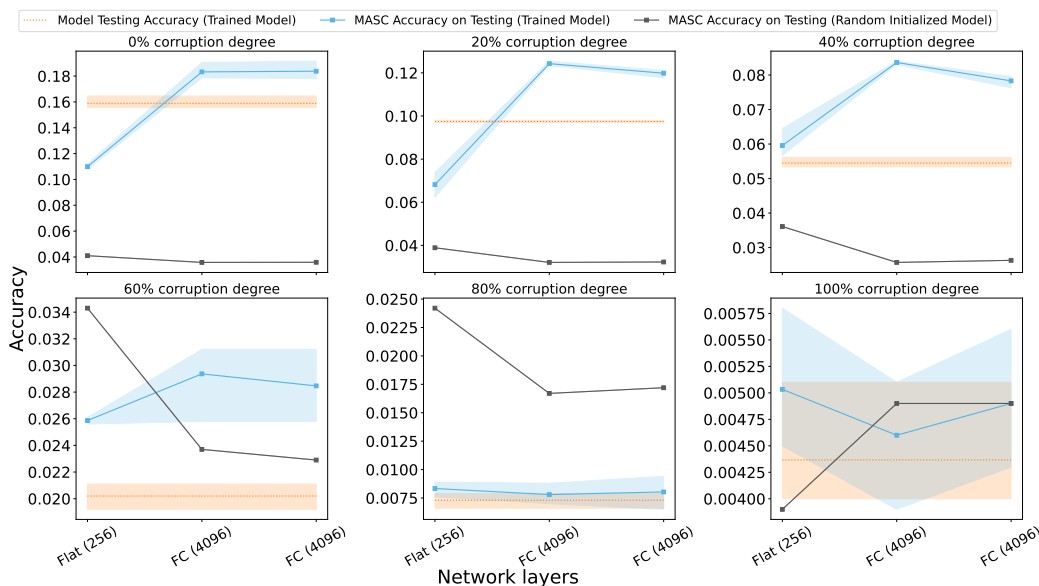

Figure 11: MASC accuracy over the layers of trained and random initialized AlexNet-Tiny ImageNet when the data is projected onto corrupted training subspaces with the indicated corruption degree.

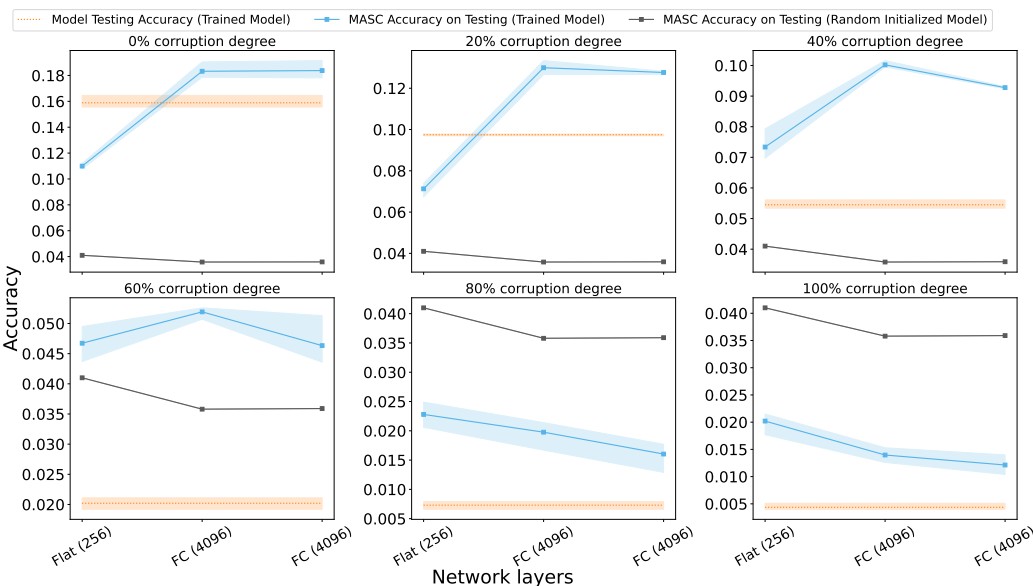

Figure 12: MASC accuracy over the layers of trained and random initialized AlexNet-Tiny ImageNet when the data set is projected onto subspace corresponding to true training labels.

---

**Algorithm 1 Minimum Angle Subspace Classifier (MASC)**

---

1: **Input:** Training subspaces $\{S_k\}_{k=1}^K$, layer output data point $\boldsymbol{x_l}$ from layer $l$ when input $\boldsymbol{x}$ is passed through the network and classes $\{C_k\}_{k=1}^K$.
2: **Output:** MASC prediction class label $y(\boldsymbol{x_l})$ according to layer $l$ .
3: **for** each class $C_k$ **do**
4:    $\boldsymbol{x_{lk}} \longleftarrow$ compute the projection of $\boldsymbol{x_l}$ onto subspace $S_k$.
5:    Compute the angle $\theta(\boldsymbol{x_l}, \boldsymbol{x_{lk}})$ between $\boldsymbol{x_l}$ and $\boldsymbol{x_{lk}}$
6: **end for**
7: Assign the label $y(\boldsymbol{x_l}) = C_k$ where $k = \arg\min_k \theta(\boldsymbol{x_l}, \boldsymbol{x_{lk}})$

---

---

**Algorithm 2 Subspaces Estimator for MASC**

---

1: **Input:** Training dataset $\mathcal{D}\{(\boldsymbol{x_i}, y_i)\}_{i=1}^m \in \mathbb{R}^d \times \mathbb{R}$, where each $\boldsymbol{x_i} \in \mathbb{R}^d$ and $y_i \in \{C_k\}_{k=1}^K$ are input-label pairs, neural network, and layer $l$.
2: **Output:** Subspaces $\{S_k\}_{k=1}^K$ for classes $K$ and given layer $l$.
3: $\mathcal{D}_l = \phi$
4: **for** each input pair $(\boldsymbol{x_i}, y_i)$ in $\mathcal{D}$ **do**
    Pass $\boldsymbol{x_i}$ through the network layers to obtain the output of layer $l$, denoted as $\boldsymbol{x_l} \in \mathbb{R}^{ld}$.
    $\mathcal{D}_l = \mathcal{D}_l \cup \{\boldsymbol{x_l}\}$
5: **end for**
6: Estimated subspaces $\{S_k\}_{k=1}^K \longleftarrow$ **PCA-Based Subspace Estimation**$(\mathcal{D}_l)$
7: **Return:** Subspaces $\{S_k\}_{k=1}^K$

---

**Algorithm 3 PCA-Based Subspace Estimation**

---

1: **Input:** Layer output $\mathcal{D}_l = \{(\boldsymbol{x_l}, y_i)\}_{i=1}^m$, where $\boldsymbol{x_l} \in \mathbb{R}^{ld}$ and $y_i \in \{C_k\}_{k=1}^K$
2: **Output:** Subspaces $\{S_k\}_{k=1}^K$ for classes $K$
3: $\mathcal{D}_{\text{new}} \leftarrow \mathcal{D}_l$
4: **for** each data point $\boldsymbol{x_l}$ in $\mathcal{D}_l$ **do**
    $\mathcal{D}_{\text{new}} \leftarrow \mathcal{D}_{\text{new}} \cup \{-\boldsymbol{x_l}\}$
5: **end for**
6: **for** each class $C_k$ in $C_K$ **do**
7:     Extract the subset of data $\mathcal{D}_{\text{new},k} = \{\boldsymbol{x_l} \mid y_i = k\}$
8:     Apply PCA to $\mathcal{D}_{\text{new},k}$ to calculate the PCA components
9:     The span of the PCA components defines the subspace $S_k$
10: **end for**
11: **Return:** Subspaces $\{S_k\}_{k=1}^K$

---

## A.8 Experimental results on two additional models (MLP-CIFAR10 and AlexNet-Tiny ImageNet.)

All the experimental results on two additional models i.e, MLP-CIFAR10 and AlexNet-Tiny ImageNet are shown in this section.

MASC accuracy over the layers of the MLP trained on CIFAR10 and AlexNet trained on Tiny ImageNet when the data is projected onto corrupted training subspaces is shown in Figure 13. MASC accuracy over the layers of the MLP trained on CIFAR10 and AlexNet trained on Tiny ImageNet when the data set is projected subspace corresponding to true training labels is shown in Figure 14. MASC accuracy over the layers of the generalized MLP network trained on CIFAR10 and AlexNet network trained on Tiny ImageNet when the data is projected onto corrupted training subspaces is shown in Figure 15.

## A.9 Number of PCA components

This section covers the number of class-wise PCA components used in all the experiments in the main paper.

Number of class-wise PCA components of corrupted training subspace over the layer of MLP trained on MNIST and CIFAR-10, CNN trained on Fashion-MNIST, and AlexNet trained on CIFAR-100 and Tiny ImageNet is shown in Figure 16. For generalized models, it is observed that for 99% variance captured, the number of PCA components is significantly smaller in comparison to the ambient dimensionality of the layer (number of units in that layer). Over corruption, it is observed that for MLP-MNIST, MLP-CIFAR-10, and CNN-Fashion-MNIST, the number of class-wise PCA components increase. And the variance between the number of dimensions decrease. For AlexNet-CIFAR-100 and AlexNet-Tiny ImageNet, it is the opposite case, wherein the number of PCA components over corruption decreases.

Number of class-wise PCA components of original training subspaces over the layer of networks used in Section 5 is shown in Figure 17. We find that for original training subspaces, although the

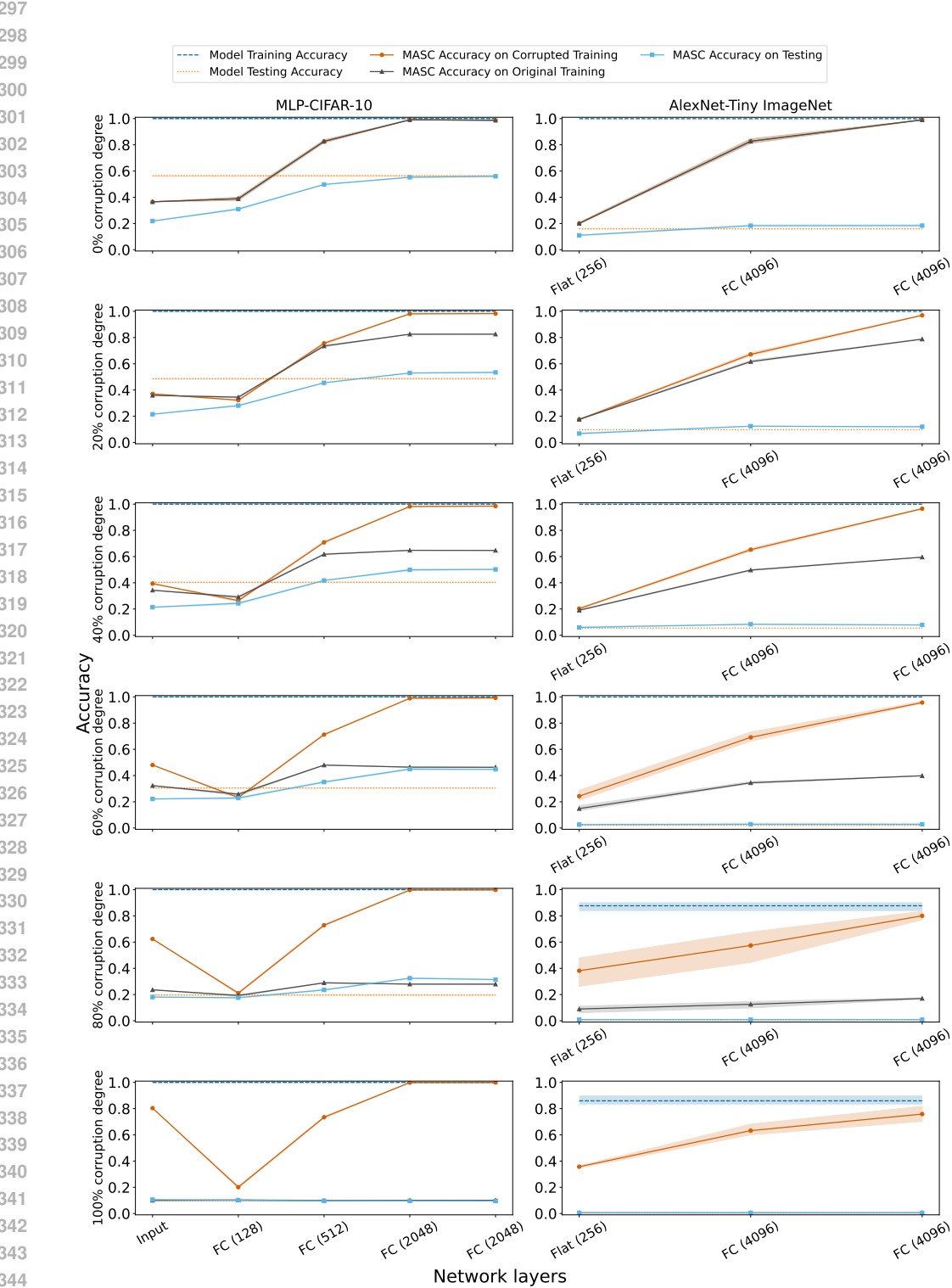

Figure 13: MASC accuracy over the layers of the network when the data is projected onto corrupted training subspaces with the indicated corruption degree. The number of class-wise PCA components of these models are shown in Figure 16 in section A.9 of the Appendix.

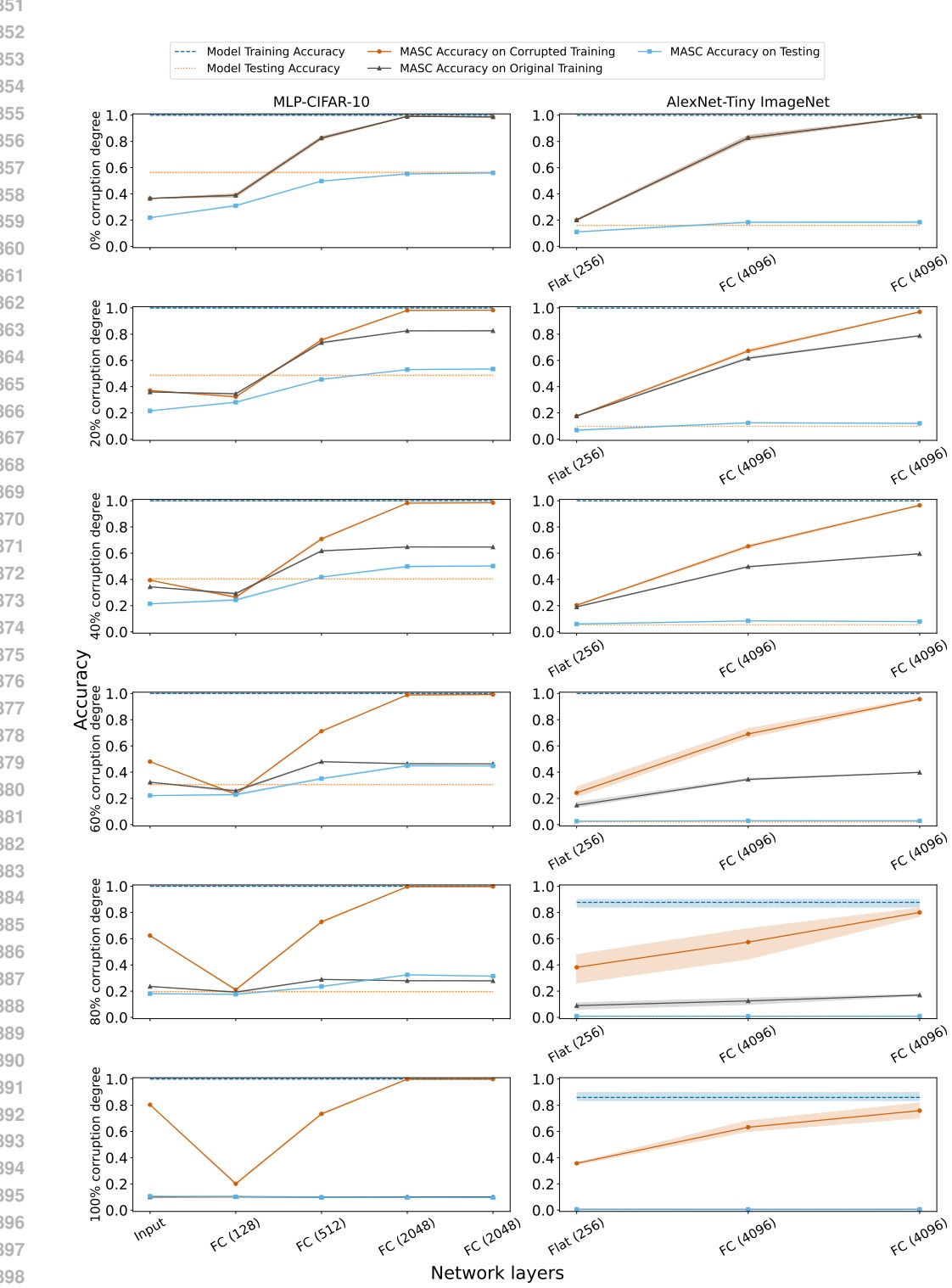

Figure 14: MASC accuracy over the layers of the network when the data set is projected subspace corresponding to true training labels. The respective number of class-wise PCA components for true training label subspaces of the models is shown in Figure 17 in section A.9 of the Appendix.

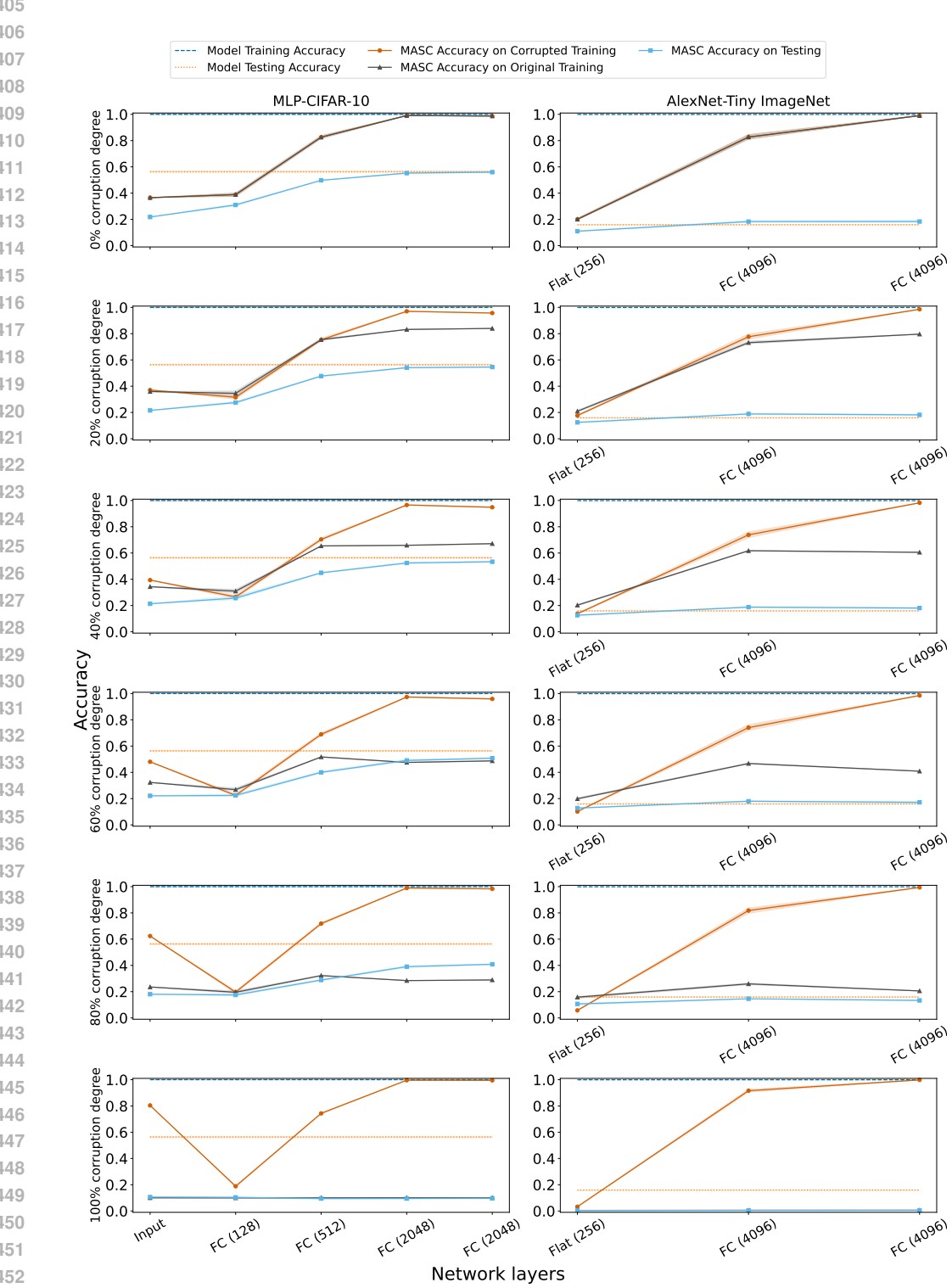

Figure 15: MASC accuracy over the layers of the generalized network when the data set is projected onto corrupted training subspaces with the indicated corruption degree. The respective number of class-wise PCA components of the models is shown in Figure 18 in Section A.9 of the Appendix.

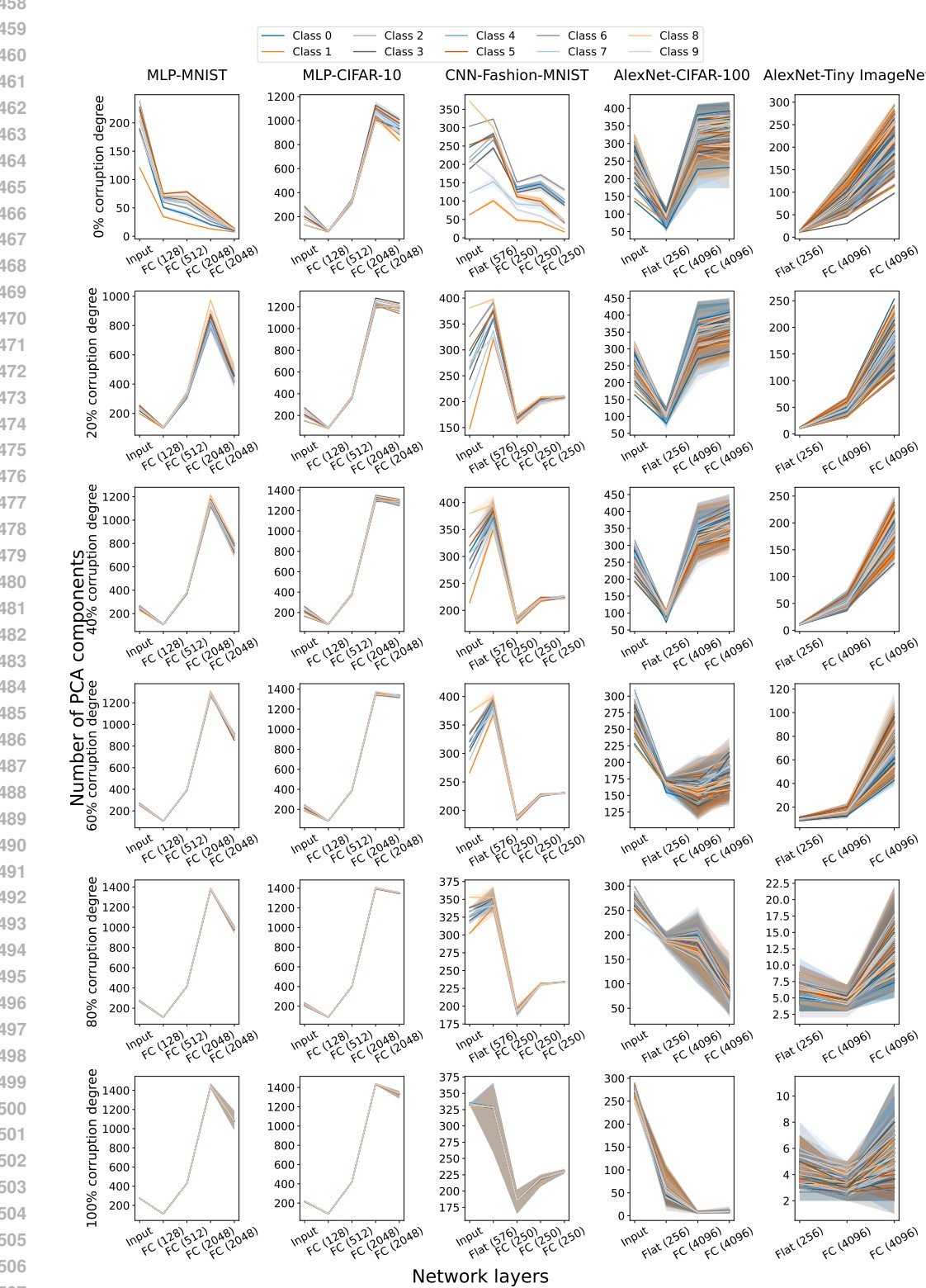

Figure 16: Class-wise number of PCA components of the corrupted training subspace used in section 4 over the layers of multiple networks with various corruptions degrees. Although it is not mentioned in the legend, all the 100 classes and 200 classes of CIFAR-100 and Tiny ImageNet respectively are plotted.

dimensionality has increased with corruption degree, the variance has remained the approximately similar.

Number of class-wise PCA components of original training subspaces over the layer of networks of the generalized model used in Section 6 is shown in Figure 18.

## A.10 SGD vs Adam

We have also trained MLP networks with $Adam$ optimizer on MNIST and CIFAR-10 with various degrees of corruption. The results for MASC accuracy using corrupted subspaces is shown in Figure 19 and its respective average number of PCA components is shown in Figure 20 . With both optimizer choices, even with high corruption degrees, we find that the MASC have better accuracy than the model on the test data. MASC on corrupted training accuracy in most cases reaches the models training accuracy at the latter layers of the network with an exception of MLP trained on MNIST. In most cases, at the initial FC (128) layer of the network, there is a drop in accuracy observed in comparison to the corresponding value for the input and then an increase in latter layers of the network; the MLP trained on CIFAR-10 with SGD is an exception, however.

For 40% corruption degree, although approximately 36% of labels are flipped, with MLP trained on MNIST, model trained on $Adam$ has MASC testing accuracy of around 95%, and around 50% for CIFAR-10. This is better than the MASC accuracy at the input layer and models testing accuracy. For 60% corruption degree with MLP trained on MNIST, model trained on $Adam$ has MASC testing accuracy of around 90% whereas for model trained on $SGD$ is around 85%. Although the networks are trained with 56% of label corruption, yet in FC (512) layer, the MASC training original accuracy is about 70% for model trained on $Adam$ whereas it is 85% for model trained on $SGD$. MASC on original training data does unfavorably on model trained with $SGD$ rather than with $Adam$, although further investigation is required.

The results for MASC accuracy using original subspaces are shown in Figure 21 and its respective average number of PCA components are shown in Figure 22. MLP models trained with $Adam$ have qualitatively similar results. The results for MASC accuracy using corrupted subspaces of generalized models are shown in Figure 23 and its respective average number of PCA components are shown in Figure 24. MLP models trained with Adam have qualitatively similar results.

## A.11 Regularization experiments

We ran some preliminary experiments with Dropout as a regularizer, which we describe below. The CNN model and AlexNet model were slightly modified to add regularization in the form of dropout layers. A dropout layer with dropout probability of 0.2 was used after every fully connected layer in the CNN. This model was trained on MNIST, FashionMNIST, and CIFAR-10. For the AlexNet model, likewise, dropout layers with Dropout probability of 0.5 was added after every fully connected layer in the network and the model was trained on CIFAR-100.

The results for CNN trained on MNIST, Fashion-MNIST with and without Dropout are shown in Figure 25, 29, 33 and their respective number of PCA components are shown in Figure 26, 30, 34.

The results for CNN trained on CIFAR-10 and AlexNet trained on CIFAR100, with and without drop out, are shown in Figure 27, 31, 35 and their respective number of PCA components are shown in Figure 28, 32, 36.

The results appeared somewhat inconclusive. In particular, the models we trained with Dropout did not have significantly better test accuracy than the corresponding models trained without Dropout. In some cases, the models with Dropout did not converge to high training accuracy in spite of training for 500 epochs. We conclude that this calls for more detailed study with more careful hyperparameter tuning.

## A.12 Tiny ImageNet : Accuracy on 0.9 and 0.99 variance captured.

MASC accuracies for the three different experiments for AlexNet trained on Tiny ImageNet with 99% variance captured and 90% variance captured is shown in Figure 37, 38, and 39. All the plots show that MASC accuracy on corrupted training, MASC accuracy on original training and MASC

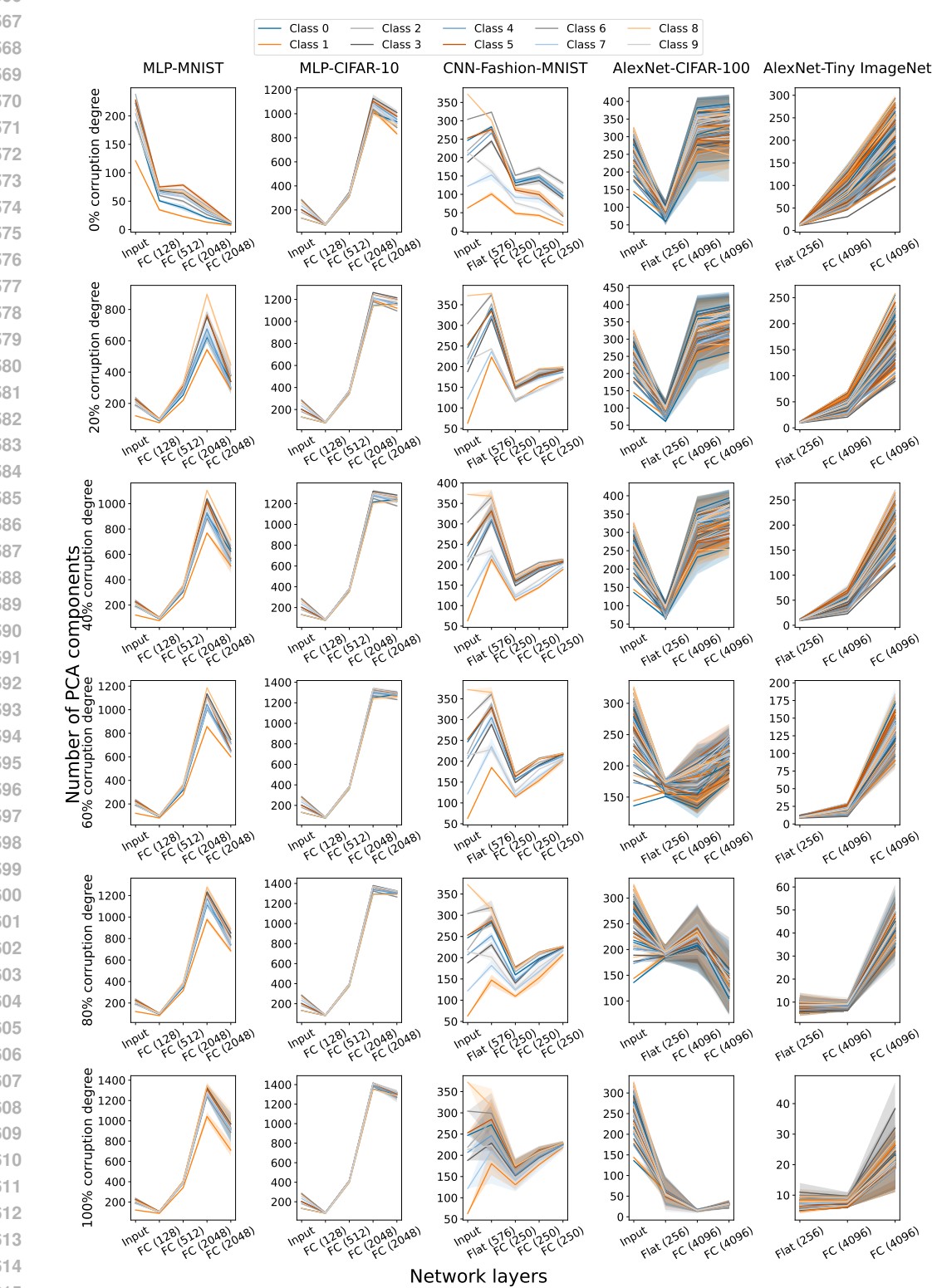

Figure 17: Class-wise number of PCA components of the subspace corresponding to true training labels used in section 5 over the layers of multiple networks with various corruptions. Although it is not mentioned in the legend, all the 100 classes and 200 classes of CIFAR-100 and Tiny ImageNet respectively are plotted.

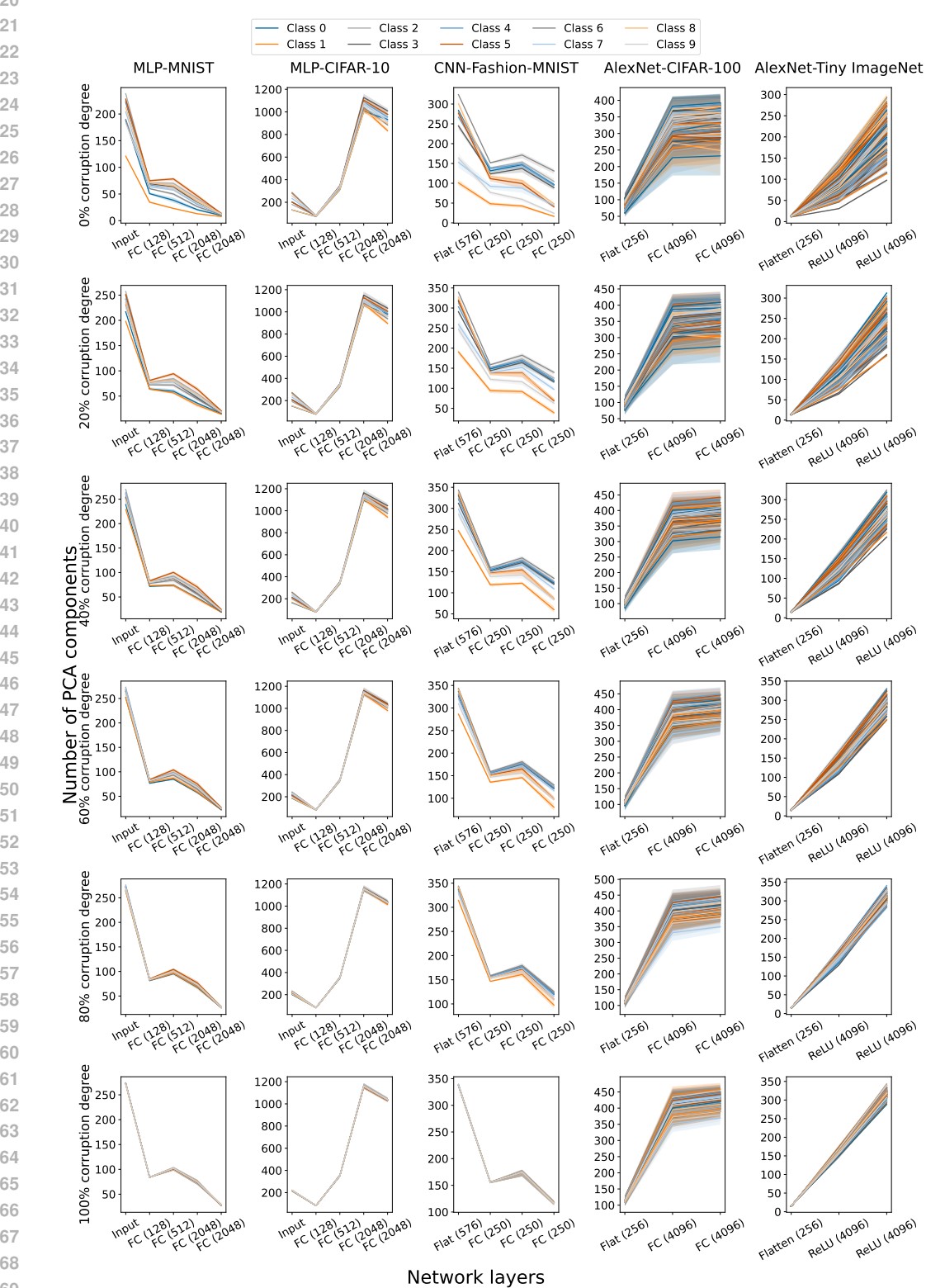

Figure 18: Class-wise number of PCA components of the corrupted training subspace used in section 6 over the layers of multiple generalized networks with various corruption degrees. Although it is not mentioned in the legend, all the 100 classes and 200 classes of CIFAR-100 and Tiny ImageNet respectively are plotted.

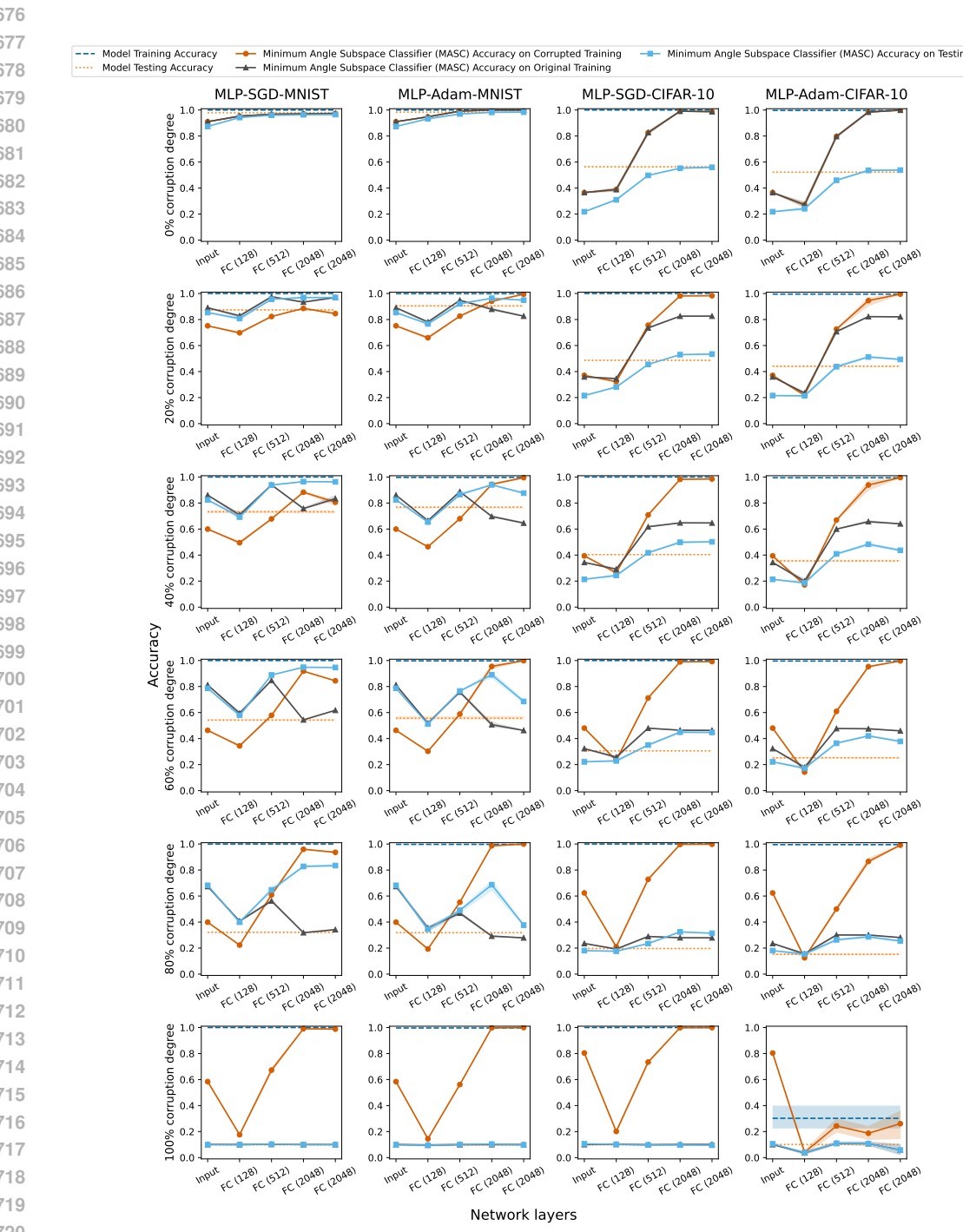

Figure 19: MASC accuracy over the layers of the MLP network when the data is projected onto corrupted training subspaces with the indicated corruption degree, for MLP models with MNIST and CIFAR10 datasets. Rows corresponds to plots which have the same corruption degree and the columns correspond to the models with $SGD$ and $Adam$ optimizer as noted. Training and testing accuracy of the model is shown. FC corresponds to fully connected layer with ReLU activation.

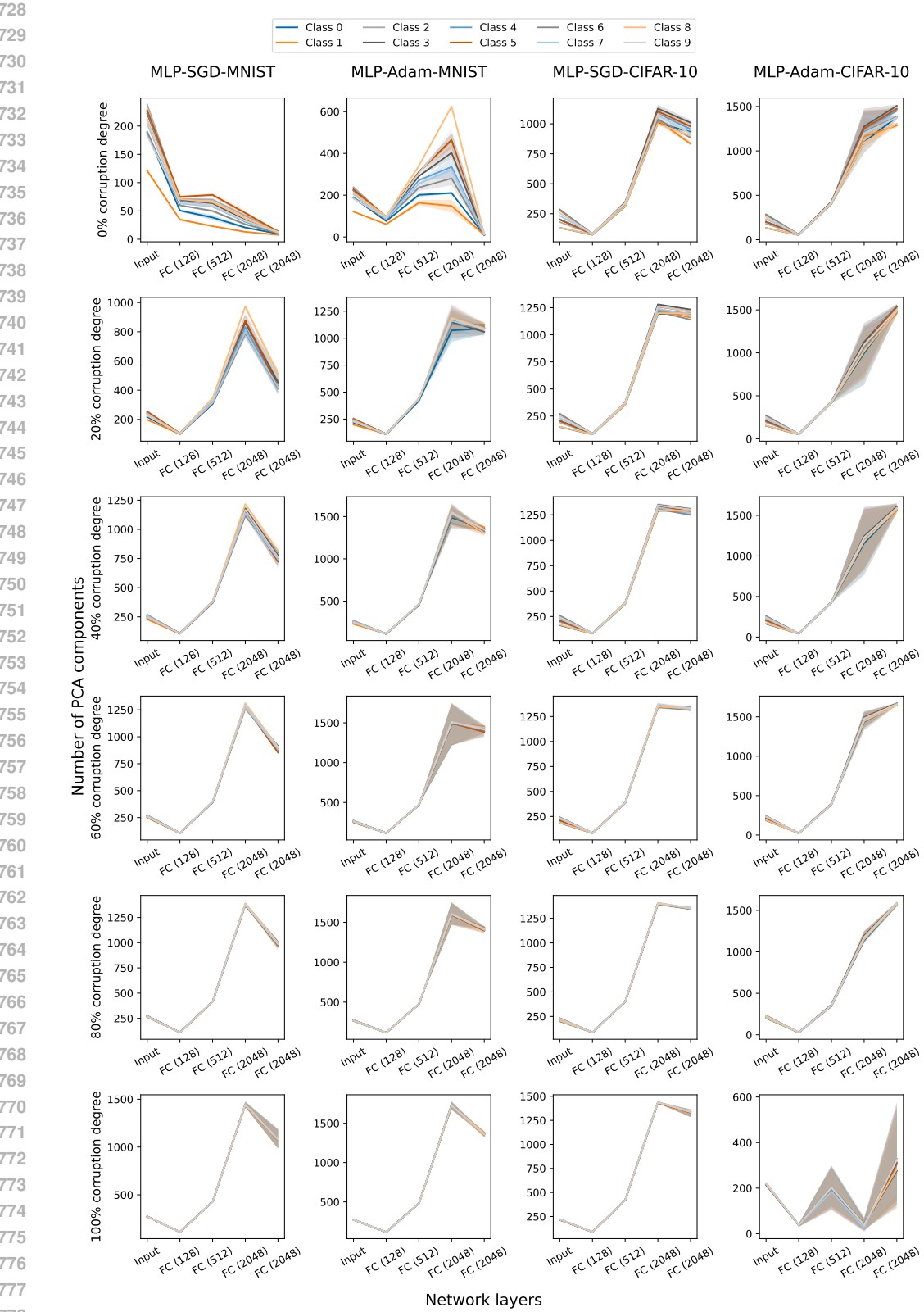

Figure 20: Class-wise number of PCA components of the corrupted training subspace over the layers of MLP networks trained with MNIST and CIFAR10 datasets with various corruption degree. Rows corresponds to plots which have the same corruption degree and the columns correspond to the models with $SGD$ and $Adam$ optimizer as noted.

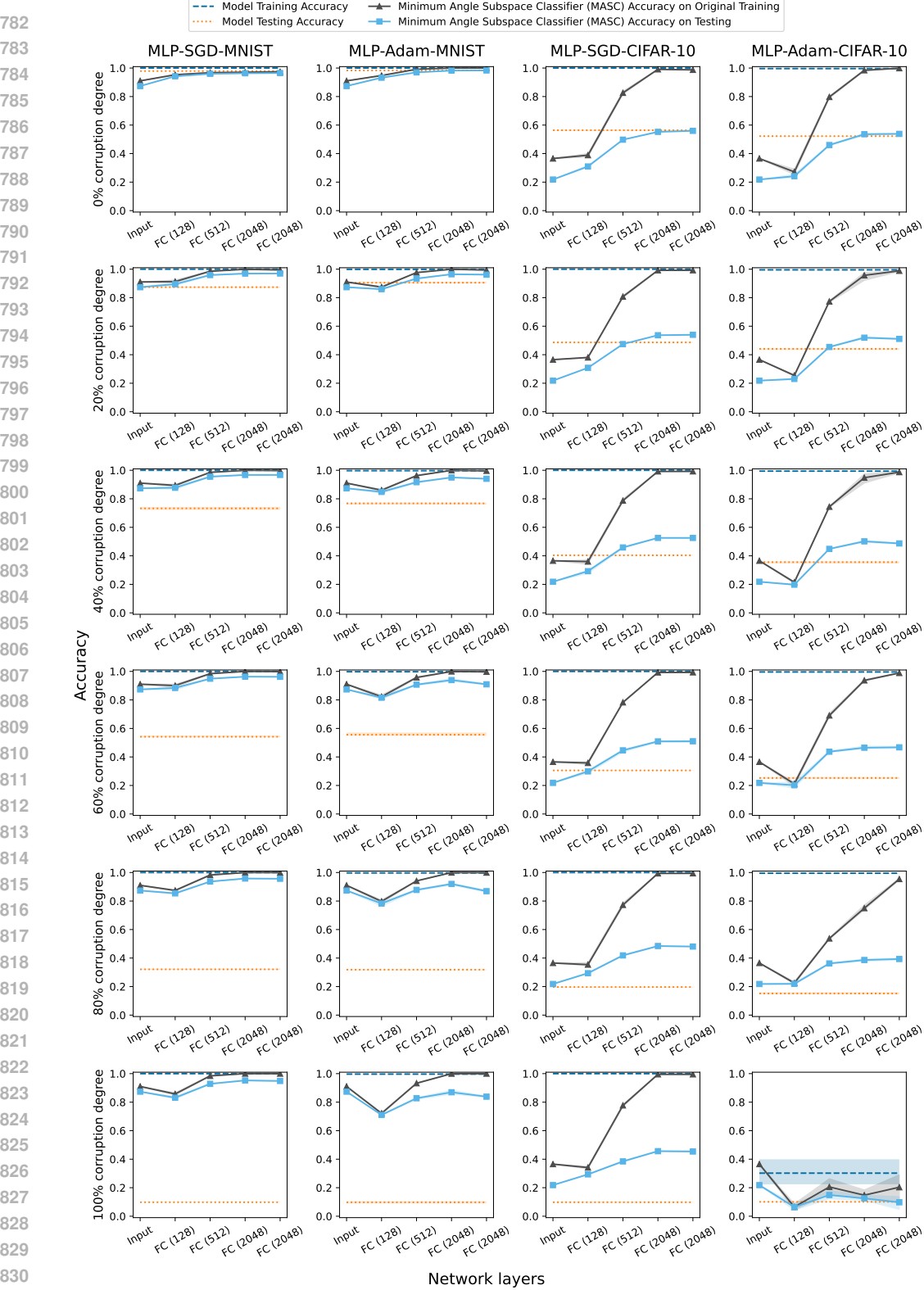

Figure 21: MASC accuracy over the layers of the MLP network when the data set is projected onto subspace corresponding to true training labels. Rows corresponds to plots which have the same corruption degree and the columns correspond to the models with $SGD$ and $Adam$ optimizer as noted. Training and testing accuracy of the model is shown. FC corresponds to fully connected layer with $ReLU$ activation.

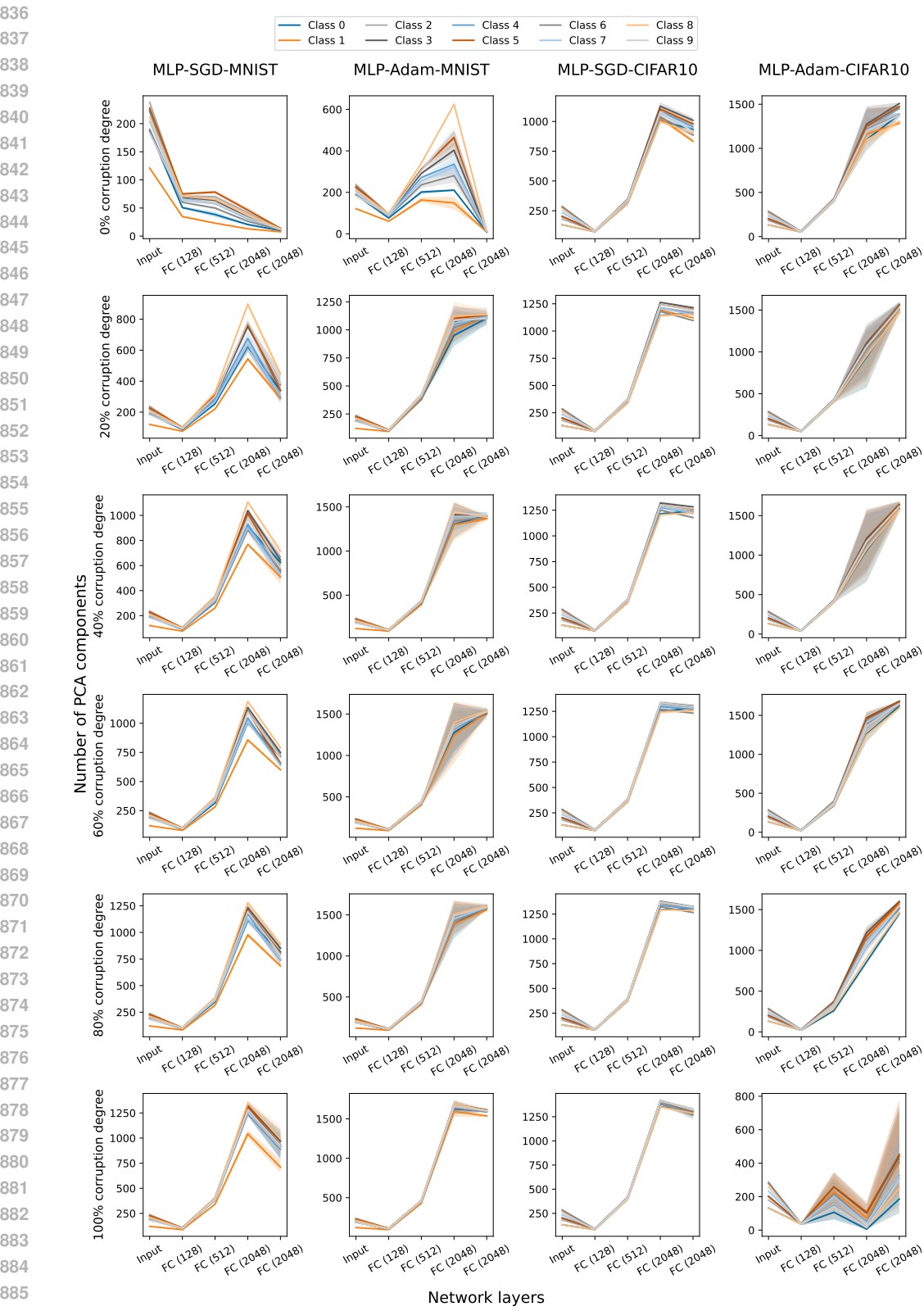

Figure 22: Class-wise number of PCA components of the subspace corresponding to true training labels over the layers of MLP networks with various corruption degrees. Rows corresponds to plots which have the same corruption degree and the columns correspond to the models with $SGD$ and $Adam$ optimizer as noted.

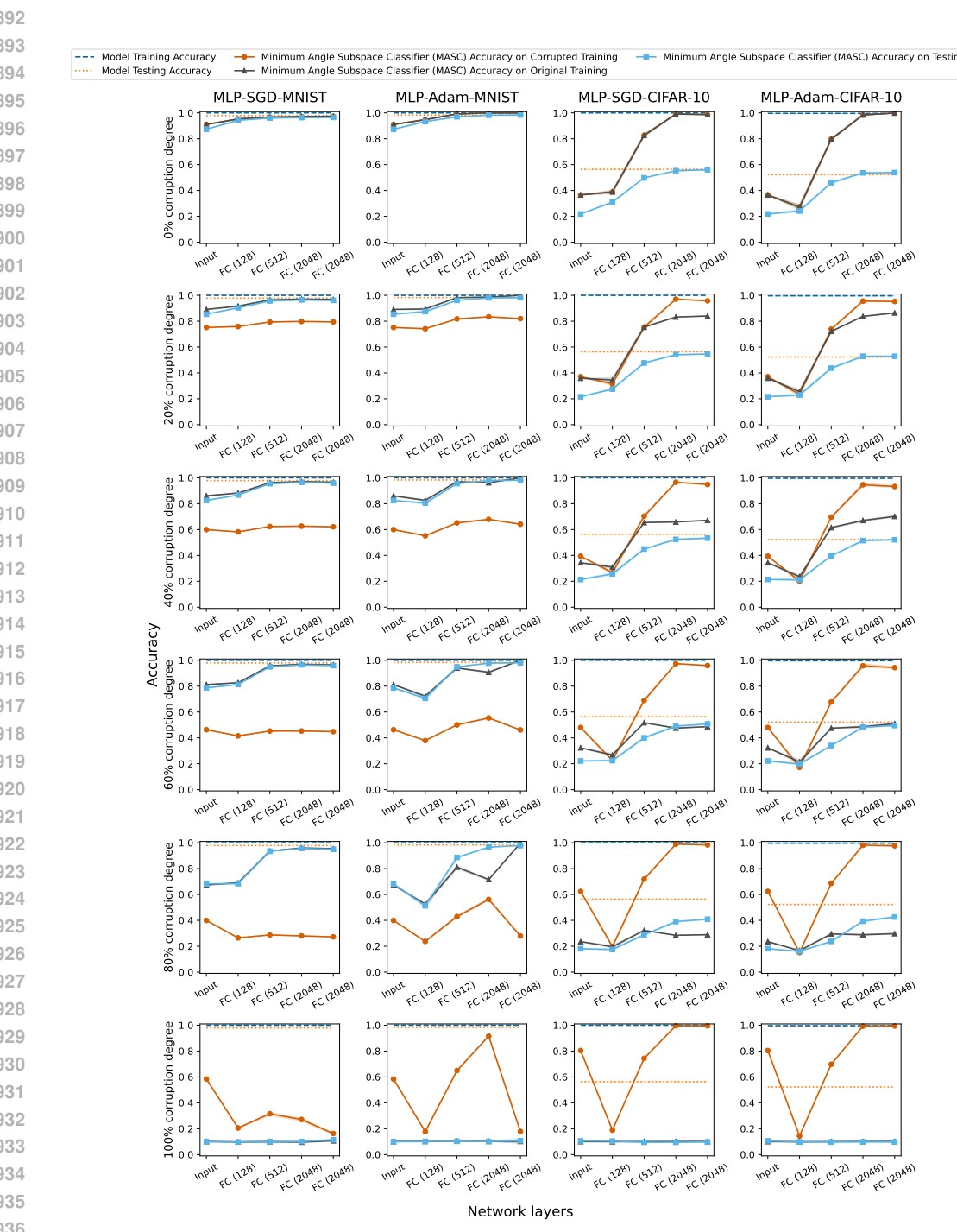

Figure 23: MASC accuracy over the layers of the generalized MLP network when the data set is projected onto corrupted training subspaces with the indicated corruption degree. Rows corresponds to plots which have the same corruption degree & the columns correspond to the generalized models with $SGD$ and $Adam$ as noted. Training & testing accuracy of the generalized model with $SGD$ and $Adam$ is shown. FC corresponds to fully connected layer with $ReLU$ activation.

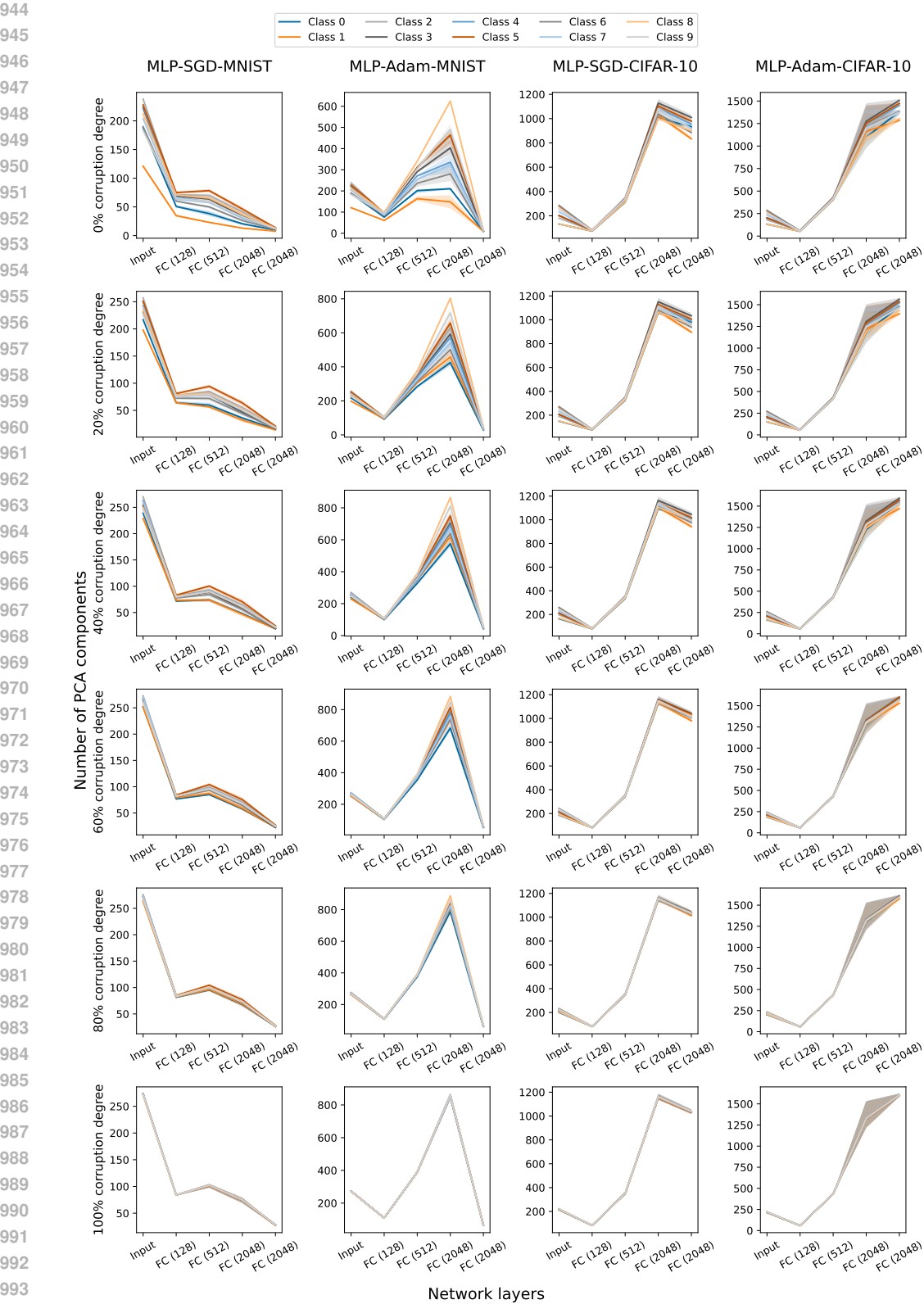

Figure 24: Class-wise number of PCA components of the corrupted training subspace over the layers of generalized MLP network with various corruption degrees. Rows corresponds to plots which have the same corruption degree and the columns correspond to the models with $SGD$ and $Adam$ optimizer as noted.

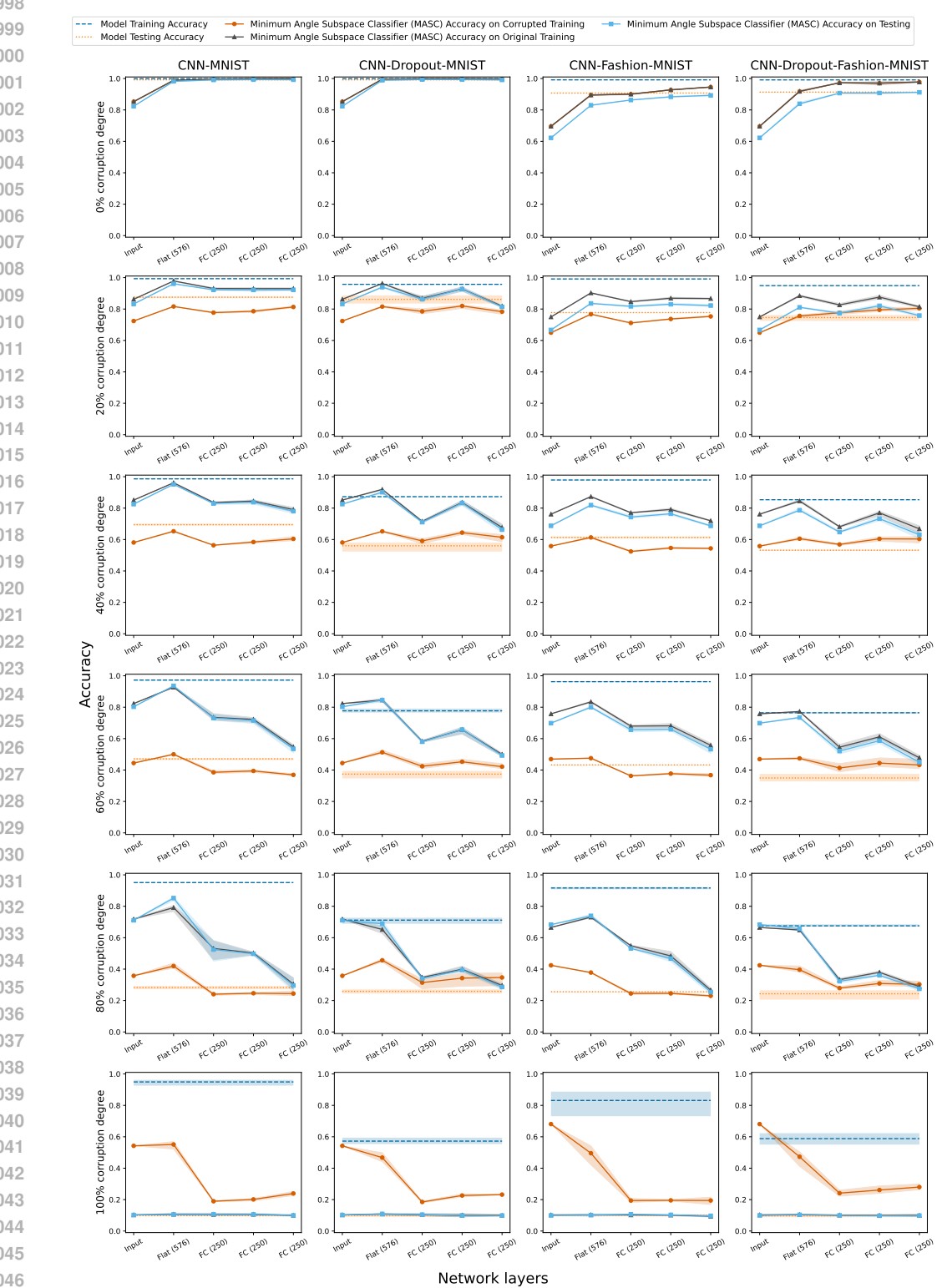

Figure 25: MASC accuracy over the layers of the network when the data is projected onto corrupted training subspaces with the indicated corruption degree, for multiple CNN models trained with and without dropout. Rows corresponds to plots with the same corruption degree & the columns correspond to the models, as noted. Training accuracy (dashed line) & testing accuracy (dotted line) of the model is shown. FC corresponds to fully connected layer with $ReLU$ activation.

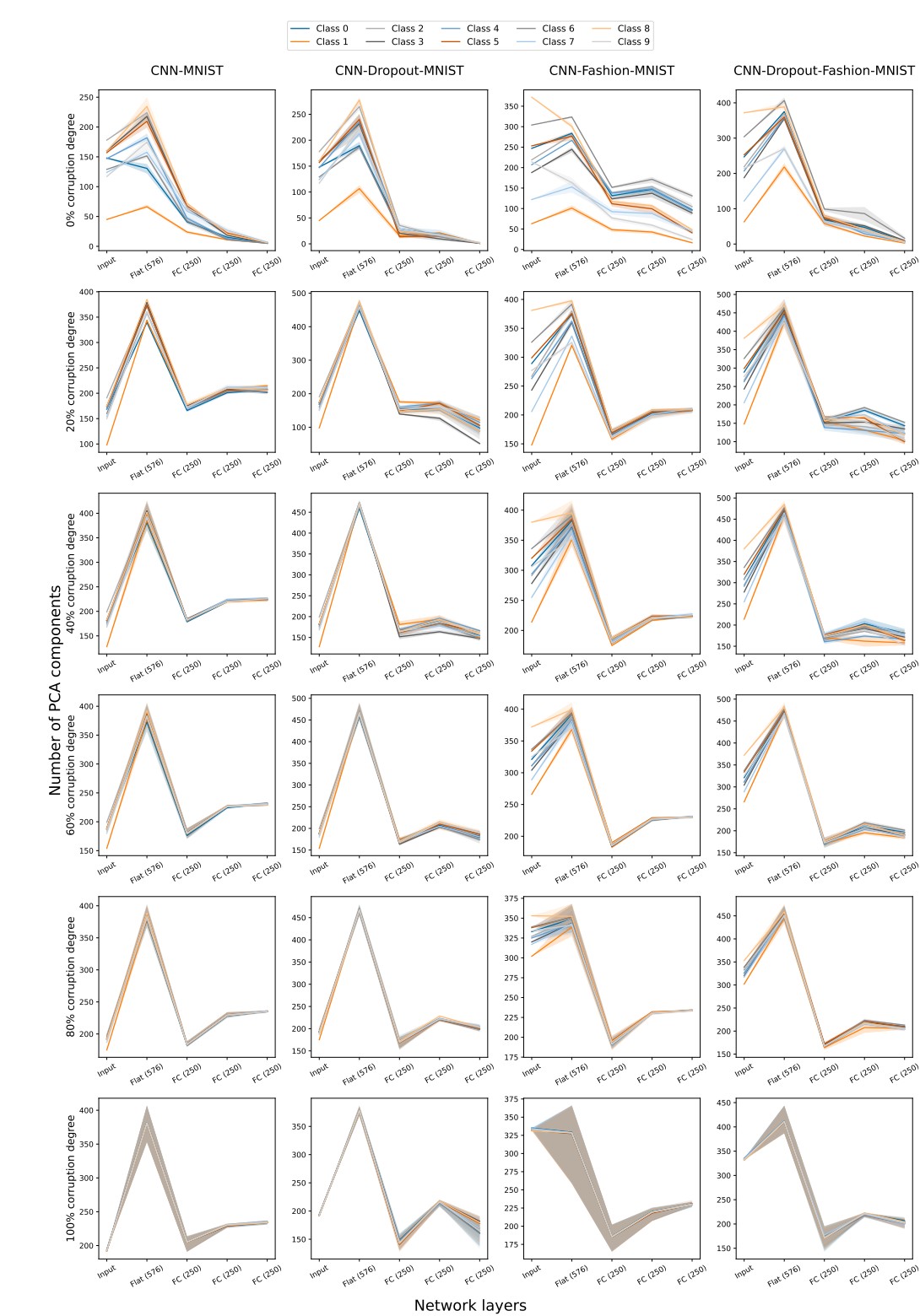

Figure 26: Class-wise number of PCA components of the corrupted training subspace over the layers of CNN networks trained with and without dropout, for various corruption degrees. Rows corresponds to plots which have the same corruption degree and the columns correspond to the models as noted.

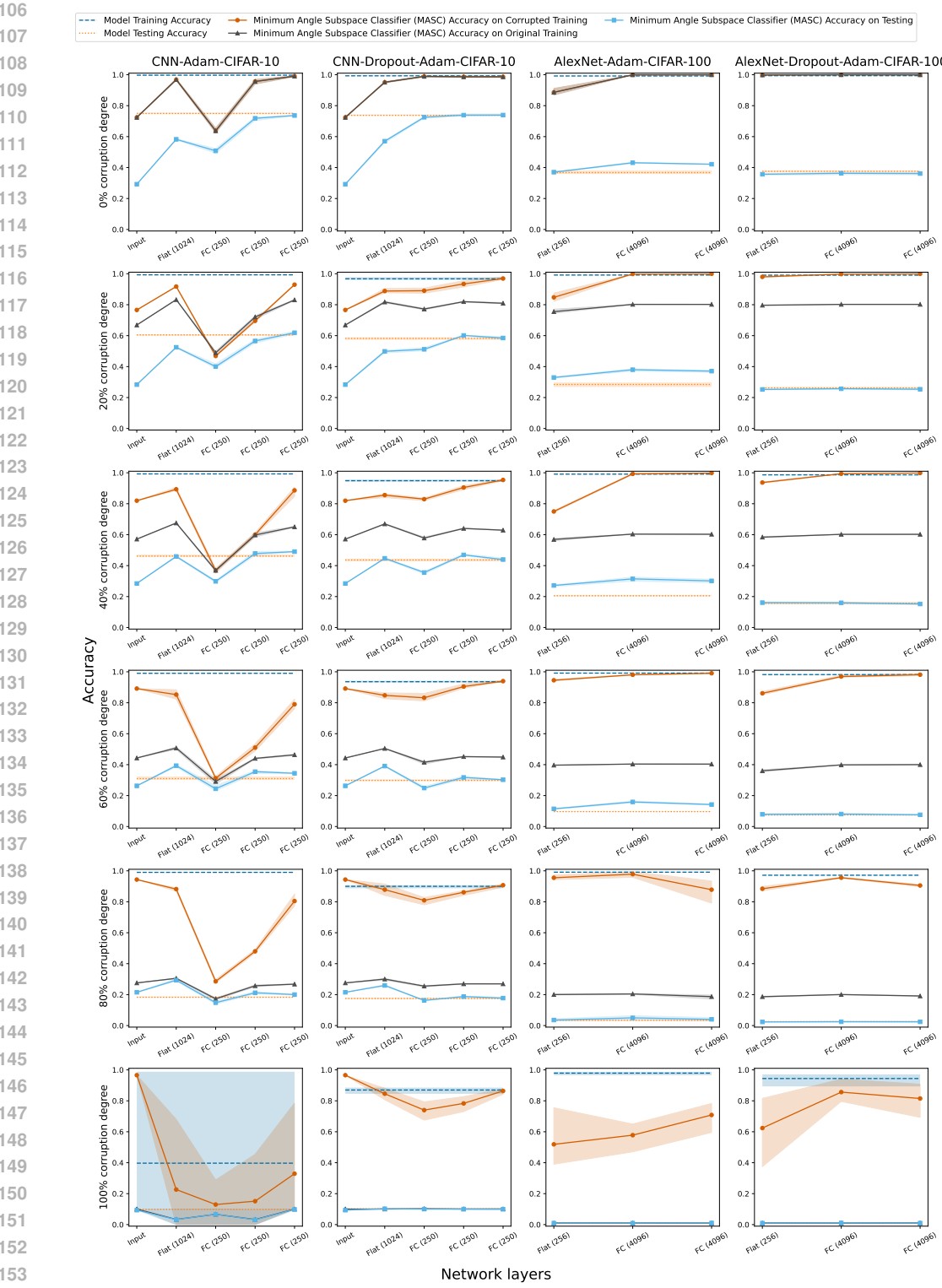

Figure 27: MASC accuracy over the layers of the network when the data is projected onto corrupted training subspaces with the indicated corruption degree, for CNN and AlexNet models trained with and without dropout. Rows corresponds to plots with the same corruption degree & the columns correspond to the models as noted. Training accuracy (dashed line) & testing accuracy (dotted line) of the model is shown. FC corresponds to fully connected layer with $ReLU$ activation hereas Flat corresponds to flatten layer without $ReLU$ activation.

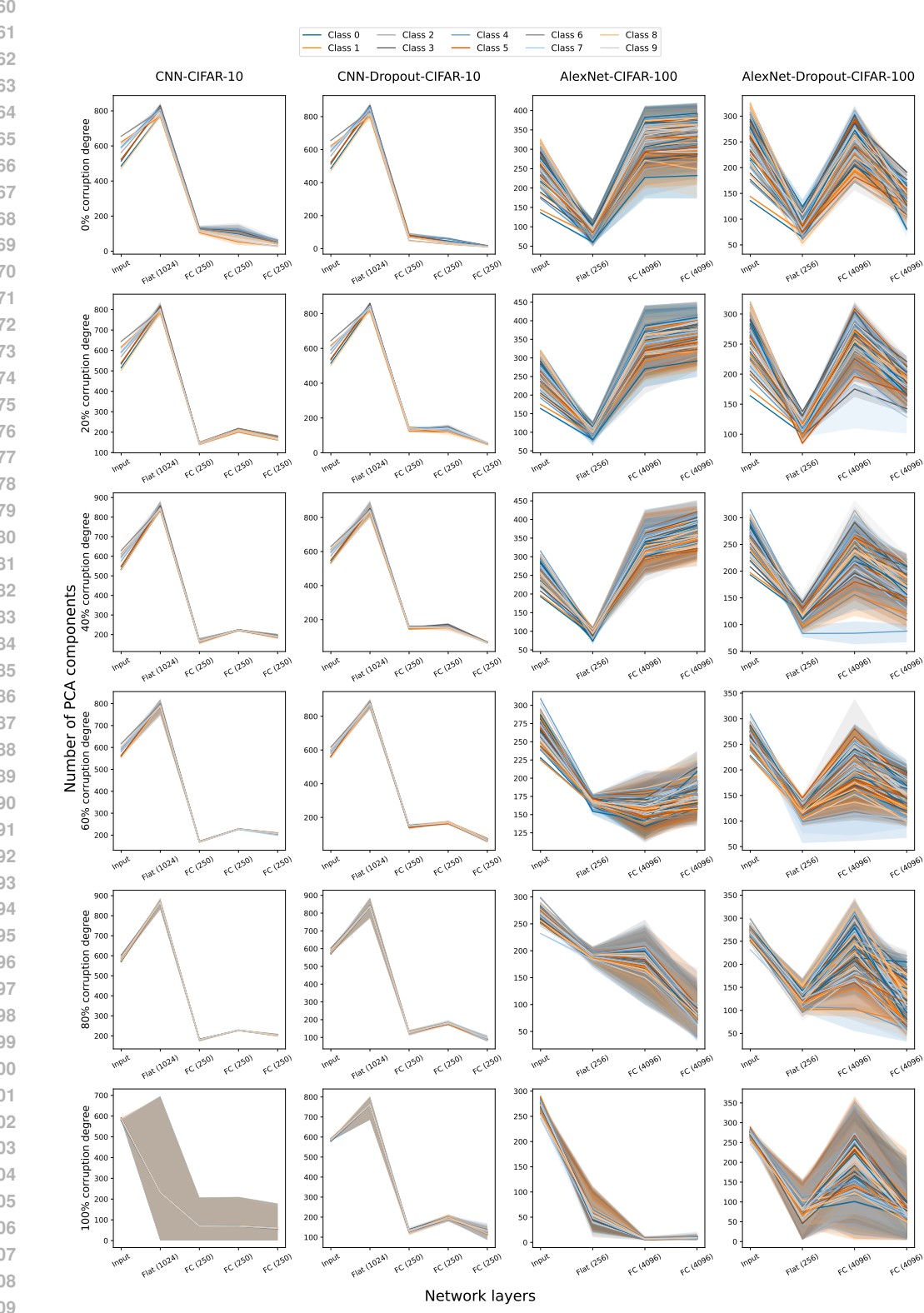

Figure 28: Class-wise number of PCA components of the corrupted training subspace over the layers of CNN and AlexNet networks trained with and without dropout, for various corruption degrees. Rows corresponds to plots which have the same corruption degree and the columns correspond to the models as noted.

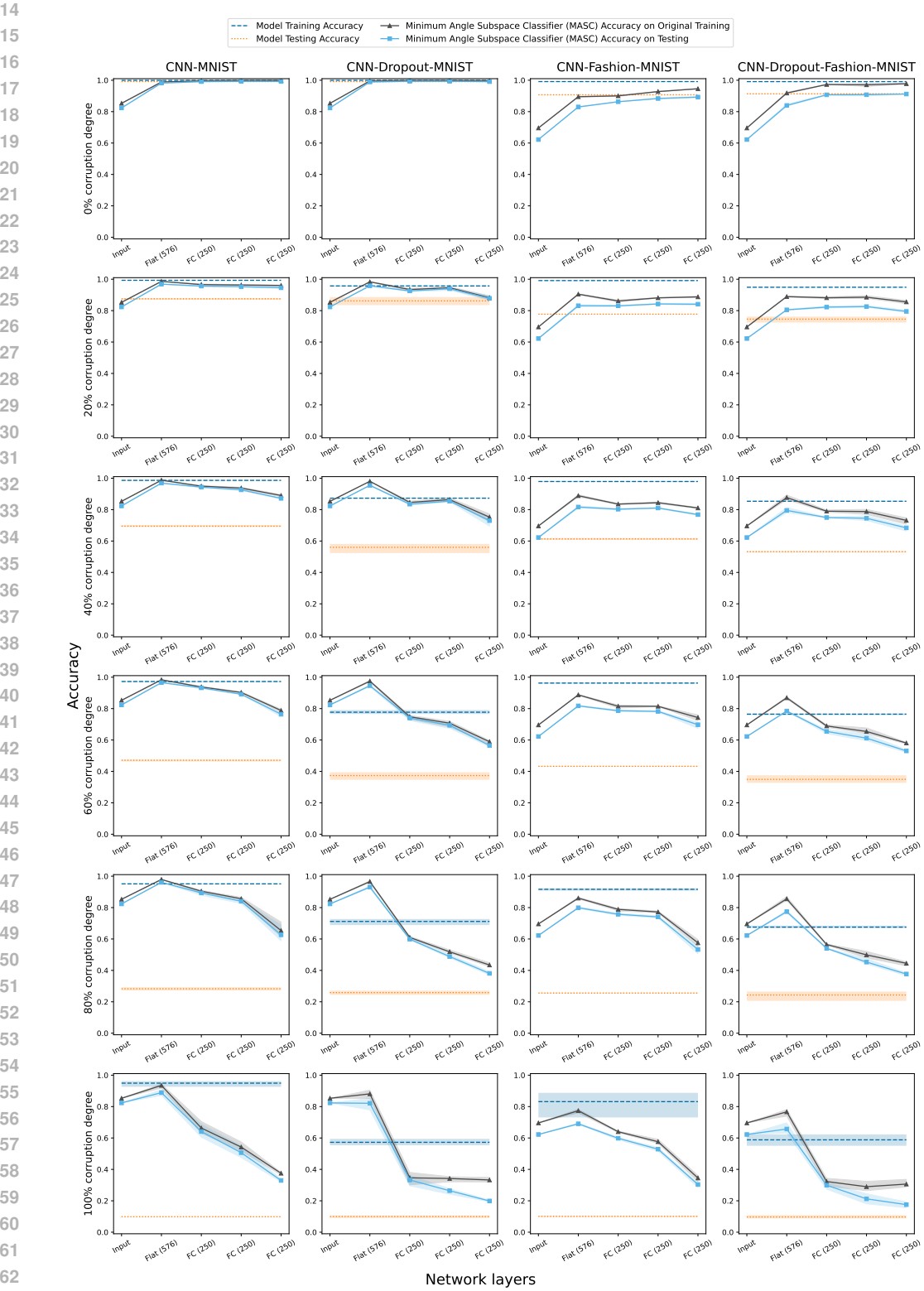

Figure 29: MASC accuracy over the layers of the CNN network, trained with and without dropout, when the data set is projected onto subspace corresponding to true training labels. Rows corresponds to plots which have the same corruption degree and the columns correspond to the models as noted. Training and testing accuracy of the model is shown. FC corresponds to fully connected layer with $ReLU$ activation whereas Flat corresponds to flatten layer without $ReLU$ activation.

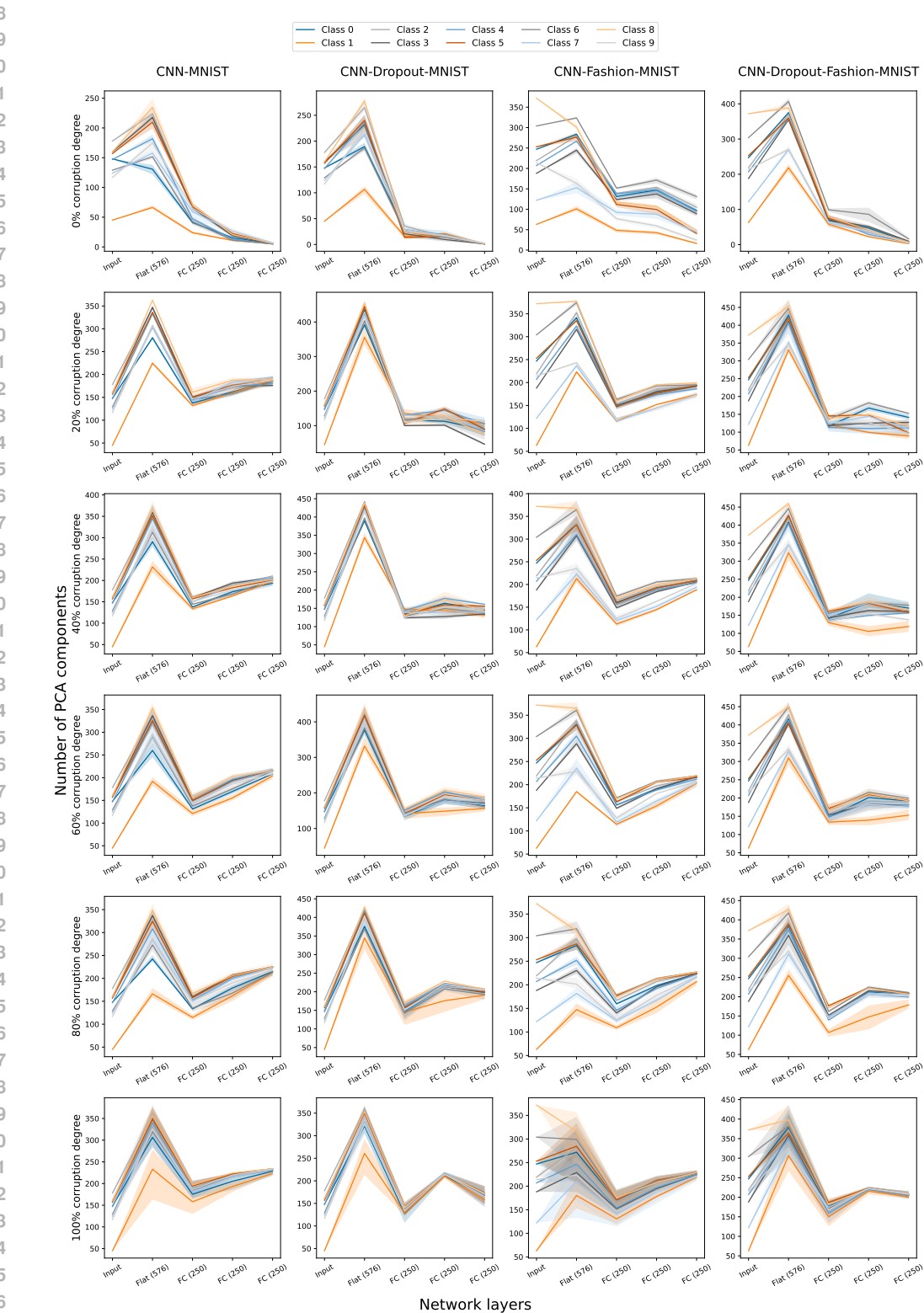

Figure 30: Class-wise number of PCA components of the subspace corresponding to true training labels over the layers of CNN networks trained with and without drop out and various corruption degrees. Although it is not mentioned in the legend, all the 100 classes of CIFAR-100 are plotted. Rows corresponds to plots which have the same corruption degree and the columns correspond to the models as noted.

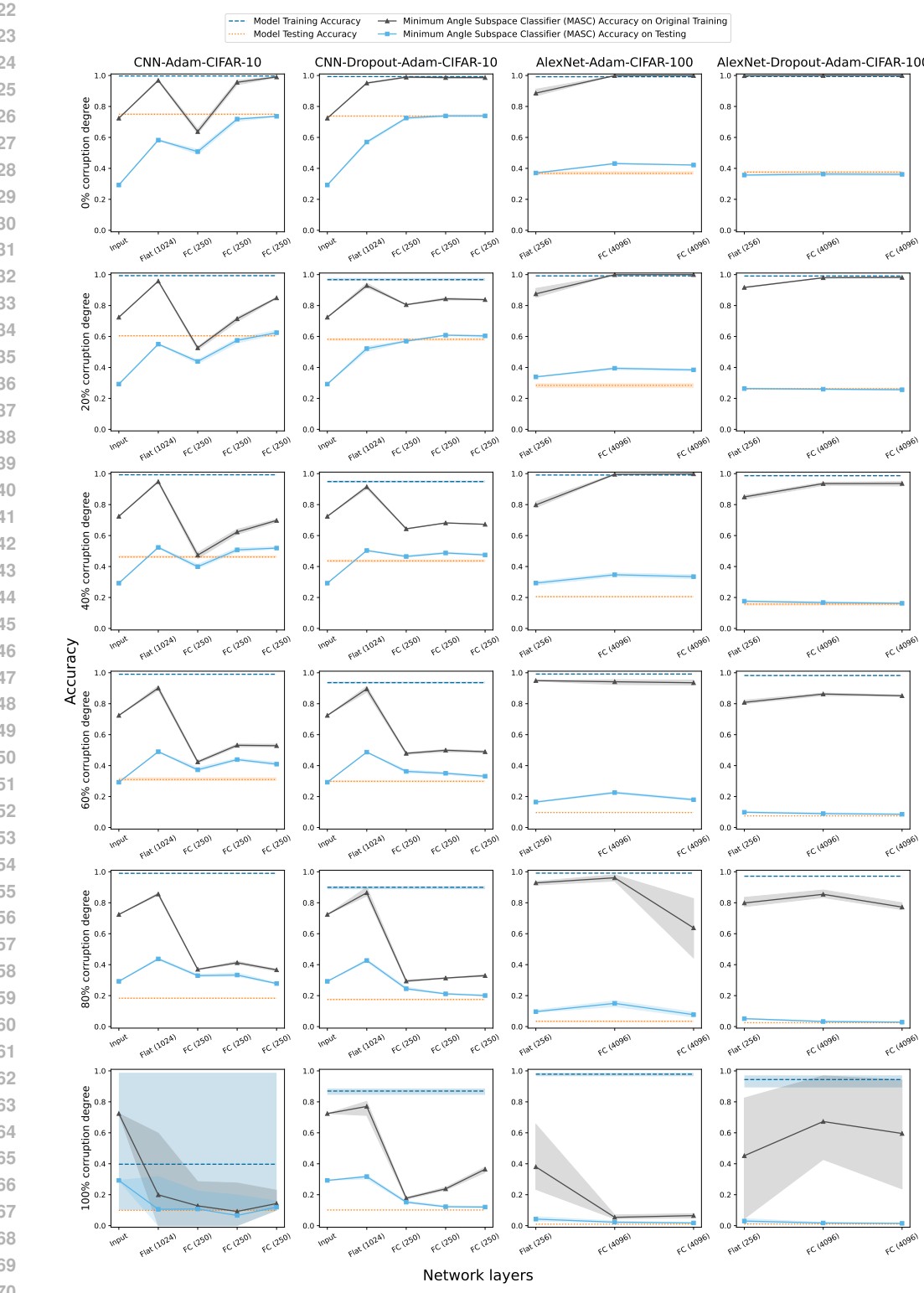

Figure 31: MASC accuracy over the layers of the CNN and AlexNet, trained with and without dropout, when the data set is projected onto subspace corresponding to true training labels. Rows corresponds to plots which have the same corruption degree and the columns correspond to the models as noted. Training and testing accuracy of the model is shown. FC corresponds to fully connected layer with $ReLU$ activation whereas Flat corresponds to flatten layer without $ReLU$ activation.

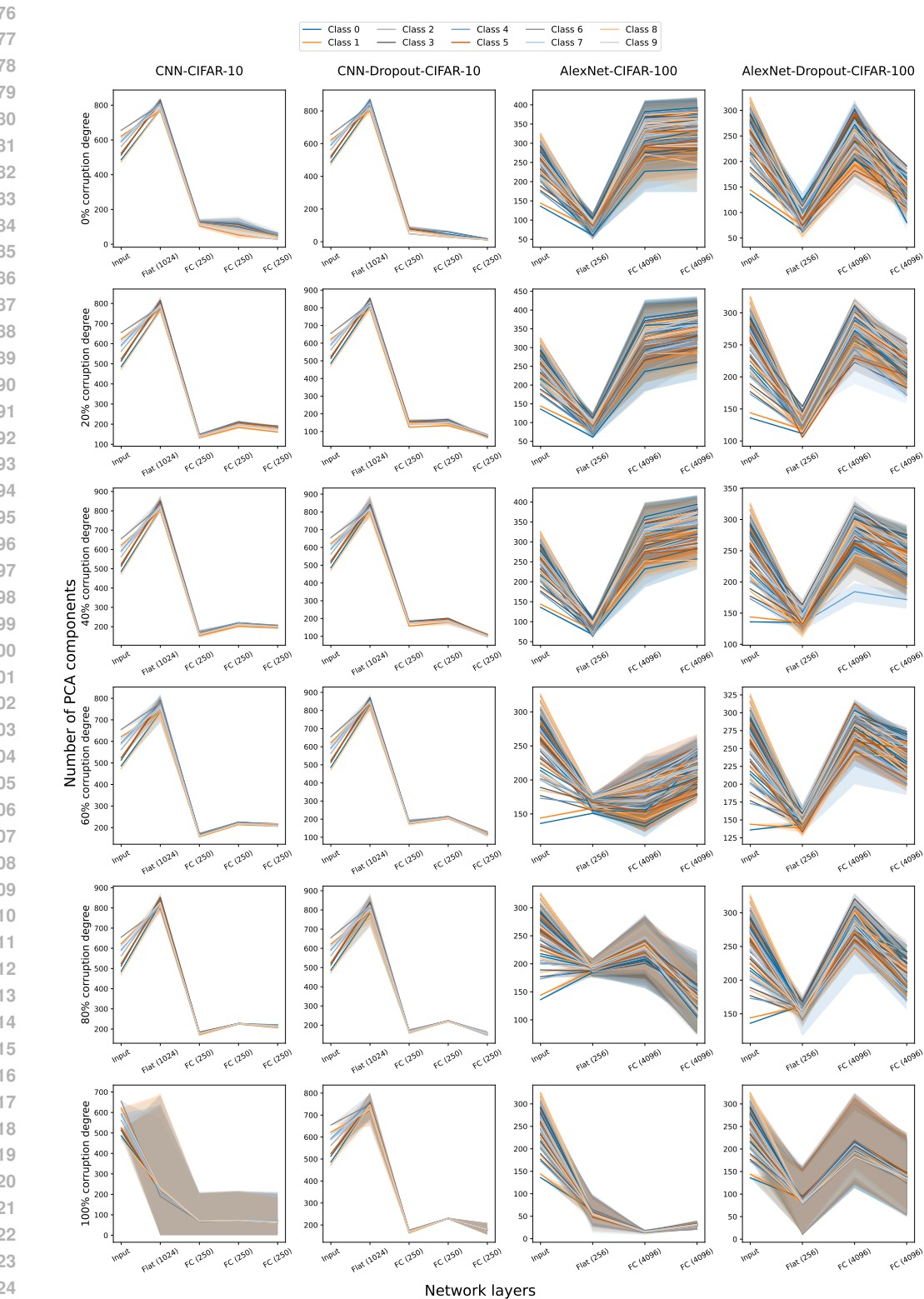

Figure 32: Class-wise number of PCA components of the subspace corresponding to true training labels over the layers of CNN and AlexNet models trained with and without drop out and various corruption degrees. Although it is not mentioned in the legend, all the 100 classes of CIFAR-100 are plotted. Rows corresponds to plots which have the same corruption degree and the columns correspond to the models as noted.

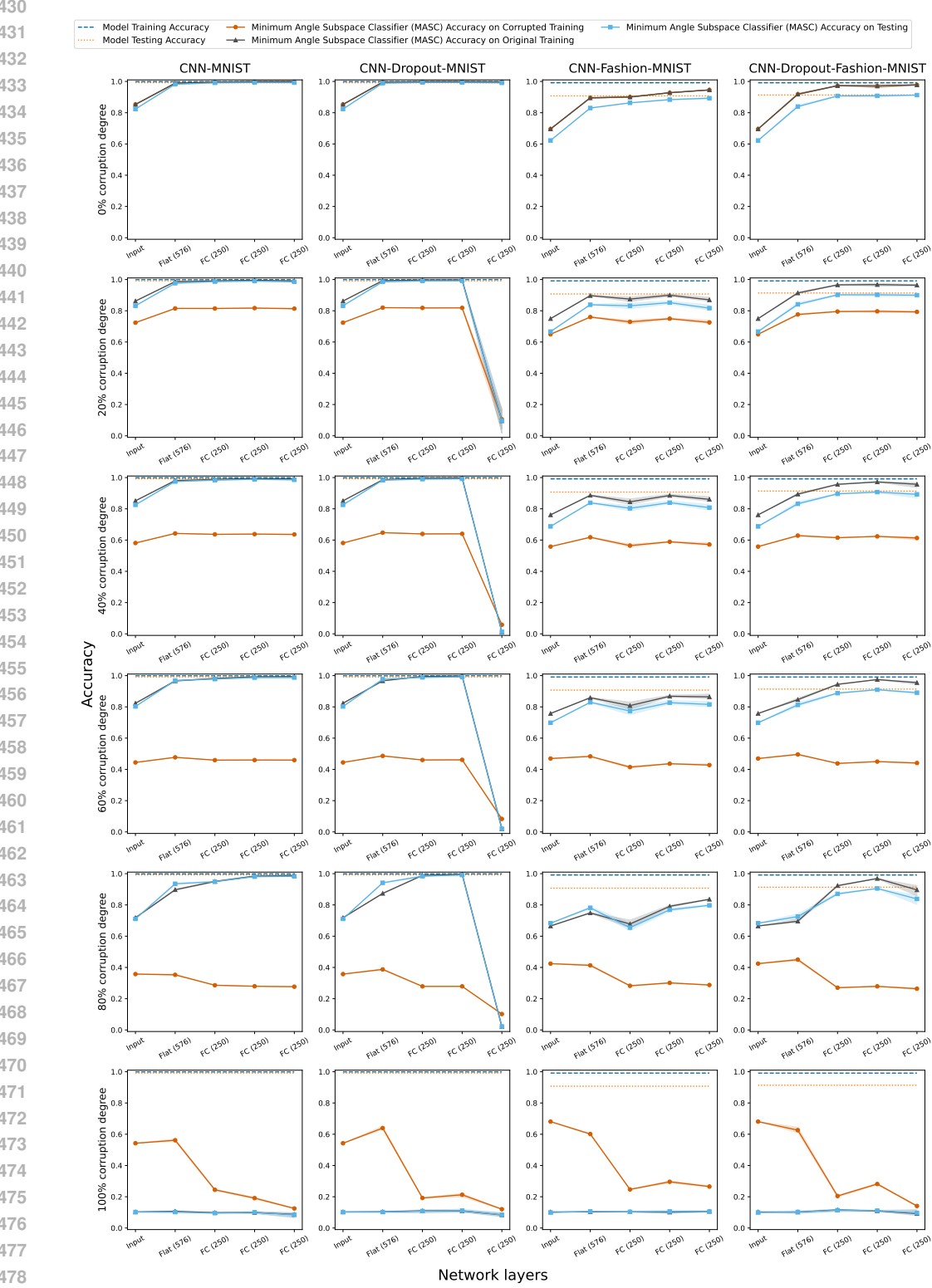

Figure 33: MASC accuracy over the layers of the generalized CNN network, trained with and without drop out, when the data set is projected onto corrupted training subspaces with the indicated corruption degree. Rows corresponds to plots which have the same corruption degree & the columns correspond to the generalized models as noted. Training & testing accuracy of the generalized model is shown. FC corresponds to fully connected layer with $ReLU$ activation.

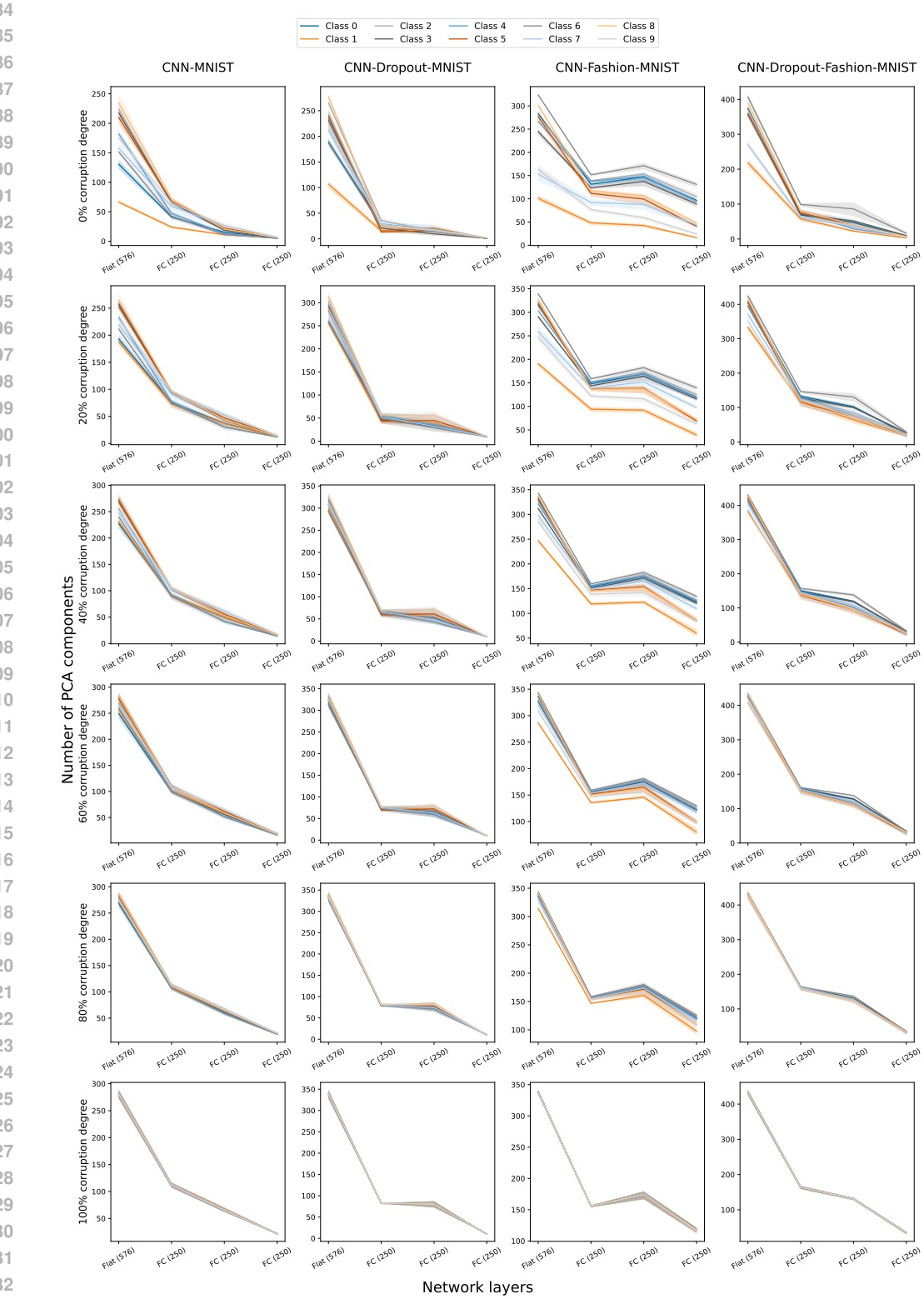

Figure 34: Class-wise number of PCA components of the corrupted training subspace over the layers of generalized CNN networks with various corruption degrees. Although it is not mentioned in the legend, all the 100 classes of CIFAR-100 are plotted. Rows corresponds to plots which have the same corruption degree and the columns correspond to the models with and without drop out as noted.

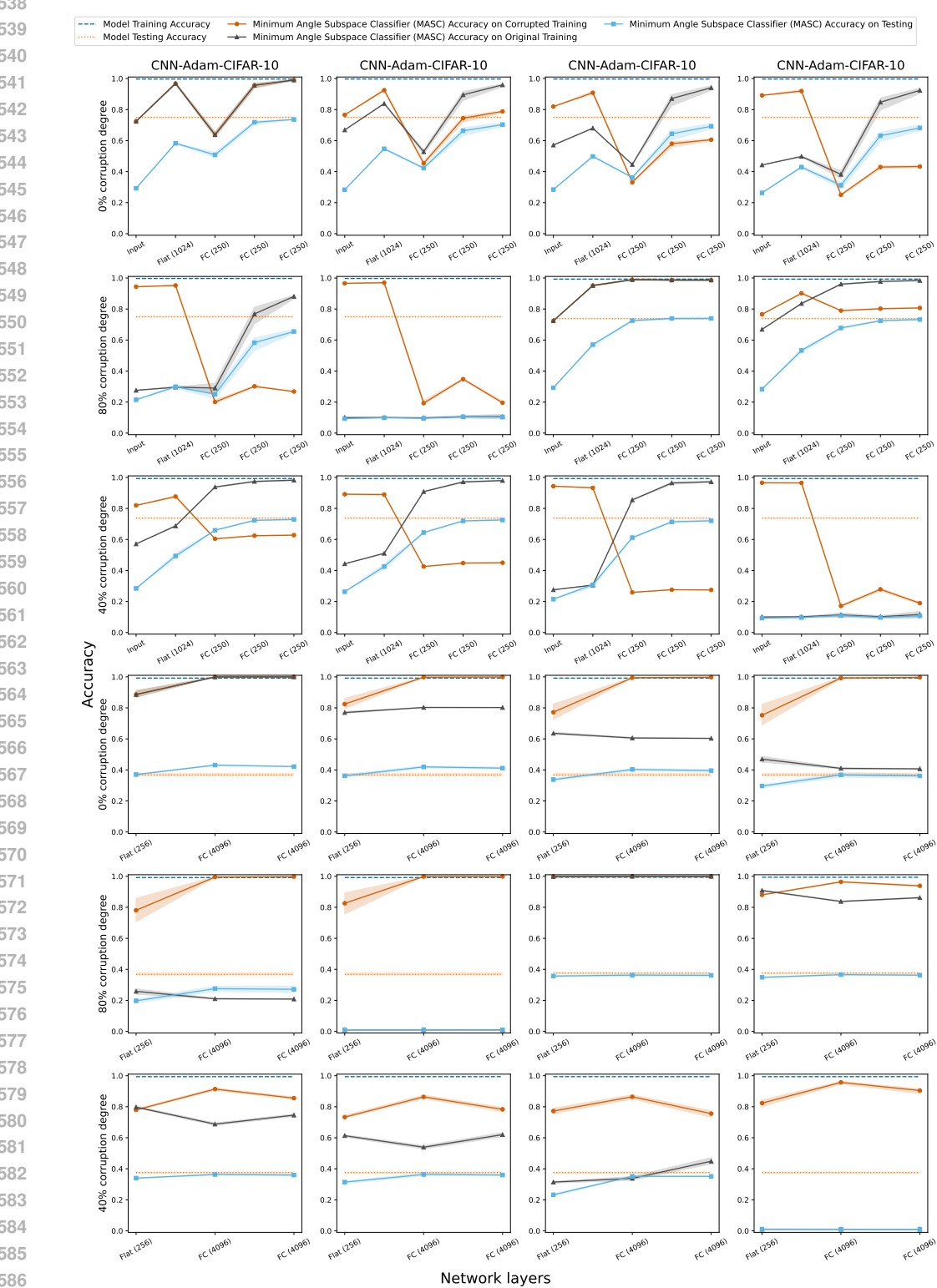

Figure 35: MASC accuracy over the layers of the generalized CNN and AlexNet network, when the data set is projected onto corrupted training subspaces with the indicated corruption degree. Rows corresponds to plots which have the same corruption degree & the columns correspond to the generalized models trained with and without drop out as noted. Training & testing accuracy of the generalized model is shown. FC corresponds to fully connected layer with $ReLU$ activation.

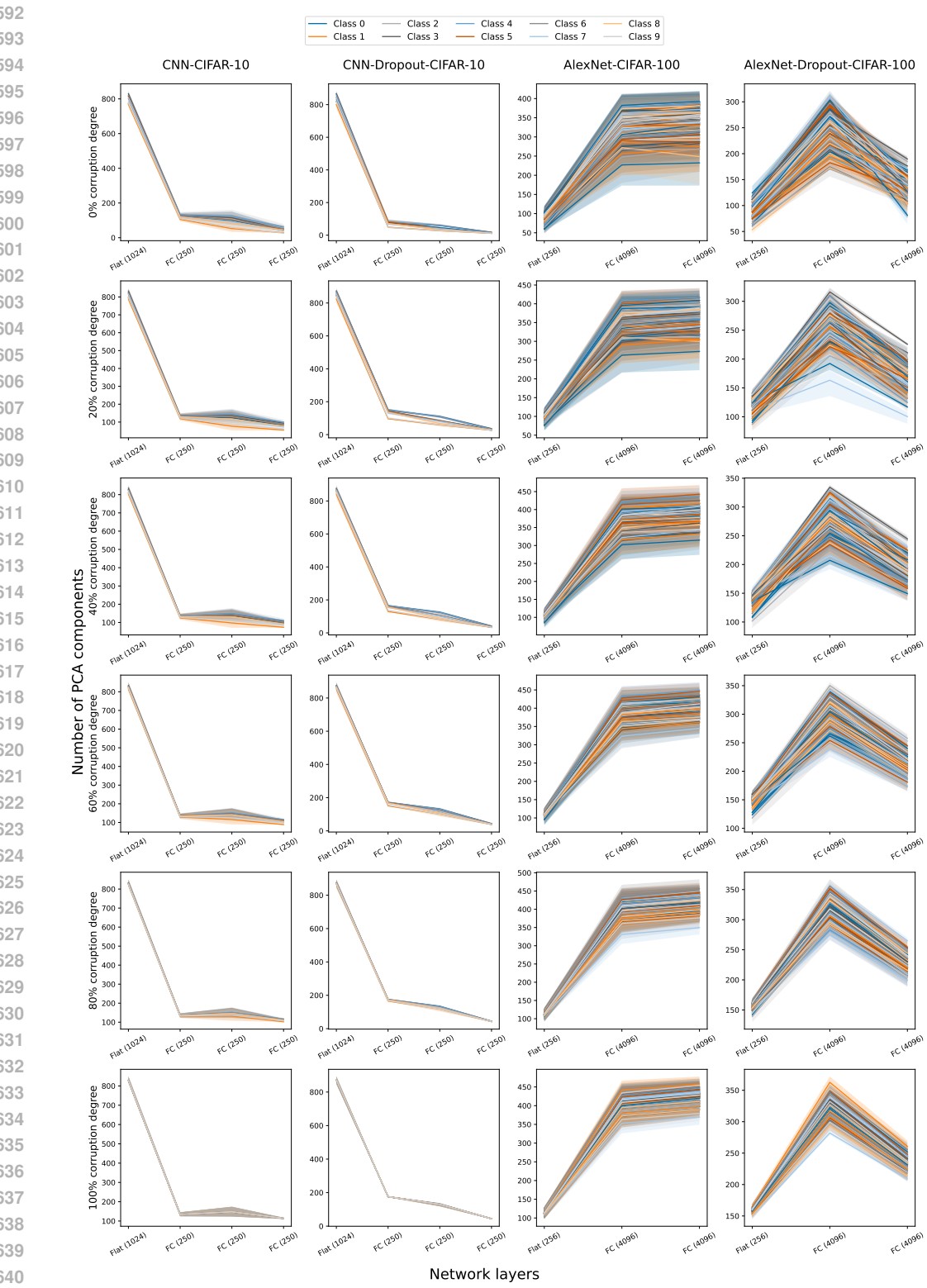

Figure 36: Class-wise number of PCA components of the corrupted training subspace over the layers of generalized CNN and AlexNet networks with various corruption degrees. Although it is not mentioned in the legend, all the 100 classes of CIFAR-100 are plotted. Rows corresponds to plots which have the same corruption degree and the columns correspond to the models with and without drop out as noted.

accuracy on testing is dependant on the percentage of variance captured by the PCA components. It can be observed by the decrease in training and testing accuracy from 99% variance to 90% variance captured plots. Although we have done this experiment on only one model and dataset, further investigation involving the the impact of number of PCA components to the MASC accuracies will be instructive.

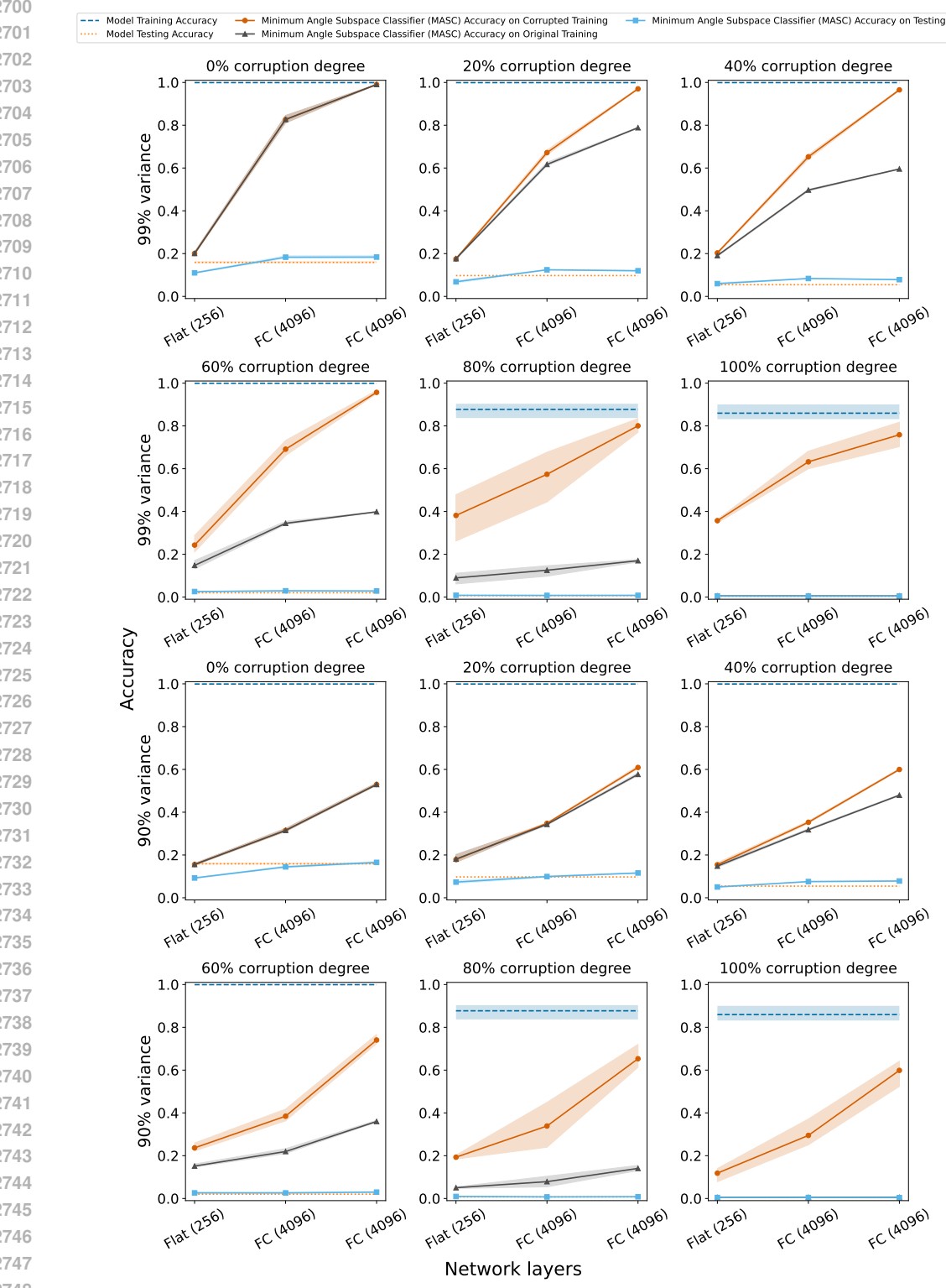

Figure 37: MASC accuracy over the layers of the AlexNet model trained on Tiny ImageNet when the data is projected onto 99% and 90% variance explained corrupted training subspaces with the indicated corruption degree, for multiple models/datasets. Rows corresponds to plots with the same corruption degree & the columns correspond to the models, as noted. Training accuracy (dashed line) & testing accuracy (dotted line) of the model is shown. FC corresponds to fully connected layer with $ReLU$ activation whereas Flat corresponds to flatten layer without $ReLU$ activation.

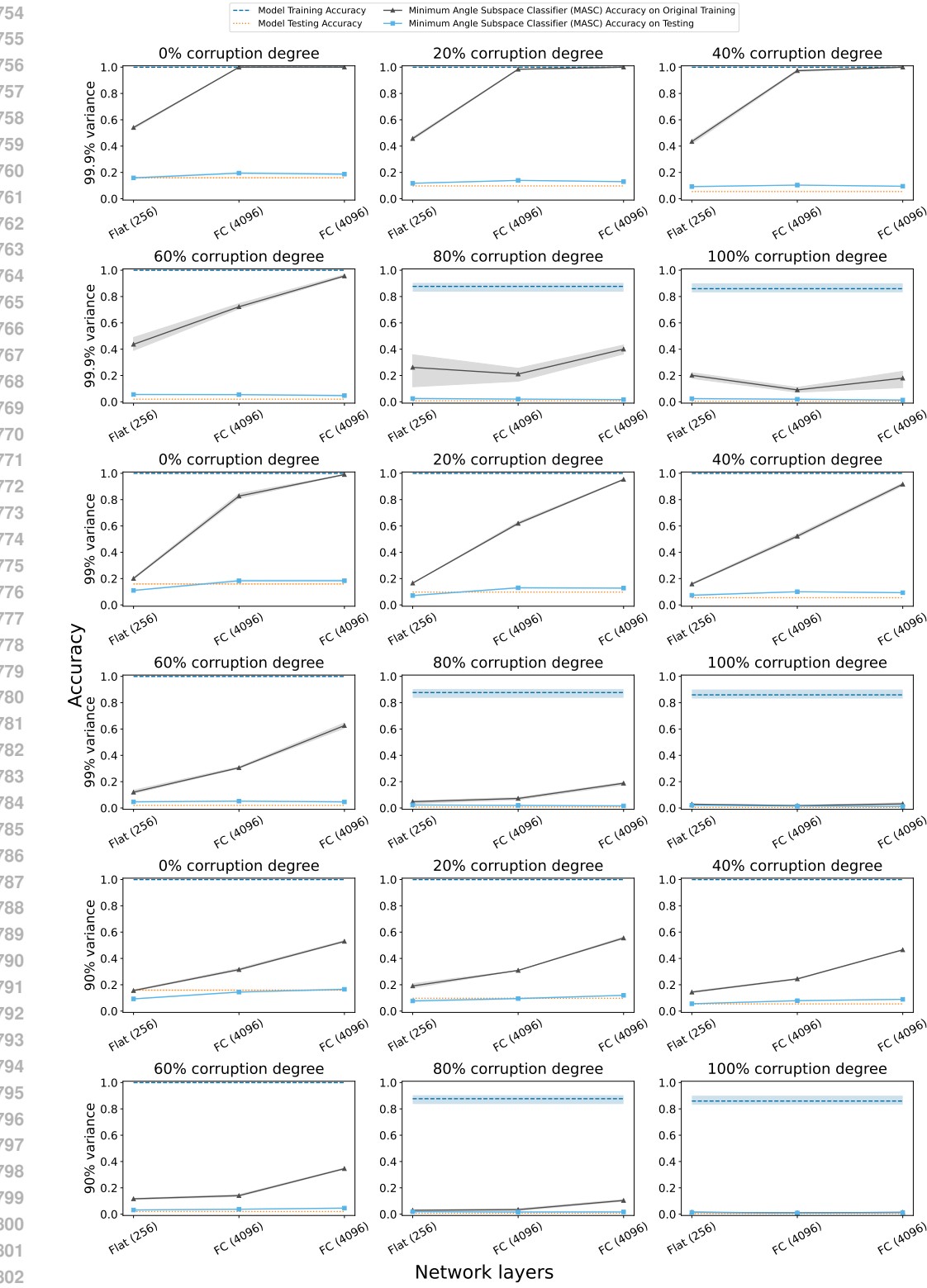

Figure 38: MASC accuracy over the layers of the AlexNet models trained on Tiny ImageNet when the data set is projected onto 99.9%, 99% and 90% variance explained subspace corresponding to true training labels. Rows corresponds to plots which have the same corruption degree and the columns correspond to the models as noted. Training and testing accuracy of the model is shown. FC corresponds to fully connected layer with $ReLU$ activation whereas Flat corresponds to flatten layer without $ReLU$ activation.

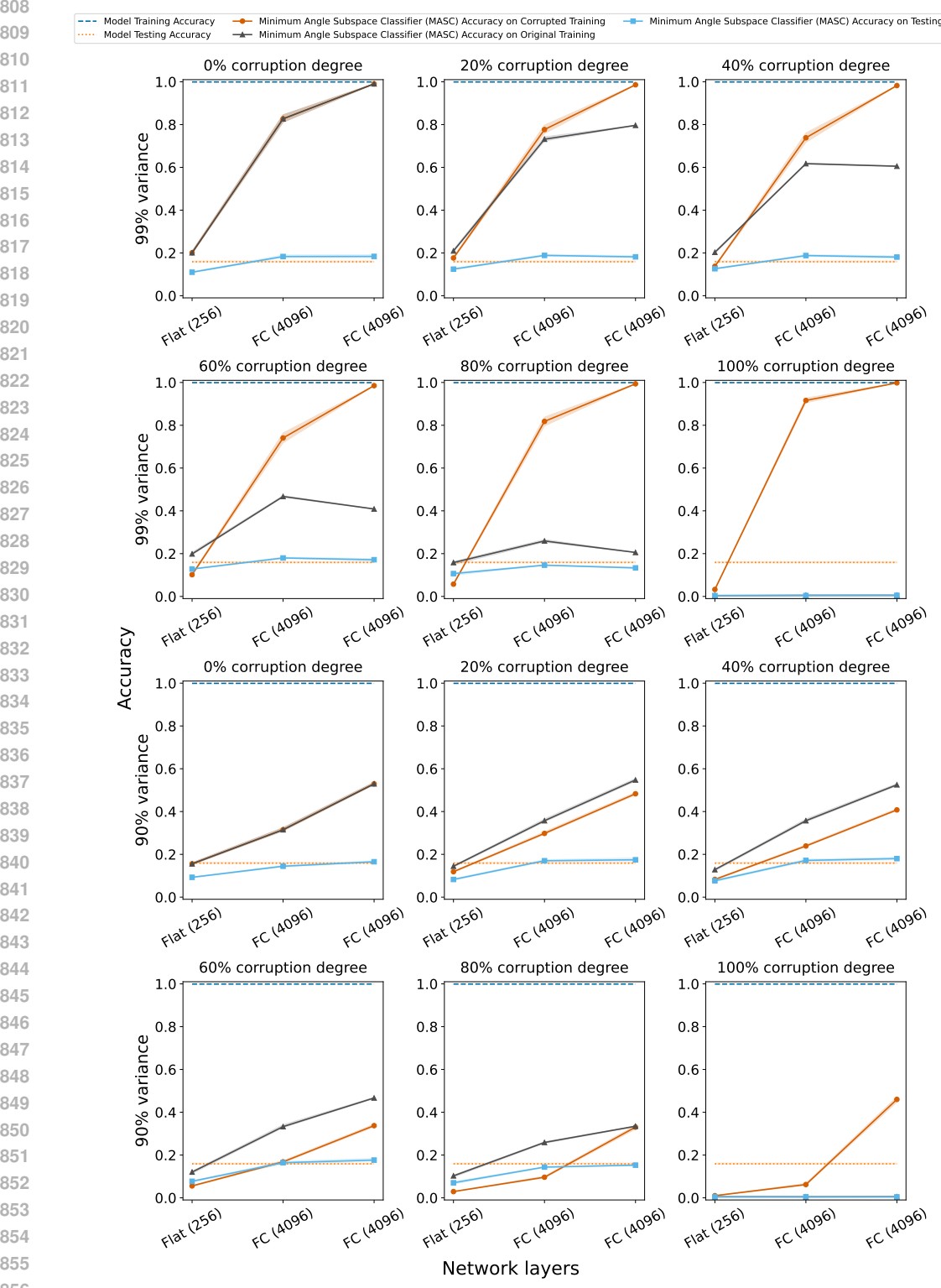

Figure 39: MASC accuracy over the layers of the generalized AlexNet model trained on Tiny ImageNet when the data set is projected onto 99% and 90% variance explained training subspaces with the indicated corruption degrees. Rows corresponds to plots which have the same corruption degree & the columns correspond to the generalized models as noted. Training & testing accuracy of the generalized model is shown. FC corresponds to fully connected layer with $ReLU$ activation whereas Flat corresponds to flatten layer without $ReLU$ activation.

