# OpenReview forum: "Decoding Generalization from Memorization in Deep Neural Networks"
_ICLR.cc/2025/Conference — Submitted to ICLR 2025_

### Official Review · Reviewer_EEmZ · 2024-10-29

**Soundness:** 3
**Presentation:** 2
**Contribution:** 3
**Rating:** 6
**Confidence:** 4

**Summary:**

This paper provide a deeper dive into the representation learning under generalization (learning with clean labels) or memorization (fitting to corrupted labels) for image classification models. The results show that even when heavily memorizing randomly corrupted labels, the intermediate layer representations could still provide non-trivial classification capabilities when probed with a simple classifier without using any extra label information. Furthermore, if the correct labels are provided post-hoc to build such probe, non-trivial accuracy can be obtained even when the original model was trained with completely random labels. This shows that even when fitting to completely random labels, the models are still learning useful visual representations for the input image, instead of arbitrarily wiring the network to build a lookup table for memorizing the random labels. This paper further show that a similar method can be used to build a "probe" with high accuracy for random labels when the original model was trained with correct labels.

**Strengths:**

This paper studies the interesting problem of representation building when the neural network memorizes corrupted training labels. Especially when learning under completely random noises, it was unclear if the neural networks simply ignore the underlying visual patterns and build an arbitrary lookup table for those random labels, or do they still learn useful visual representations. The experiments to study this question is clearly formulated. The conclusions are supported with experiments on multiple datasets and model architectures.

**Weaknesses:**

1. The presentation of the paper could be improved, especially the experiment results and figures. It is a bit difficult to digest or find the most relevant information from their figures. For example, Figure 1, 2, 3 each occupies a full page, and each contains 30 sub-panels. Even if the authors would like to include all the results in the main paper, I would still recommend choosing a subset of panels that can clearly support the conclusions and highlight them (e.g. making them bigger and putting them in a separate figure), or even consider alternative visualization to present a summary of the information contained in each of the 30 panels.

2. While the message in this paper is clear, it seems a bit specialized to the specifically chosen Minimum Angle Subspace Classifier (MASC). It is a bit unclear if it is the robustness of this classifier that enabled such phenomenon or does it hold for other simple classifiers as well. I think it is still interesting if it only holds for MASC, but including studies with other simple classifiers would make the results more comprehensive.

3. It looks like the conclusion in this paper is biasing towards that the models learn similar representation in both memorization and generalization model, because the learned representation can be turned to do memorization or generalization when the corrupted or true labels are revealed after the representation learning. If this is indeed the message, it would be great if the paper could have some way to measure the representation similarity, which would not only further confirm the message, but might also be able to allow us to do comparative studies such as, do Convolutional Nets have a stronger bias towards learning similar representations than MLPs?

4. (Minor) The MASC algorithm needs to use labels. I believe it is using the same labels as model training (e.g. corrupted labels in most cases) based on the comparison results in Section 4. It would be better if the paper could clearly clarify this in the methodology presentation (e.g. Section 2).

**Questions:**

See "Weakness" above.

---

> ### Author Response · Authors · 2024-11-25
> **Part-1 of author response to Reviewer EEmZ**
>
> We would like to thank the reviewer for their thoughtful review and interesting questions. We respond individually to the weaknesses/questions below.
>
> > W1. The presentation of the paper could be improved, especially the experiment results and figures. It is a bit difficult to digest or find the most relevant information from their figures. For example, Figure 1, 2, 3 each occupies a full page, and each contains 30 sub-panels. Even if the authors would like to include all the results in the main paper, I would still recommend choosing a subset of panels that can clearly support the conclusions and highlight them (e.g. making them bigger and putting them in a separate figure), or even consider alternative visualization to present a summary of the information contained in each of the 30 panels.
>
> We appreciate this suggestion. We have kept the results of one dataset per model in the main paper (namely MLP-MNIST, CNN-Fashion-MNIST and AlexNet-CIFAR-100) and have moved the remaining two models/datasets (i.e. MLP-CIFAR10 and AlexNet-TinyImageNet) to the Appendix (Figures 11, 12 and 13), while making changes in the main text, accordingly. This has brought down the number of subpanels to 18 per figure, down from 30 previously, for all three figures. We believe this has improved the cognitive load on first reading of the paper. Additionally, following the suggestion from another reviewer, we have overlaid some of the Figure 1 traces in Figure 2, to allow for easier comparison between the case of the MASC classifier using corrupted labels vs. true labels.
>
> > W2. While the message in this paper is clear, it seems a bit specialized to the specifically chosen Minimum Angle Subspace Classifier (MASC). It is a bit unclear if it is the robustness of this classifier that enabled such phenomenon or does it hold for other simple classifiers as well. I think it is still interesting if it only holds for MASC, but including studies with other simple classifiers would make the results more comprehensive.
>
> This is a great question but we are unable to address it during the Discussion period due to computational constraints. We think the results of the MASC classifier open the door to the possibility of designing better classifiers that operate on layerwise outputs. We have discussed this important point in Discussion (Lines 525-527).

---

> ### Author Response · Authors · 2024-11-25
> **Part-2 of author response to Reviewer EEmZ**
>
> > W3. It looks like the conclusion in this paper is biasing towards that the models learn similar representation in both memorization and generalization model, because the learned representation can be turned to do memorization or generalization when the corrupted or true labels are revealed after the representation learning. If this is indeed the message, it would be great if the paper could have some way to measure the representation similarity, which would not only further confirm the message, but might also be able to allow us to do comparative studies such as, do Convolutional Nets have a stronger bias towards learning similar representations than MLPs?
>
> While this is an important question that requires detailed investigation, it wasn’t our intent in the paper to say that models learn similar representations in both memorization or generalization. This direction could be valuable in understanding the distinction between representations in these two regimes (i.e. memorized and generalized networks) and how it relates to their ability to generalize. Since it requires deeper consideration and detailed experiments, it has been beyond the scope of what we could address during the Discussion period. We have however briefly mentioned this question in the Discussion section (Lines 529-531).
>
> > W4. (Minor) The MASC algorithm needs to use labels. I believe it is using the same labels as model training (e.g. corrupted labels in most cases) based on the comparison results in Section 4. It would be better if the paper could clearly clarify this in the methodology presentation (e.g. Section 2).
>
> Thank you for pointing this out. We have added text in the erstwhile Section 2 (i.e. current Section 3) to fix this (Lines 159-161): “MASC is using labels of the dataset while creating the class-specific subspaces. For experiments in Section 4, MASC uses corrupted training labels whereas in Section 5, MASC uses true training labels to create class-specific subspaces.”
>
> We are happy to respond to any further questions/suggestions or offer additional clarifications, should the reviewer need them.

---

> > ### Author Response · Authors · 2024-11-30
> > **Gentle reminder to Reviewer EEmZ**
> >
> > Dear Reviewer,
> >
> > We hope this finds you well.
> >
> > In case it escaped your attention, we want to remind you about our response to your review.
> >
> > As you might be aware, the revised deadline for reviewer responses is approaching, i.e. Dec 2nd, AoE.
> >
> > Once again, we appreciate your time and efforts. We look forward to constructively engaging with you.

---

### Official Review · Reviewer_Hw6o · 2024-11-01

**Soundness:** 1
**Presentation:** 2
**Contribution:** 2
**Rating:** 3
**Confidence:** 3

**Summary:**

First, the paper presents experiments where the model attains high training accuracy but low test accuracy, the classical overfitting story. Then, they show that by using the intermediate activation of the neural network, they can get reasonable generalization performance beyond the naive test accuracy. The higher-level claim is that the generalization ability is not lost during the overfitting process, that ability can be "decoded" via techniques like MASC.

**Strengths:**

The idea that overfitting happens in the later layer of the network, though not new, is interesting in itself. The authors present a concrete technique how to extract the generalization performance from those activations.

**Weaknesses:**

First of all, the technique discussed in the paper is not a practical one -- they are not intended for people to use in practice to replace more careful network architecture design, hyperparameter tuning, data curation, etc.

So the paper has to be viewed from a scientific perspective.
- Novelty: the idea of using unsupervised technique on activation to get good generalization performance is not new. It roots in the rich literature of representation learning, which are not adequately discussed in the paper.
- Scientific rigor: The phenomenon isn't as robust as the experiment suggests, there are several places in Figure 1 and 2 where the MASC accuracy is not higher than test accuracy.  Especially on ImageNet.
- Experiment setting: It's sometime a bad criticism to say that the authors didn't run large-scale experiments. But in this case, they propose a phenomenon that's intended to hold a cross a wide range of scales, but the largest neural network they implementated is AlexNet. Even under academic budget, the author would need to show results on ResNet (from 2015). Let alone more recent models like MobileNet and ViT to be convincing.

**Questions:**

Given that no mild-scale experiment were presented, how does the phenomenon generalize to large scale neural networks? With ResNet50 and careful training, we can get high accuracy on CIFAR100 and ImageNet.

---

> ### Author Response · Authors · 2024-11-25
> **Part-1 of author response to Reviewer Hw6o**
>
> We thank the reviewer for the detailed comments and questions. We find ourselves especially unable to agree with some of the points made in W2 and W3. We respond individually below.
>
> > W1: First of all, the technique discussed in the paper is not a practical one -- they are not intended for people to use in practice to replace more careful network architecture design, hyperparameter tuning, data curation, etc.
> >
> > So the paper has to be viewed from a scientific perspective.
>
> Indeed, the focus of the paper was on the scientific question – do representations of intermediate layers of deep networks that memorize corrupted training data contain information that allows for significantly better generalization than demonstrated by the model, and we believe that our results are novel and surprising. That said – and as we have noted in the Discussion section (Lines 522 - 529) – the work could have implications for the pragmatic question of extracting better generalization from networks, especially those whose training data is noisy. This has however been beyond the scope of this paper. Indeed, scientific study of similar questions has previously resulted in the design of techniques with practical utility. For example, [Arpit et al, 2017] ran a number of experiments to study the nature of memorization and generalization in networks trained on corrupted data. One of their observations that test accuracy peaked transiently, early on in training, was leveraged by many papers (e.g. [Vahdat, NIPS 2017], [Tanaka et al, CVPR 2018], [Ma et al, ICML 2018], [Yi et al, CVPR 2019], [Liu et al, NeurIPS 2020]) to build new techniques that had improved generalization, in practice.
>
> So, in principle we believe that there is potential to have practical techniques that could be designed in light of our results.
>
> > W2: Novelty: the idea of using unsupervised technique on activation to get good generalization performance is not new. It roots in the rich literature of representation learning, which are not adequately discussed in the paper.
>
> Thank you for this point. We have a new Related Work section with a more comprehensive survey of related literature. We are working on a more detailed account of the representation learning literature, which we expect to include in an updated version to be posted before the end of the Discussion period.
>
> That said, we want to emphasize that the question of extracting generalization from the internals of memorized networks without altering weights is one that has not been studied before. This question, as well as the comprehensive experiments supporting the surprising results, we believe constitute the novelty of this work.
>
> Furthermore, the results we have demonstrated are surprising, and the opposite of what has previously been expected in the literature.
>
> For example, [Alain & Bengio, 2016 arXiv:1610.01644v4 updated in 2018] use linear probes on intermediate layers, they explicitly avoid doing so for networks trained with corrupted labels, since they believe that the probes would overfit. To quote their words (at the end Section 3.4): “Note that we also want to avoid a situation where our probes are simply overfitting on the features because there are too many features. It was recently demonstrated that very large models can fit random labels on ImageNet (Zhang et al., 2016). This is a situation that we want to avoid because the probe measurements would be entirely meaningless in that situation.”
>
> We therefore feel that the above paper conveys a view representative of the field.
>
> EDIT: A few sentences above were edited following a discussion with reviewer vreZ.

---

> ### Author Response · Authors · 2024-11-25
> **Part-2 of author response to Reviewer Hw6o**
>
> > W3: Scientific rigor: The phenomenon isn't as robust as the experiment suggests, there are several places in Figure 1 and 2 where the MASC accuracy is not higher than test accuracy. Especially on ImageNet.
>
> We disagree with this characterization – namely that the phenomenon is not robust. It isn’t our claim in our original submission that MASC test accuracy is better than model test accuracy in every layer. In fact, we specifically point out (see Lines 196-198 of original submission) that in every model we have experimented on, except those with 100% corruption degree, “we find that our Minimum Angle Subspace Classifier (MASC) *in at least one layer*  has better testing accuracy than the corresponding model itself.” (emphasis added).
>
> With  AlexNet-Tiny ImageNet, we acknowledge that from the figures it might have been difficult to assess this point, especially for higher corruption degrees. To address this, we have added a separate Section A.5 in the Appendix with the AlexNet-Tiny ImageNet test results in order to make the distinction clear. Specifically, for the case of AlexNet-Tiny ImageNet, in the table below, we list by what percentage the MASC classifier outperformed the model for the best layer for each corruption degree.
>
> | Corruption degree                                                                                   | 20%   | 40%   | 60%    | 80%    |
> |-----------------------------------------------------------------------------------------------------|-------|-------|--------|--------|
> | Subspace constructed using corrupted labels - Percentage the MASC classifier outperformed the model | 27.51 | 53.67 | 45.04  | 13.69  |
> | Subspace constructed using true labels - Percentage the MASC classifier outperformed the model      | 33.36 | 84.19 | 156.93 | 210.95 |
>
> We thank the reviewer for bringing this to our attention and hope that they find this clarification useful.
>
> > W4: Experiment setting: It's sometime a bad criticism to say that the authors didn't run large-scale experiments. But in this case, they propose a phenomenon that's intended to hold a cross a wide range of scales, but the largest neural network they implementated is AlexNet. Even under academic budget, the author would need to show results on ResNet (from 2015). Let alone more recent models like MobileNet and ViT to be convincing.
> > Q1: Given that no mild-scale experiment were presented, how does the phenomenon generalize to large scale neural networks? With ResNet50 and careful training, we can get high accuracy on CIFAR100 and ImageNet.
>
> Thank you for this question. As the reviewer correctly surmises, budgetary constraints did not afford us the ability to run the mentioned experiments. Indeed, for each model type, we trained 18 models from scratch (a grand total of 234 models spanning the entire paper); models with corrupted training data take longer to train, as is known. Furthermore, determining class-conditioned subspaces and running the MASC classifier on layerwise outputs is also computationally expensive.
>
> We note that it wouldn’t simply be a question of training a single ResNet50 model on ImageNet, but training 18 different ResNet50 models, in addition to the overhead for hyperparameter tuning and determining subspaces and running the MASC classifier for each of the models. However, we believe that the experiments already run on 7 different model/dataset pairs point to the fact that the phenomenon is robust across multiple models/datasets. We will also release the code with the accepted paper, that will facilitate others in running these experiments on other models.
>
> We are happy to respond to any further questions/suggestions or offer additional clarifications, should the reviewer need them.

---

> > ### Author Response · Authors · 2024-11-30
> > **Gentle reminder to Reviewer Hw6o**
> >
> > Dear Reviewer,
> >
> > We hope this finds you well.
> >
> > In case it escaped your attention, we want to remind you about our response to your review.
> >
> > As you might be aware, the revised deadline for reviewer responses is approaching, i.e. Dec 2nd, AoE.
> >
> > Once again, we appreciate your time and efforts. We look forward to constructively engaging with you.

---

### Official Review · Reviewer_vreZ · 2024-11-01

**Soundness:** 3
**Presentation:** 3
**Contribution:** 2
**Rating:** 6
**Confidence:** 4

**Summary:**

In this paper, the authors probe the representations of DNNs trained with noised labels.

To test the hypothesis that intermediate representations "generalize" even if the output layer doesn't, the authors identify class-conditional subspaces in the hidden representations of each layer via PCA. They then show that these subspaces can be used to classify test points by simply choosing the subspace onto whose projection the angular distance to the point is minimal. They then show that such a classifier often has better test performance on models trained with noised labels, confirming the hypothesis.

These subspaces are themselves constructed out of noised labels, but the authors also test the case where they are not, also finding consistent better test performance.

The authors present these findings as additional evidence of the complex interplays between generalization and memorization.

**Strengths:**

The paper investigates important questions in a novel way. It is well written and the claims are mostly backed by evidence.

**Weaknesses:**

The paper's main weakness is a fairly important lack of situating itself with respect to prior work. A number of papers have probed intermediate layers and latent representations of DNNs [e.g. 2,3], in an attempt to understand this very memorization-vs-generalization debate. It does feel to me that many of the claims made in this paper are obvious in light of this literature. While the specific empirical investigation done here is novel to me, the conclusions drawn by the authors are not.
It is hard to resist thinking that most readers familiar with the literature could have easily predicted the outcomes of these experiments. While validating known hypotheses is fundamental to science, it does feel like the contribution here is limited to that.

The paper's other main weakness is it's lack of scale (and proper analysis of scale), and the oddly poor performance of some models. I know this is an easy criticism to throw, but the largest model shown in the paper has 40M parameters and something like 18% test accuracy on an unperturbed training set. In contrast, a ResNet-152 from 2015 (He et al.) has a similary number of parameters and ~80% accuracy on the full ImageNet dataset, and an 11M parameter DenseNet (Abai et al) gets 60% accuracy on TinyImageNet.

> As a result, memorization is generally considered antithetical to generalization

I see this written a lot, but this is a demonstrably false narrative. I think it's been the pretty clear narrative even since the 2016/17 works of Zhang et al & Arpit et al (in their _Main Contributions_ section, they write: _"DNNs learn simple patterns first, before memorizing, [..] in other words, DNN optimization is content-aware, taking advantage of patterns shared by multiple training examples"_) that generalization and memorization are **not at odds**; i.e. deep models do both but we don't understand how much of which and to what degree they contribute to test performance. It's also been fairly clear since ~2017, including the work of Arpit et al, that DNNs learn in some kind of hierarchical order (or frequency-based, DNNs learn lower frequencies first [1]), even in the presence of label noise. The latter already suggests that intermediate representations should be amenable to have _general_ information extracted from them, even if the last classifying later "overfits".

[1] Neural Networks Learn Statistics of Increasing Complexity, Nora Belrose, Quintin Pope, Lucia Quirke, Alex Troy Mallen, Xiaoli Fern, ICML 2024
[2] Understanding intermediate layers using linear classifier probes, Guillaume Alain, Yoshua Bengio, 2016
[3] On the geometry of generalization and memorization in deep neural networks, Cory Stephenson, suchismita padhy, Abhinav Ganesh, Yue Hui, Hanlin Tang, SueYeon Chung, ICLR 2021

**Questions:**

I can't think of prior work having done this exact experiment, but it is very much in line with all that I know from the whole memorization vs generalization literature; and in that sense I come out of having read the paper with the feeling of not having learned anything. What is the one thing the authors think we can learn from this paper that's not in existing papers?

Coming back to scale, the effect is pretty minimal on AlexNet+TinyImagenet. Since this is the most "large-scale representative" of the tasks, it definitely begs the question of what actually happens at scale. Without training on all of ImageNet with a modern model, I wonder if something could be learned by repeating the experiment with e.g. 2x and 0.5x the number of parameters.

Section 5 constructs subspaces based on noisy labels for models without noisy labels (so presumably "no memorization"), and shows that this can be used with MACS to correctly retrieve the noised labels in some layer in most cases. I really fail to see how this "this supports the idea that memorization can not only coexist with generalization, but that in some cases memorization can be accompanied by superior generalization." I suspect this just shows that the subspace is expressive enough to separate somewhat arbitrary points, which I suspect can be explained by the occasionally high number of principal components. Conversely this may say nothing about generalization, since the underlying space is presumably a generalizing one, it just says that the overarching space of the subspace is also expressive enough to separate arbitrary points. We know this is the case from a long history of probing overparameterized deep neural networks. I would like the authors to expand on Section 5 and explain why they think it shows what they claim it shows.

Some suggested improvements:
- Table 1: use space between groups of 3 digits for large numbers
- Adam is a method with a paper that should be cited
- Figure 1: increase label size, and generally reconsider the zoom level of the plot, details are hard to see. Another tip is to hide the shared axes' labels to create more space for the figure (IIRC using pyplot just using `sharey=True` should accomplish this)
- Figure 2 should also have lines for the "Minimum Angle Subspace Classifier (MASC) Accuracy on Corrupted Training" lines of Figure 1, otherwise I don't see how to compare the two properly (and support the claim that "accuracies on the true training labels, as well as the test set are dramatically better here than with the experiments where subspaces were determined for the corrupted training data")
- The abstract is very long and I recommend being much much more concise. Again, what is the one thing the authors think we can learn from this paper that's not in existing papers?

---

> ### Author Response · Authors · 2024-11-25
> **Part-1 of author response to Reviewer vreZ**
>
> We thank the reviewer for their detailed review and thoughtful comments and questions. We must say we find ourselves unable to agree with the reviewer’s view that the results here are obvious in light of prior work (W2). We respond individually below.
>
> > W1: “The paper's main weakness is a fairly important lack of situating itself with respect to prior work. A number of papers have probed intermediate layers and latent representations of DNNs [e.g. 2,3], in an attempt to understand this very memorization-vs-generalization debate.
>
> Thank you. This point is well taken. We have now added a Related Work section (Section 2) that surveys prior work and situates this paper in context of that prior work.
>
> > W2: It does feel to me that many of the claims made in this paper are obvious in light of this literature. While the specific empirical investigation done here is novel to me, the conclusions drawn by the authors are not. It is hard to resist thinking that most readers familiar with the literature could have easily predicted the outcomes of these experiments. While validating known hypotheses is fundamental to science, it does feel like the contribution here is limited to that.”
>
> We disagree that the claims here are obvious and offer some justification below of why we believe the results here are not obvious.
> If there are specific experiments that the reviewer can point to from prior literature that, in their mind, render our results obvious, we would appreciate specific pointers that we can respond to.
>
>
> * While [Alain & Bengio, 2016 arXiv:1610.01644v4 updated in 2018] that the reviewer cites as [2], use linear probes on intermediate layers, they explicitly avoid doing so for networks trained with corrupted labels, since they believe that the probes would overfit. To quote their words (at the end Section 3.4): “Note that we also want to avoid a situation where our probes are simply overfitting on the features because there are too many features. It was recently demonstrated that very large models can fit random labels on ImageNet (Zhang et al., 2016). This is a situation that we want to avoid because the probe measurements would be entirely meaningless in that situation.”
>
> * [Stephenson et al, ICLR 2021] that the reviewer cites as [3] do study memorized models, but they conclude that memorization happens in later layers via circumstantial evidence: (1) Since rewinding early layer weights to their early stopping values recovers some generalization, but rewinding later layer weights doesn’t (Figure 4). (2) They also correlate this point with the decreasing object manifold’s radius and dimension (Figure 3).
> On the contrary, our results provide more direct evidence suggesting that later layers in most models investigated retain significant ability to generalize, and we are able to demonstrate the same without modifying the weights of the trained network. Thus, our results challenge the conclusions of [Stephenson et al, ICLR 2021] and therefore do not obviously follow from it.
>
>
> > Q1: I can't think of prior work having done this exact experiment, but it is very much in line with all that I know from the whole memorization vs generalization literature; and in that sense I come out of having read the paper with the feeling of not having learned anything. What is the one thing the authors think we can learn from this paper that's not in existing papers?
>
> Thank you for this question. We believe that the sentences below succinctly describes the central message of this paper:
> We demonstrate that Deep Network models trained using training data with shuffled labels to high training accuracy often have significant generalization ability that can be decoded easily from their internals and we design a technique to do so (the MASC classifier). While the ability of such memorized models to generalize early on in such training was known, the fact that layerwise outputs, especially in later layers largely retain this ability later in training was previously unknown and had not been demonstrated.
>
> EDIT: A few sentences above were edited following a discussion with reviewer vreZ.

---

> ### Author Response · Authors · 2024-11-25
> **Part-2 of author response to Reviewer vreZ**
>
> > W3: “The paper's other main weakness is it's lack of scale (and proper analysis of scale), and the oddly poor performance of some models. I know this is an easy criticism to throw, but the largest model shown in the paper has 40M parameters and something like 18% test accuracy on an unperturbed training set. In contrast, a ResNet-152 from 2015 (He et al.) has a similary number of parameters and ~80% accuracy on the full ImageNet dataset, and an 11M parameter DenseNet (Abai et al) gets 60% accuracy on TinyImageNet.”
>
> Thank you for this question. There is a technical reason for this, which we explain below.
>
> The reason for the “oddly poor performance of some models” is that we deliberately did not use regularization techniques such as dropout, or batch norm, to decouple the potential effects of explicit regularization on improved generalization performance of our probes.
>
> Indeed, this is fairly standard practice when studying the internal mechanisms of memorized models. E.g. [Stephenson et al, ICLR 2021] also deliberately do not use regularization and as a result their models also have poor performance (see e.g. Figure A.9, where their AlexNet model trained on CIFAR-100 has similarly poor performance).
>
> We have also briefly mentioned this point in the paper (Lines 182-185), since this may not be obvious.
>
> Likewise, we did not use early stopping, since we wish to study the internals of the model upon memorization. We have now added a new section Section A.4 in the appendix showing the comparison of test accuracy numbers with the early stopping of the model.
>
> > Q2: Coming back to scale, the effect is pretty minimal on AlexNet+TinyImagenet. Since this is the most "large-scale representative" of the tasks, it definitely begs the question of what actually happens at scale. Without training on all of ImageNet with a modern model, I wonder if something could be learned by repeating the experiment with e.g. 2x and 0.5x the number of parameters.
>
> Thank you for this question.
>
> With  AlexNet-Tiny ImageNet, the effect is in fact more pronounced than is apparent, and the reason for this perception is that the figure is ineffective in conveying this point due to the common scaling of the y axis and the relatively low chance prediction values for Tiny ImageNet (0.5%).
>
> Specifically, for AlexNet-Tiny ImageNet, in the table below, we list by what percentage the MASC classifier outperformed the model for the best layer for each corruption degree; we hope that the reviewer will agree that the effect is in fact significant, in this case too.
>
> | Corruption degree                                                                                   | 20%   | 40%   | 60%    | 80%    |
> |-----------------------------------------------------------------------------------------------------|-------|-------|--------|--------|
> | Subspace constructed using corrupted labels - Percentage the MASC classifier outperformed the model | 27.51 | 53.67 | 45.04  | 13.69  |
> | Subspace constructed using true labels - Percentage the MASC classifier outperformed the model      | 33.36 | 84.19 | 156.93 | 210.95 |
>
>
> Furthermore, we have added a separate Section A.5 in the Appendix with the AlexNet-Tiny ImageNet test results plotted separately and furthermore, we have also explicitly enumerated the test accuracy numbers in two tables in order to make this distinction clear.
>
> We thank the reviewer for pointing this out, since this may be a misconception that many readers might have had on looking at the previous plots.

---

> ### Author Response · Authors · 2024-11-25
> **Part-3 of author response to Reviewer vreZ**
>
> > W4: “As a result, memorization is generally considered antithetical to generalizationI see this written a lot, but this is a demonstrably false narrative. I think it's been the pretty clear narrative even since the 2016/17 works of Zhang et al & Arpit et al (in their Main Contributions section, they write: "DNNs learn simple patterns first, before memorizing, [..] in other words, DNN optimization is content-aware, taking advantage of patterns shared by multiple training examples") that generalization and memorization are not at odds; i.e. deep models do both but we don't understand how much of which and to what degree they contribute to test performance. It's also been fairly clear since ~2017, including the work of Arpit et al, that DNNs learn in some kind of hierarchical order (or frequency-based, DNNs learn lower frequencies first [1]), even in the presence of label noise. The latter already suggests that intermediate representations should be amenable to have general information extracted from them, even if the last classifying later "overfits".”
>
> Thank you for this observation. We don’t disagree with you at all on this. Since the sentence in question has the potential to lead to a misunderstanding, as you correctly point out, we have removed this sentence from the text, in order to minimize the possibility of such a misunderstanding.
>
> > Q3: Section 5 constructs subspaces based on noisy labels for models without noisy labels (so presumably "no memorization"), and shows that this can be used with MACS to correctly retrieve the noised labels in some layer in most cases. I really fail to see how this "this supports the idea that memorization can not only coexist with generalization, but that in some cases memorization can be accompanied by superior generalization." I suspect this just shows that the subspace is expressive enough to separate somewhat arbitrary points, which I suspect can be explained by the occasionally high number of principal components. Conversely this may say nothing about generalization, since the underlying space is presumably a generalizing one, it just says that the overarching space of the subspace is also expressive enough to separate arbitrary points. We know this is the case from a long history of probing overparameterized deep neural networks. I would like the authors to expand on Section 5 and explain why they think it shows what they claim it shows.
>
> Thank you. We agree and have removed this sentence from the paper. The intent of this experiment was simply to check if a generalized model can also support memorization in this manner by suitably training a MASC classifier to this end.
>
>
> > Some suggested improvements:
> > * Table 1: use space between groups of 3 digits for large numbers
> > * Adam is a method with a paper that should be cited
> > * Figure 1: increase label size, and generally reconsider the zoom level of the plot, details are hard to see. Another tip is to hide the shared axes' labels to create more space for the figure (IIRC using pyplot just using sharey=True should accomplish this)
> > * Figure 2 should also have lines for the "Minimum Angle Subspace Classifier (MASC) Accuracy on Corrupted Training" lines of Figure 1, otherwise I don't see how to compare the two properly (and support the claim that "accuracies on the true training labels, as well as the test set are dramatically better here than with the experiments where subspaces were determined for the corrupted training data")
> > * The abstract is very long and I recommend being much much more concise. Again, what is the one thing the authors think we can learn from this paper that's not in existing papers?
>
> Thank you for these very helpful suggestions that will improve the paper. We have made changes in response to each of these suggestions, as detailed below:
>
> * We have used space between groups of 3 digits for large numbers.
> * We have cited Adam: A method for stochastic optimization paper.
> * Figure 1: We have increased the label size, and in response to another reviewer’s suggestion, moved results of some of the models to the Appendix.
> * Thank you, this is a good point which will help in comparison. We have added Minimum Angle Subspace Classifier (MASC) Accuracy on Corrupted Training as well as Minimum Angle Subspace Classifier (MASC) Accuracy on Testing from experiment 1 in Figure 2.
> * We have shortened the abstract.
>
> We are happy to respond to any further questions/suggestions or offer additional clarifications, should the reviewer need them.

---

> > ### Author Response · Authors · 2024-11-30
> > **Gentle reminder to Reviewer vreZ**
> >
> > Dear Reviewer,
> >
> > We hope this finds you well.
> >
> > In case it escaped your attention, we want to remind you about our response to your review.
> >
> > As you might be aware, the revised deadline for reviewer responses is approaching, i.e. Dec 2nd, AoE.
> >
> > Once again, we appreciate your time and efforts. We look forward to constructively engaging with you.

---

### Official Review · Reviewer_ALTv · 2024-11-03

**Soundness:** 2
**Presentation:** 3
**Contribution:** 2
**Rating:** 3
**Confidence:** 4

**Summary:**

This paper proposes to investigate the sometimes confusing relationship between neural network generalization and memorization. To this end, the authors propose a method of analysis they call Minimum  Angle Subspace Classifier (MASC) which is a kind of combination between nearest neighbour classifier and a dimensional reduction method: it's a sort-of nearest subspace classifier with distance defined via the angle. Using MASC as their probe they find that the internal representations of the neural networks are able to generalize significantly better than the neural networks themselves.

**Strengths:**

- Understanding the relationship between generalization and memorization is an important and worthy area of study.
- Good empirical breadth, with the analysis being applied across 5 standard datasets and three different architectures.
- The authors seem reasonably well versed in at least some of the vast literature on the topic of memorization and generalization.

**Weaknesses:**

- I believe there is a methodological issue with the MASC based analysis. The original neural networks are trained to either 500 epochs or 99% - 100% percent accuracy on the training set. This means that the models are likely to be overfit on the training data, especially when training with corrupted data. This may well be the intention of the authors, but it has implications for their conclusions. In their analysis, the authors take a representation drawn from different layers in the architecture and build an alternate classifier on that representation. If the classifier is sufficiently regularized (which their MASC appears to be in at least some cases) and the representation still possesses sufficient variability across the input examples (i.e. the input data points are not collapsed on one-another), then it's not surprising that the MASC classifier is able to exceed the unregularized and overfit neural network classifier.

- There is insufficient analysis of the properties of the MASC as an experimental probe. The degree to which the MASC is regularized is a very important property for the interpretation of the results. This aspect of the MASC is only sparingly discussed mainly in the appendix and it's unclear how this property globally impacts the conclusions of the paper.

- There is a largely undiscussed lack of consistency across empirical results. The authors present a healthy breadth of experiments but across the main findings presented in Figs, 1, 2 and 3, the authors mainly focus on the MLP experiments and to some extent the CNN results. They largely neglect to incorporate the AlexNet results into their narrative. This is understandable, because these are often inconsistent with the pattern of results across the other datasets and architectures, but it leaves the reader questioning the generality of the stated findings.

- A very important baseline or control is missing. The MASC needs to be applied to a randomly initialized model (for each architecture). My interpretation of section 4 is that the authors are claiming that the generalization observed using their MASC (on the true uncorrupted training data) compared to the original neural network prediction is revealing hidden but learned generalization ability of the internal representation of the neural network. But this isn't necessarily so. The MASC classifier on a random projection of the input could potentially do just as well. Indeed it seems that as the level of corruption increases,  the dimensionality of the embedding is the most important determiner of MASC performance. This would likely also be predicted for a random embedding with a sufficiently regularized classifier.

- The paper occasionally slides into nearly nonsensical rhetoric. For example: in line 76 the authors write: "We ask, why it is that in models trained with shuffled labels do we have poor generalization accompanying perfect / high training accuracy." The obvious answer is: "because there is little or no common structure between the training set and the test set on which generalization is evaluated. Most of the paper is more lucid than this would imply, but statements like this weakens the overall strength of the message of the paper.

Clarity: The paper is readable though its clarity would benefit significantly from more formal, mathematical definitions and descriptions of the different empirical probes that are used. It took me a while to decipher what the training set was for the "MASC Accuracy on Corrupted Training".

Overall, the paper offers little novel insight into the relationship between memorization and classification. The results are largely consistent with previous findings (cited in the paper) and I do not see how this paper contributes significantly to that literature. Specifically, the idea that hidden layer representations can possess information about the nature of the data and/or task that is not conveyed to later layers in neural networks trained with corrupted data is not a significant contribution beyond the finding of, for example, Arpit et al (2017) and Zhang et al (2017).

**Questions:**

I would like the authors to respond to points I listed under weaknesses.

**Details Of Ethics Concerns:**

None.

---

> ### Author Response · Authors · 2024-11-25
> **Part-1 of author response to Reviewer ALTv**
>
> We thank the reviewer for the detailed and constructive review. We appreciate their recognition of the strengths of this work. We respond individually to the reviewer’s points on weaknesses/questions below.
>
> >  W1: I believe there is a methodological issue with the MASC based analysis. The original neural networks are trained to either 500 epochs or 99% - 100% percent accuracy on the training set. This means that the models are likely to be overfit on the training data, especially when training with corrupted data.This may well be the intention of the authors, but it has implications for their conclusions. In their analysis, the authors take a representation drawn from different layers in the architecture and build an alternate classifier on that representation. If the classifier is sufficiently regularized (which their MASC appears to be in at least some cases) and the representation still possesses sufficient variability across the input examples (i.e. the input data points are not collapsed on one-another), then it's not surprising that the MASC classifier is able to exceed the unregularized and overfit neural network classifier.
> >
> > W2: There is insufficient analysis of the properties of the MASC as an experimental probe. The degree to which the MASC is regularized is a very important property for the interpretation of the results. This aspect of the MASC is only sparingly discussed mainly in the appendix and it's unclear how this property globally impacts the conclusions of the paper.
>
> Thank you for this question. We don’t believe that MASC is highly regularized, as explained below.
> * Indeed, the intention is to fit the training data to perfect/high training accuracy.
> * Although we do not incorporate explicit regularization in the MASC classifier, it is an interesting question if MASC might have implicit regularization which is causing the improved generalization performance. We believe that this is not the case and the strongest evidence we have is from the experiments in Section 6 where we take a generalized model (i.e. a model trained with uncorrupted data) and apply a MASC classifier with subspaces corresponding to corrupted data (i.e. data with shuffled labels). If the MASC classifier were indeed strongly regularized, we would expect it to be unable to memorize this corrupted data. However, we find that it is able to memorize the shuffled training labels with very high accuracy in multiple models (MLP-CIFAR-10, AlexNet-CIFAR-100 and AlexNet-Tiny ImageNet), and this is especially the case with the larger models (See Figure 3, 13).

---

> ### Author Response · Authors · 2024-11-25
> **Part-2 of author response to Reviewer ALTv**
>
> > W3: There is a largely undiscussed lack of consistency across empirical results. The authors present a healthy breadth of experiments but across the main findings presented in Figs, 1, 2 and 3, the authors mainly focus on the MLP experiments and to some extent the CNN results. They largely neglect to incorporate the AlexNet results into their narrative. This is understandable, because these are often inconsistent with the pattern of results across the other datasets and architectures, but it leaves the reader questioning the generality of the stated findings.
>
> Thank you for pointing this out. It isn’t true that AlexNet results are particularly inconsistent with the other results, and this unfortunate perception is the result of the plot being ineffective by virtue of the y axis ranges being too large and the relatively low chance accuracy for AlexNet on Tiny ImageNet (0.5%).
>
> Specifically, for AlexNet-Tiny ImageNet, in the table below, we list by what percentage the MASC classifier outperformed the model for the best layer for each corruption degree; we hope that the reviewer will agree that the effect is in fact significant, in this case too.
>
> | Corruption degree                                                                                   | 20%   | 40%   | 60%    | 80%    |
> |-----------------------------------------------------------------------------------------------------|-------|-------|--------|--------|
> | Subspace constructed using corrupted labels - Percentage the MASC classifier outperformed the model | 27.51 | 53.67 | 45.04  | 13.69  |
> | Subspace constructed using true labels - Percentage the MASC classifier outperformed the model      | 33.36 | 84.19 | 156.93 | 210.95 |
>
> Furthermore, we have added a separate Section A.5 in the Appendix with the AlexNet-Tiny ImageNet test results plotted separately and furthermore, we have also explicitly enumerated the test accuracy numbers in two tables in order to make this distinction clear, which also emphasizes the consistency of our results on AlexNet. Thank you for bringing our attention to the fact that the AlexNet results hadn’t received adequate exposition – a shortcoming, we have now fixed. These points are also true, similarly for AlexNet trained on CIFAR-100.

---

> ### Author Response · Authors · 2024-11-25
> **Part-3 of author response to Reviewer ALTv**
>
> > W4: A very important baseline or control is missing. The MASC needs to be applied to a randomly initialized model (for each architecture). My interpretation of section 4 is that the authors are claiming that the generalization observed using their MASC (on the true uncorrupted training data) compared to the original neural network prediction is revealing hidden but learned generalization ability of the internal representation of the neural network. But this isn't necessarily so. The MASC classifier on a random projection of the input could potentially do just as well. Indeed it seems that as the level of corruption increases,  the dimensionality of the embedding is the most important determiner of MASC performance. This would likely also be predicted for a random embedding with a sufficiently regularized classifier.
>
> This is an excellent point and we are grateful that the reviewer brought it up. For each model investigated (except AlexNet), we have now run this control experiment and the results are available in Section A.6 of the Appendix. First of all, we note that this class of control experiments has previously been run on Deep Networks (see e.g. [Alain & Bengio, “Understanding intermediate layers using linear classifier probes, ICLR Workshop, 2017]) with fairly high accuracies reported, albeit not in the memorized networks setting.
>
> We find that indeed accuracies of the MASC classifier on the random initialization outperforms the network, except for low corruption degrees (i.e. <=20% corruption degree). However, in the experiments where subspaces are trained on corrupted training data from corrupted models, by-and-large, the MASC classifier usually, and on at least one layer outperforms the MASC classifier trained on the random initialization with exceptions being the 80% corruption degree models on MLP-MNIST and 100% corruption degree on CNN-FashionMNIST. Notably, for the experiments where subspaces are constructed with true labels on corrupted models, the MASC classifier on these models outperforms the MASC classifier on random initializations usually and certainly in at least one layer on every model tested. These results are consistent with the main message of the paper, namely that even with memorized models, the layerwise representations of the models are organized in a manner that they develop significant ability to generalize during training over and above that bestowed by a random initialization, and in particular, they do not lose this ability, as one might have naively expected, due to label noise. If they were losing this ability, then the MASC classifier on the subspaces would end up performing significantly worse than the MASC classifier run on randomly initialized models.
>
> The experiment on the AlexNet-Tiny ImageNet models is running at this time and we expect to update the manuscript with its results before the end of the Discussion period.
>
> > W5: The paper occasionally slides into nearly nonsensical rhetoric. For example: in line 76 the authors write: "We ask, why it is that in models trained with shuffled labels do we have poor generalization accompanying perfect / high training accuracy." The obvious answer is: "because there is little or no common structure between the training set and the test set on which generalization is evaluated. Most of the paper is more lucid than this would imply, but statements like this weakens the overall strength of the message of the paper.
>
> Thank you. We have removed this sentence and replaced it with one that frames this in terms that are more direct and hopefully in a manner that doesn’t distract from the message of the paper.

---

> ### Author Response · Authors · 2024-11-25
> **Part- 4 of author response to Reviewer ALTv**
>
> > W6: Clarity: The paper is readable though its clarity would benefit significantly from more formal, mathematical definitions and descriptions of the different empirical probes that are used. It took me a while to decipher what the training set was for the "MASC Accuracy on Corrupted Training".
>
>
> Thank you for this point. We have rewritten parts of the Methodology section  (Lines 159-161, 202-203), in response. We hope that the exposition is clearer, as a result. We are also working on improving the exposition further, including by adding a more detailed algorithm, which we expect to incorporate in an update by the end of the Discussion period.
>
> > W7: “Overall, the paper offers little novel insight into the relationship between memorization and classification. The results are largely consistent with previous findings (cited in the paper) and I do not see how this paper contributes significantly to that literature.” Specifically, the idea that hidden layer representations can possess information about the nature of the data and/or task that is not conveyed to later layers in neural networks trained with corrupted data is not a significant contribution beyond the finding of, for example, Arpit et al (2017) and Zhang et al (2017).
>
> We must say we disagree with this point.
>
> On the one hand, it was previously thought that intermediate layer representations would also inevitably have poor generalization due to overfitting for the setting of memorized networks. For example, [Alain & Bengio, 2016 arXiv:1610.01644v4 updated in 2018], who use linear probes on intermediate layers, explicitly avoid probing networks trained with corrupted labels, since they believe that the probes would inevitably overfit. To quote their words (at the end Section 3.4): “Note that we also want to avoid a situation where our probes are simply overfitting on the features because there are too many features. It was recently demonstrated that very large models can fit random labels on ImageNet (Zhang et al., 2016). This is a situation that we want to avoid because the probe measurements would be entirely meaningless in that situation.”
>
> In this context, therefore, our work adds an important and surprising result to the field, which we believe has the potential to contribute to further advances towards a fundamental understanding of generalization and memorization.
>
> Secondly, [Stephenson et al, ICLR 2021] do study memorized models, but they conclude that memorization happens in later layers via circumstantial evidence: (1) Since rewinding early layer weights to their early stopping values recovers some generalization, but rewinding later layer weights doesn’t (Figure 4). (2) They also correlate this point with the decreasing object manifold’s radius and dimension (Figure 3).
>
> On the contrary, our results provide more direct evidence suggesting that later layers in most models investigated retain significant ability to generalize, and we are able to demonstrate the same without modifying the weights of the trained network. Thus, our results challenge the conclusions of [Stephenson et al, ICLR 2021] and therefore do not obviously follow from it.
> Again, in this context, we believe our work adds an important set of results that advance the field, by challenging the conclusions of a previous line of experiments that offered circumstantial evidence for a hypothesis.
>
>
> We are happy to respond to any further questions/suggestions or offer additional clarifications, should the reviewer need them.
>
> EDIT: A few sentences above were edited following a discussion with reviewer vreZ.

---

> > ### Author Response · Authors · 2024-11-30
> > **Gentle reminder to Reviewer ALTv**
> >
> > Dear Reviewer,
> >
> > We hope this finds you well.
> >
> > In case it escaped your attention, we want to remind you about our response to your review.
> >
> > As you might be aware, the revised deadline for reviewer responses is approaching, i.e. Dec 2nd, AoE.
> >
> > Once again, we appreciate your time and efforts. We look forward to constructively engaging with you.

---

### Official Review · Reviewer_UhrT · 2024-11-04

**Soundness:** 1
**Presentation:** 1
**Contribution:** 1
**Rating:** 1
**Confidence:** 4

**Summary:**

This paper investigates the generalization and memorization phenomena in training overparameterized models with the presence of label noise.  The authors propose MASC, which matches the sample representation angle with each class's representation primary subspace to determine the sample's possible true class. And they state the method can decouple generalization from overfitted network learned from noisy labels.

**Strengths:**

Very comprehensive experiments.

**Weaknesses:**

1. Lack of literature: First, the paper lacks related works that don't provide pictures of previous work, the discussion of previous work only appears in the first paragraph of the introduction and most papers are focused on experiments. While there are many theoretical papers trying to understand the problem[1][2].
2. Novelty: The entire Sec.3 tries to convey the idea: that the top component of the corrupted model still contains the class information to some degree. That is not a surprise since [1][2] both indicate that the model first learns the primary eigenspace that contains the correct label information and only learns and memorizes the label noise slowly in the later stage. The proposed method, which tries to cut off the representation space and only keep the main component using PCA, is not that novel because it is also the top eigenspace. Although learning the label noise reduces the energy of the clean label subspace that was learned at the beginning of training, it is not a surprise that the top eigenspace still contains meaningful information.
3. Novelty: Sec.4 tries to convey the idea: that the MASC built on a corrupted model using the correct label is a good classifier to reflect the true class distribution. However, the MASC is just a clustering algorithm, where we use the PCA to get a class vector and then use angle (cosine similarity) to cluster the example. It even somehow works with the original picture. Since the model already established some representation while learning the clean space, it is expected that it can perform well.

4. Writing: The paper is hard to read and follow due to its dense language, complex sentence structures, and grammar errors.  For example, ln 76-96 has multiple dense complex sentences and creates a hinge for readers to understand the main contribution. Ln 304-318 are also hard to follow.  The Methodology part only verbally describes the algorithm as well as the terminology part, it took me much longer time to understand the algorithm. For grammar errors, ln 305 we wanted to -> want, etc. Also, the table elements are not well separated in Table 1, Fig 1,2,3 exceeds the page margin.

[1] Gradient Descent with Early Stopping is Provably Robust to Label Noise for Overparameterized Neural Networks.
[2] When and how epochwise double descent happens.

**Questions:**

See weakness.

---

> ### Author Response · Authors · 2024-11-25
> **Part-1 of author response to Reviewer UhrT**
>
> We thank the reviewer for their detailed comments and we appreciate that they found our experiments comprehensive. We have addressed the previous lack of detailed exposition of Related Work. We however, fundamentally disagree on the reviewer’s contention with respect to the apparent lack of novelty in our work. We provide a detailed response, which we believe strongly rebuts these points.
>
> > W1: Lack of literature: First, the paper lacks related works that don't provide pictures of previous work, the discussion of previous work only appears in the first paragraph of the introduction and most papers are focused on experiments. While there are many theoretical papers trying to understand the problem[1][2].
>
> Thanks for pointing this out. We have now added a Related Work section that puts our work in the broader context of work in the field.

---

> ### Author Response · Authors · 2024-11-25
> **Part-2 of author response to Reviewer UhrT**
>
> > W2: Novelty: The entire Sec.3 tries to convey the idea: that the top component of the corrupted model still contains the class information to some degree. That is not a surprise since [1][2] both indicate that the model first learns the primary eigenspace that contains the correct label information and only learns and memorizes the label noise slowly in the later stage. The proposed method, which tries to cut off the representation space and only keep the main component using PCA, is not that novel because it is also the top eigenspace. Although learning the label noise reduces the energy of the clean label subspace that was learned at the beginning of training, it is not a surprise that the top eigenspace still contains meaningful information.
>
> We disagree with the reviewer’s contention that our work lacks novelty in light of the references cited.
>
> For example, [1] studies the intermediate representations primarily at the early stopping point. Here, they establish that the weights of the network at this point are significantly closer to the initialization than they are to the final “overfit” network (Figure 3). Secondly, they plot loss histograms early on in training and also late in training (Figure 5b, 5c). They find that early on in training, these loss histograms are well separated, supporting generalization, whereas late in training, there is significant overlap which supports poor generalization and overfitting.
> This result in fact runs contrary to our results which demonstrate the ability to extract test accuracies comparable to early stopping numbers very late in training. We have added a comparison of our MASC test accuracies with those of test accuracies with early stopping of the model in Section A.4 of the Appendix to clearly demonstrate this point.
>
> As the reviewer correctly notes that the act of learning “reduces the energy of the clean label subspace that was learned at the beginning of training.” Indeed, previous work has conveyed that the obvious expectation was that this process of subsequent training removes the information relevant to generalization.
> For example, [Alain & Bengio, 2016 arXiv:1610.01644v4 updated in 2018], who use linear probes on intermediate layers, explicitly avoid probing networks trained with corrupted labels, since they believe that the probes would inevitably overfit. To quote their words (at the end Section 3.4): “Note that we also want to avoid a situation where our probes are simply overfitting on the features because there are too many features. It was recently demonstrated that very large models can fit random labels on ImageNet (Zhang et al., 2016). This is a situation that we want to avoid because the probe measurements would be entirely meaningless in that situation.”
>
>
> In this context, therefore, our work adds an important and surprising result to the field, which we believe has the potential to contribute to further advances towards a fundamental understanding of generalization and memorization.
> With [2], they study the case of epoch-wise double descent with a linear model and find that for a range of noise values, the generalization error falls in later epochs. We note that for the models we have studied, the model’s generalization does not behave in a manner consistent with the predictions in this paper. Secondly, the prediction of double descent only is for a range of noise values. We however find that the MASC classifier is able to obtain improved generalization for the entire range of corruption degrees, especially for subspaces corresponding to true labels. Thirdly, their theoretical model predicts that the effects described do not occur for large-sized networks. Indeed, they find experimentally that ResNet-18 (11M parameters) does not experimentally exhibit the predictions of their model. However, in our hands, AlexNet (~40M parameters) is able to show robust results in our experiments. We therefore believe that our models do not occupy the regime that this paper [2] treats. As such, we disagree with the claim that this paper predicts our results, for these reasons.
>
> EDIT: A few sentences above were edited following a discussion with reviewer vreZ.

---

> > ### Comment · Reviewer_vreZ · 2024-11-26
> >
> > > We note that Yoshua Bengio is an author on Alain & Bengio, 2016 as well as Arpit et al, (2017), so we believe that the authors of Arpit et al, 2017 would also find our results surprising.
> >
> > Sorry for jumping in here, but I must push back on this since I did write in my review that, in Arpit et al's Main Contributions section, they write: "DNNs learn simple patterns first, before memorizing, [..] in other words, DNN optimization is content-aware, taking advantage of patterns shared by multiple training examples".
> >
> > I do appreciate your work and stand by the score I gave it, but I do worry that your are overindexing here on this one remark by Alain. I'm unsure how much "reading between the lines" this requires, but to me Arpit et al's diversity of results and its subtext (and number of authors) are very much reflective of the uncertainty and internal debate in late 2016 & 2017 that I'm aware existed in Mila/the Bengio lab around the result of Zhang on overfitting. In that sense, I would encourage you to refrain from speculating what these authors would have found surprising in 2017, and would find surprising still in 2024--since, as Reviewer UhrT points out, there has since been work formalizing these phenomena and tipping the scales very much on one side.
> >
> > I would also encourage you to not overindex on your own results, they tell a fairly specific story about fairly specific settings that we must be careful not to overgeneralize from (no pun intended).

---

> > > ### Author Response · Authors · 2024-11-27
> > > **Reponse to Comment by Reviewer vreZ**
> > >
> > > Thank you. We appreciate this point. We see that our comment may be seen as one that is speculating about the opinions of all authors in [Arpit et al, 2017]. We are therefore removing references to this suggestion with a note at the end indicating that we have edited our response following the present discussion.
> > >
> > > That said, we stand by our quote of the paragraph from [Alain & Bengio, 2016]. We note that the paragraph in question was added in the arxiv version uploaded by Alain & Bengio on Nov 22, 2018, well over a year after the appearance of [Arpit et al, 2017] in ICML, 2017.
> > >
> > > Your last point (and unintended pun) are also well taken! :)

---

> ### Author Response · Authors · 2024-11-25
> **Part-3 of author response to Reviewer UhrT**
>
> > W3: Novelty: Sec.4 tries to convey the idea: that the MASC built on a corrupted model using the correct label is a good classifier to reflect the true class distribution. However, the MASC is just a clustering algorithm, where we use the PCA to get a class vector and then use angle (cosine similarity) to cluster the example. It even somehow works with the original picture. Since the model already established some representation while learning the clean space, it is expected that it can perform well.
>
> We strongly disagree. We haven’t seen any prior work suggest that this is obvious. In fact, as mentioned previously, [Alain & Bengio, 2016] explicitly indicate that they expect the opposite to happen, if intermediate representations are probed.
>
> > W4: Writing: The paper is hard to read and follow due to its dense language, complex sentence structures, and grammar errors. For example, ln 76-96 has multiple dense complex sentences and creates a hinge for readers to understand the main contribution. Ln 304-318 are also hard to follow. The Methodology part only verbally describes the algorithm as well as the terminology part, it took me much longer time to understand the algorithm. For grammar errors, ln 305 we wanted to -> want, etc. Also, the table elements are not well separated in Table 1, Fig 1,2,3 exceeds the page margin.
>
> Thanks for pointing this out. In response, we have added a Main Contributions subsection to make our contributions clear. We are working on rewriting the text in lines 304-318, in order to improve clarity; we expect to incorporate the same in a version that will be uploaded before the end of the Discussion period. Likewise, we are working on revamping the Methodology section and including a detailed and more formal exposition of the algorithm, which will appear in the next version. Grammar errors in line 305 have been corrected. Issues with the table elements and those with Figure 1, 2 and 3 have also been addressed.
>
> We are happy to respond to any further questions/suggestions or offer additional clarifications, should the reviewer need them.

---

> > ### Author Response · Authors · 2024-11-30
> > **Gentle reminder to Reviewer UhrT**
> >
> > Dear Reviewer,
> >
> > We hope this finds you well.
> >
> > In case it escaped your attention, we want to remind you about our response to your review.
> >
> > As you might be aware, the revised deadline for reviewer responses is approaching, i.e. Dec 2nd, AoE.
> >
> > Once again, we appreciate your time and efforts. We look forward to constructively engaging with you.

---

### Author Response · Authors · 2024-11-25
**Overall Response**

We thank all the reviewers for taking the time to read our paper carefully and post detailed constructive reviews that we have found useful. These have led to a number of changes that we believe materially improve the paper.

We appreciate the reviewers noting that the paper considers an important area of study (Reviewers vreZ, ALTv) whose literature we are well-versed with (Reviewer ALTv) and for our clearly formulated experiments (Reviewer EEmZ)  and design of concrete techniques (Reviewer Hw6o) used in the experiments. Reviewer vreZ has stated that we have investigated important questions in a novel way and acknowledges that the paper is well-written and that our claims are mostly backed by evidence. Multiple reviewers have recognized the empirical depth of our work that spans comprehensive experiments on multiple models and datasets (Reviewers UhrT, ALTv and EEmZ).

We apologize for taking our time, but we have formulated comprehensive responses to each reviewer, substantively responding to their concerns. These responses appear after each review.

Below, we summarize the major changes that have been made in response to the reviews.

Summary of changes:
---
1. Abstract has been shortened.

2. Main contributions has been added in introduction.

3. Section 2. Related work has been added.

4. Sections added to Appendix, including those describing the results of new experiments that were performed in response to reviewer comments:

     * A.4 New results on early stopping accuracies and comparison with MASC decoding results

          Results for MLP-MNIST, CNN- FashionMNIST and AlexNet-CIFAR100 shown in Figures 5, 6.

     * A.5 Experiments with AlexNet model trained on Tiny ImageNet

          Figures 7, 8 and Table 4, 5 are added.

     * A.6 New results on randomly-initialized control models and their comparison with MASC accuracies on trained models.

          Results for MLP-MNIST and CNN- FashionMNIST in Figures 9, 10  are added.

     * A.8 Experimental results on two additional models (MLP-CIFAR10 and AlexNet-Tiny ImageNet)

          Figures 13, 14, 15  are added.

5. In the interest of better readability, Figure 1, 2, and 3 have been redesigned to include only the results of one dataset per model (namely MLP-MNIST, CNN-Fashion-MNIST and AlexNet-CIFAR-100). Results of other models have been moved to Appendix Section A.8.

6. Multiple minor edits have been made, in response to reviewer comments.

EDIT (Nov 27, 2024): We have now uploaded an updated version of the manuscript, which in addition to the above changes, incorporates the following changes:

* In section A.6 new results on randomly-initialized control AlexNet - Tiny ImageNet and their comparison with MASC accuracies on trained models is added (Figure 11 and 12).

* Following reviewer suggestions, we have made further changes to related work section.

* New section A.7  with a detailed and more formal exposition of the minimum angle subspace classifier algorithm is added (Algorithm 1, 2, and 3 ).

We remain available to respond to any further questions or concerns of the reviewers, whose time we much appreciate.

---

### Meta-Review · Area_Chair_iVXT · 2024-12-19

**Metareview:**

The paper investigates the memorization phenomenon in deep networks and raises an intruiging hypothesis that deep neural networks memorize much less than previous research would suggest. However, the work fell short of defining convincing metrics and deriving clear insights. Reviewers have raised significant concerns that were not cleared during the rebuttal phase. As such, I am recommending rejection at this stage. Thank you for submitting your paper for consideration in the ICLR program and I hope the comments will be helpful in improving the paper.

**Additional Comments On Reviewer Discussion:**

While the rebuttal added clarity to the methodology, refined comparisons with prior work, and strengthened the empirical evidence, reviewers remained unconvinced of the novelty and broader applicability, particularly in larger-scale settings. Despite notable contributions, concerns about the interpretation and generality of the findings influenced the final decision.

---

### Decision · Program_Chairs · 2025-01-22

Reject